# BrainPro: Towards Large-scale Brain State-aware EEG Representation Learning

## Abstract

Electroencephalography (EEG) reflects underlying brain states, whose activities are distributed across brain regions and manifest as spatial patterns on the scalp. Learning these spatially structured, state-related patterns requires consistent spatial representations across datasets. However, existing EEG foundation models are typically based on self-attention, which does not preserve location-specific information and struggles to align signals recorded with different channel configurations. Moreover, brain states contain both shared and state-specific regional activity, suggesting that learning neurophysiologically plausible, state-aware representations can complement the shared representations targeted by current models and improve downstream decoding. To address these limitations, we propose BrainPro, a large EEG model that combines a retrieval-based spatial learning mechanism for cross-layout spatial alignment with a brain state-decoupling module that learns both shared and state-specific representations through parallel encoders and region-aware reconstruction. Pre-trained on a large EEG corpus, BrainPro achieves state-of-the-art performance across nine public BCI datasets spanning emotion, motor, speech, stress, mental disease, and attention tasks. Analyses of spatial filters, channel-drop robustness, and encoder contributions further validate the effectiveness of its spatial alignment and state-aware pathways. These results show that BrainPro achieves improved interpretability of learned spatial patterns and produces representations that benefit diverse EEG decoding tasks.

## 1 Introduction

Electroencephalography (EEG) reflects a dynamic mixture of underlying *brain states*-the distributed patterns of neural activity linked to physiological or cognitive processes and their behavioral consequences (Greene et al., 2023). Neuroimaging work shows that affective processes engage motor cortical regions, which are also involved in motor-related activity (Schnitzler et al., 1997), while emotions additionally recruit limbic and paralimbic areas (Putkinen et al., 2020). This illustrates the coexistence of shared and state-specific components across brain states. Although EEG captures these patterns only at a coarse spatial scale, it nonetheless reflects meaningful state-associated differences in spectral power and scalp distributions (Greene et al., 2023). Together, these observations suggest that useful EEG representations should be sensitive to both broadly shared dynamics and spatial variations associated with different brain states.

Recent EEG foundation models (EFMs) have improved transferability through large-scale pre-training (Zhou et al., 2025). However, most EFMs (Jiang et al., 2024; Wang et al., 2024; 2025) rely on masked reconstruction or related objectives (Yang et al., 2023), which are designed to learn broadly shared EEG representations but provide no mechanism to capture additional state-specific structure linked to different underlying *brain states*. Consequently, factors such as affective, motor, or cognitive processes become entangled within a single latent space, limiting the model's ability to form complementary state-aware representations. Introducing dedicated state-specific representations alongside shared ones is neurophysiologically plausible, since different brain states involve both common and state-dependent regional activity. These additional state-aware features may also benefit downstream tasks that depend on subtle state-related variations in EEG signals.

A second challenge concerns the modeling of spatial information. EEG datasets differ substantially in electrode layouts, including channel count, spacing, and regional coverage. These differences

make it difficult for implicit mechanisms such as channel embeddings or attention (Jiang et al., 2024; Wang et al., 2025) to capture spatial patterns in a consistent way. Since different neural processes can influence the spatial distribution of EEG activity, learning spatial features in a manner that is consistent across heterogeneous montages is important for representing potential brain-state-related representations.

Together, these limitations motivate the central question of this work: **How can we explicitly learn brain-state-aware EEG representations in the presence of heterogeneous electrode montages?**

To address these challenges, We introduce **BrainPro**, a neurophysiologically grounded large EEG model. BrainPro consists of:

- **Retrieval-based spatial learning**, which aligns dataset-specific montages to a universal channel and region template and learns both channel- and region-level spatial filters. This yields explicit (justified at Appendix D) and interpretable spatial representation learning across datasets (shown in Figure 3).

- **Brain-state decoupling**, which employs a shared encoder and multiple state-specific encoders trained with a decoupling loss to model complementary representations associated with different brain states, reflecting the fact that distinct mental processes can produce overlapping yet partially differentiable spatiotemporal patterns on the scalp.

- **Region-aware masked reconstruction**, which further extends standard masked-modeling objectives by emphasizing spatial regions relevant to each brain state, providing neurophysiologically informed supervision during pre-training.

Together, these components form a unified pre-training framework designed to make brain-state-related representations more explicit while handling heterogeneous electrode montages in a principled manner. Pre-trained on 2,180 hours of EEG from multiple datasets, BrainPro demonstrates strong performance across nine downstream BCI tasks spanning motor, emotion, speech, stress, mental disease, and attention tasks. Ablation studies show that retrieval-based spatial learning, brain-state decoupling, and region-aware reconstruction each contribute to the overall performance. In addition, qualitative inspection of the learned spatial filters reveals patterns that are consistent with commonly reported scalp-level differences across processes, suggesting that BrainPro captures meaningful structure related to brain states.

## 2 PRELIMINARIES

**Definition 1** (Brain State). A *brain state* refers to a distributed pattern of neural activity linked to a broad physiological or cognitive condition (Greene et al., 2023). Such states influence large-scale network dynamics and are reflected in EEG through characteristic temporal and spatial patterns on the scalp. In this work, a brain state denotes the annotated process assumed to be dominant in an EEG segment.

**Definition 2** (Overlapping Brain States). Different brain states may recruit some of the same cortical regions even when they arise from distinct underlying processes. We say two states *overlap* when they involve partially shared spatial patterns, for example, the engagement of motor cortical regions by both affective and motor-related processes (Schnitzler et al., 1997; Putkinen et al., 2020).

*Remark* 1 (Shared vs. State-Specific Representations). Because different processes can involve both shared regions (e.g., sensorimotor cortex) and process-specific areas (e.g., limbic structures for emotion) (Greene et al., 2023), it is useful to distinguish between (i) shared patterns that appear across multiple states and (ii) variations more closely associated with particular processes. Modeling both aspects can provide more informative representations for downstream decoding.

**Definition 3** (Explicit Spatial Encoding). *Explicit spatial encoding* refers to learning spatial filters whose weights are tied across channels that occupy the same or nearby anatomical locations, even when datasets differ in their electrode configurations. By enforcing consistent spatial filtering for spatially corresponding channels, the model can capture location-specific patterns in a way that generalizes across heterogeneous montages. This supports learning spatial variations related to different underlying processes; further justification is provided in Appendix D.

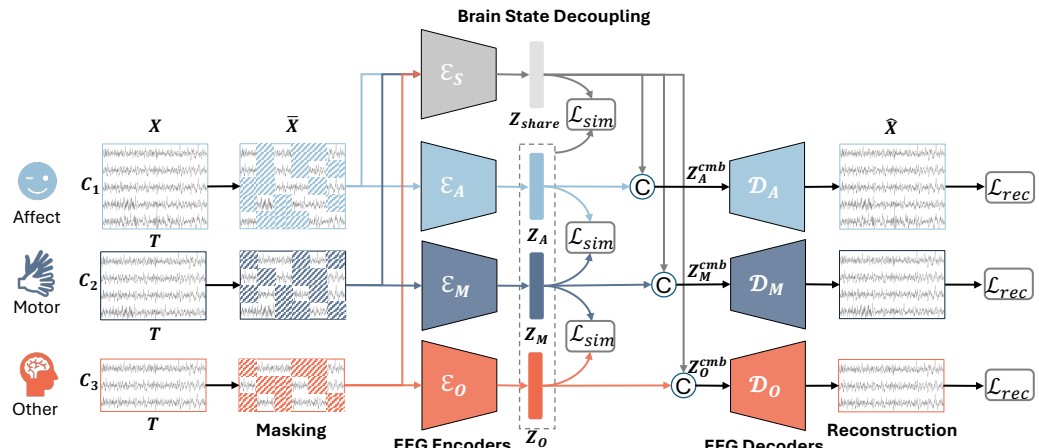

Figure 1: BrainPro framework. BrainPro consists of a shared encoder, $\mathcal{E}_S$, for shared EEG representations and multiple brain state-specific encoders ($\mathcal{E}_A$ for affect, $\mathcal{E}_M$ for motor, and $\mathcal{E}_O$ for others). Pre-training combines a region-aware masked reconstruction loss and a brain-state decoupling loss to learn disentangled and neurophysiology-guided representations.

**Existing EFM Pre-training**   Most EEG foundation models (EFMs) rely on masked reconstruction or related self-supervised objectives to learn general-purpose representations. At a high level, an encoder $f_\theta$ maps an input segment $\mathbf{X}$ to a latent representation $\mathbf{Z}$, and the model is trained to reconstruct masked portions of the signal:

$$\mathbf{Z} = f_\theta(\mathbf{X}).$$

These approaches are effective for capturing broad signal characteristics but do not explicitly model how spatial patterns on the scalp may relate to different underlying processes. Spatial information is typically handled implicitly, for example, through channel embeddings or attention mechanisms, making it difficult to represent spatial variations that may reflect different brain states. In addition, standard reconstruction-based objectives treat all variability uniformly and do not encourage the representation to distinguish between broadly shared EEG activity and variations associated with specific processes.

**BrainPro Pre-training**   BrainPro extends this formulation by explicitly incorporating spatial alignment and a factorization of the learned representation. First, a retrieval-based spatial module aligns dataset-specific montages to a universal channel and region template, providing consistent spatial features that help the model capture process-associated spatial patterns. Second, the latent representation is decomposed into a shared component and a state-associated component:

$$\mathbf{Z} = f_{\text{shared}}(\mathbf{X}) \parallel f_{y_{\text{state}}}(\mathbf{X}),$$

where each $f_{y_{\text{state}}}$ corresponds to a process category annotated in the data. This encourages the model to capture *both general EEG characteristics and variations that may relate to different processes*. These components allow BrainPro to learn representations that incorporate explicit spatial cues and process-associated differences while remaining compatible with heterogeneous electrode layouts.

## 3 METHOD

### 3.1 OVERVIEW OF BRAINPRO FRAMEWORK

The overall BrainPro framework is illustrated in Figure 1. It integrates retrieval-based spatial learning (in EEG encoders), brain-state decoupling, and region-aware masked reconstruction into a unified pre-training pipeline (Algorithm 1).

Given an EEG segment, the model first extracts temporal features and retrieves channel- and region-level spatial filters based on a universal template, enabling explicit and montage-robust spatial learn-

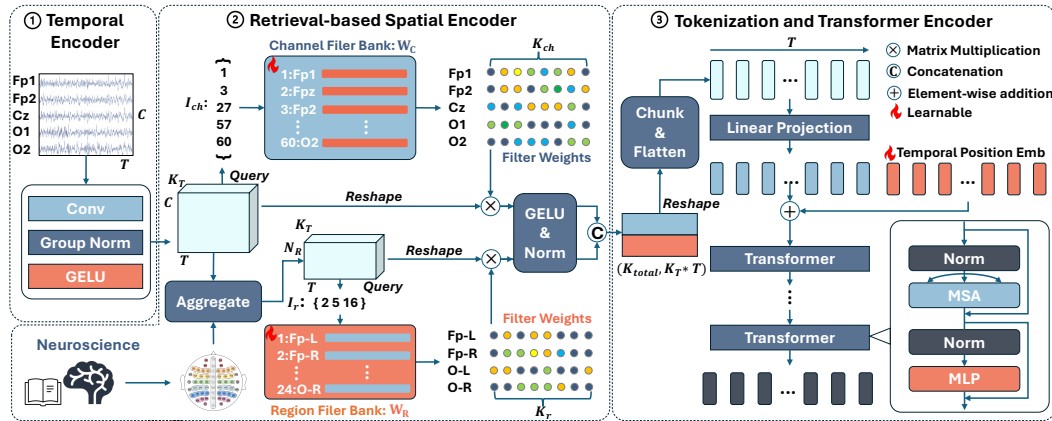

Figure 2: EEG encoder of BrainPro. Each encoder consists of three stages: (1) temporal encoder using temporal CNNs, (2) retrieval-based spatial encoder with channel/region filter banks, (3) patchification, token embedding, and Transformer encoders for spatiotemporal modeling.

ing. These features are then processed by a Transformer encoder to obtain spatiotemporal representations (Section 3.2 and Figure 2).

To further separate shared and state-specific neural representations, BrainPro uses a dual-path design consisting of one shared encoder and multiple state-specific encoders. A margin-based cosine similarity loss is used to decouple these representations. (Section 3.3).

To reinforce this decoupling and better capture brain-state–related spatial structure, reconstruction is performed using both shared and state-specific representations, with region-aware weighting that highlights neurophysiologically relevant areas. This region-aware masked reconstruction injects spatial priors that guide the model toward learning state-associated patterns. The overall training objective therefore combines the decoupling loss with a region-weighted masked reconstruction loss (Section 3.4)

For downstream tasks, any combination of shared and state-specific encoders can be activated through a simple manual prompt mechanism, allowing flexible configuration tailored to the target application. Additional details are provided in Appendix C.

## 3.2 RETRIEVAL-BASED SPATIAL LEARNING

To achieve explicit learning of spatial information under different brain states with various EEG channel montages across datasets, we adopt a retrieval-based spatial learning strategy. We first define a universal template with $C_{\text{pre}}=60$ electrodes (based on the SEED dataset (Zheng & Lu, 2015), excluding reference channels) and $N_{\text{region}}=24$ functional regions following Ding et al. (2024). The detailed universal channel template and brain region definitions are provided in Appendix F. We maintain learnable filter banks aligned to the positions in the universal template for both channel- and region-level retrieval:

$$\mathbf{W}_{\text{C}} \in \mathbb{R}^{C_{\text{pre}} \times K_{\text{C}}}, \quad \mathbf{W}_{\text{R}} \in \mathbb{R}^{N_{\text{region}} \times K_{\text{R}}}, \tag{1}$$

where $K_{\text{C}}$ and $K_{\text{R}}$ denote the number of channel-wise and region-wise spatial filters, respectively. Given an input EEG, $X$, we first use a temporal encoder, $\mathcal{F}_{temp}$ to learn temporal dynamics encoded in EEG signals, denoted by $\mathbf{H}_{\text{temp}}$. More details can be find in Appendix A. We further reshape the $\mathbf{H}_{\text{temp}}$ into $\mathbf{H}_{\text{temp}}^{reshp} \in \mathbb{R}^{C \times K_T * T}$ for easy spatial learning.

**Fine-grained channel features.** Let $I_{\text{ch}} \subseteq \{1, \ldots, C_{\text{pre}}\}$ be the set of indices in the universal template that are present in the current sample. This set is obtained by channel-name matching or, if unavailable, by nearest-neighbor mapping in standardized 3D head coordinates. We then retrieve the corresponding filters from the channel filter bank $\mathbf{W}_{\text{C}}[I_{\text{ch}}]$ and apply them to the temporal features:

$$\mathbf{H}_{\text{C}} = \sigma\left(\mathbf{W}_{\text{C}}[I_{\text{ch}}]^{\top} \mathbf{H}_{\text{temp}}^{reshp}\right) \in \mathbb{R}^{K_{\text{C}} \times K_T * T}, \tag{2}$$

where $\sigma(\cdot)$ denotes normalization followed by activation (we use GELU+GroupNorm).

**Coarse region features.**   We aggregate temporal features within each functional region present in the current sample and apply the region filter bank. Each channel has a corresponding brain region label from $I_r = \{r_1, \ldots, r_C\}$ that specifies which functional region in the universal template it belongs to. Let $\mathcal{R}_{\text{uniq}}$ denote the set of unique functional regions covered in the sample, and $\mathcal{C}_j \subseteq I_r$ the subset of channels belonging to region $j$. For each region index $j \in \{1, \ldots, |\mathcal{R}_{\text{uniq}}|\}$, we first average the temporal features of its constituent channels:

$$\mathbf{M}_{\text{region}}[j] = \frac{1}{|\mathcal{C}_j|} \sum_{c \in \mathcal{C}_j} \mathbf{H}_{\text{temp}}^{reshp}[c, :] \in \mathbb{R}^{1 \times K_T * T}. \tag{3}$$

Then we retrieve the corresponding region spatial filters using $\mathcal{R}_{\text{uniq}}$. The region-wise representations are then processed by the region filter bank:

$$\mathbf{H}_{\text{R}} = \sigma\big(\mathbf{W}_{\text{R}}[\mathcal{R}_{\text{uniq}}]^\top \mathbf{M}_{\text{region}}\big) \in \mathbb{R}^{K_{\text{R}} \times K_T * T}. \tag{4}$$

Finally, the fine- and coarse-level features are concatenated and reshaped to form the spatial representation:

$$\mathbf{H}_{\text{spatial}} = \text{Reshape}(\text{Concat}(\mathbf{H}_{\text{C}}, \mathbf{H}_{\text{R}})) \in \mathbb{R}^{K_T * K_{\text{total}} \times T}, \quad K_{\text{total}} = K_{\text{C}} + K_{\text{R}}. \tag{5}$$

This $\mathbf{H}_{\text{spatial}}$ is further processed by a Transformer encoder to obtain spatiotemporal representations for brain-state decoupling and brain-region–aware reconstruction. Additional details on the tokenization procedure and the Transformer encoder are provided in Appendix B.

## 3.3   BRAIN-STATE DECOUPLING

A key innovation of BrainPro is to disentangle shared and brain-state-specific representations with the help of multiple encoders and a decoupling loss. As shown in Figure 1, the model contains one shared encoder $\mathcal{E}_{\text{S}}$ and $K$ brain-state-specific encoders $\{\mathcal{E}_k\}_{k=1}^K$ (e.g., affect, motor, etc.). In this work we consider $K{=}3$ states because affective and motor processes are among the few EEG states with well-established and reliably distinguishable neurophysiological signatures (e.g., frontal–limbic activation for affect; sensorimotor rhythms for motor), and these labels are consistently available across large-scale public datasets. Grouping all remaining datasets into "other" avoids forcing noisy or incompatible labels into the decoupling process. This choice reflects a practical and neuroscientifically grounded grouping, and the framework can naturally extend to additional states when richer annotations become available. Given an EEG segment $\mathbf{X}$, the encoders produce

$$\mathbf{Z}_{\text{shared}} = \mathcal{E}_{\text{S}}(\mathbf{X}), \qquad \mathbf{Z}_k = \mathcal{E}_k(\mathbf{X}), \quad k = 1, \ldots, K. \tag{6}$$

**Selective gradient updates.**   Each EEG sample is associated with a brain-state label $y_{\text{state}}$. To ensure decoupling, we update only the shared encoder $\mathcal{E}_{\text{S}}$ and the corresponding state-specific encoder $\mathcal{E}_{y_{\text{state}}}$. Outputs from other encoders are detached:

$$\mathbf{Z}_j = \text{stopgrad}(\mathcal{E}_j(\mathbf{X})), \qquad j \neq y_{\text{state}}. \tag{7}$$

This allows us to use inactive encoders for decoupling loss computation while preventing unintended parameter updates.

**Decoupling loss.**   To enforce representation separation, we define a decoupling loss $\mathcal{L}_{\text{dec}}$ consisting of two margin-based cosine similarity losses, denoted as $\mathcal{L}_{\text{sim}}$. The similarity loss is

$$\mathcal{L}_{\text{sim}}(\mathbf{a}, \mathbf{b}) = \max\big(\text{sim}_{\cos}(\mathbf{a}, \mathbf{b}) - m, 0\big), \tag{8}$$

where $m{=}0.1$ is a margin. The first similarity term penalizes similarity between shared and active brain-state representations, and the second penalizes similarity between the active state encoder and all inactive encoders:

$$\mathcal{L}_{\text{dec}} = \underbrace{\mathcal{L}_{\text{sim}}(\mathbf{Z}_{\text{shared}}, \mathbf{Z}_{y_{\text{state}}})}_{\text{shared-active}} + \underbrace{\sum_{k \neq y_{\text{state}}} \mathcal{L}_{\text{sim}}(\mathbf{Z}_{y_{\text{state}}}, \mathbf{Z}_k)}_{\text{active-inactive}}. \tag{9}$$

This encourages the encoders to learn disentangled and state-specific representations.

Table 1: Overview of downstream BCI tasks and datasets.

| BCI Task | Dataset | Sampling Rate | # Ch. | Duration | # Samples | Label |
|---|---|---|---|---|---|---|
| I.Emotion Recognition | FACED | 250 Hz | 32 | 10 s | 10,332 | 9-class |
| | SEED-V | 1000 Hz | 62 | 1 s | 117,744 | 5-class |
| | SEED-VII | 1000 Hz | 62 | 1 s | 281,679 | 7-class |
| II.Motor Imagery | BCI-IV-2A | 250 Hz | 22 | 4 s | 5,088 | 4-class |
| | SHU-MI | 250 Hz | 32 | 4 s | 11,988 | 2-class |
| III.Imagined Speech | BCIC2020-3 | 256 Hz | 64 | 4 s | 5,250 | 5-class |
| IV.Mental Disorder Diagnosis | Mumtaz2016 | 256 Hz | 19 | 4 s | 3,525 | 2-class |
| V.Mental Stress Detection | MentalArithmetic | 500 Hz | 20 | 4 s | 1,080 | 2-class |
| VI.Mental Attention Detection | ATTEN | 500 Hz | 28 | 4 s | 4,680 | 2-class |

**Note:** All datasets are preprocessed into fixed-length EEG segments following their official protocols.

### 3.4 MASKING AND REGION-AWARE RECONSTRUCTION

Another key goal of BrainPro is to use a standard masked-reconstruction objective while augmenting it with spatial priors so that the model is encouraged to learn representations that better capture variations associated with different brain states. To this end, we mask the input patches with a ratio of $\rho=0.5$ (Jiang et al., 2024) and pass the masked signal $\bar{\mathbf{X}}$ through both the shared encoder and the active state-specific encoder. Their outputs are concatenated and fed into a decoder:

$$\hat{\mathbf{X}} = \mathcal{D}_{y_{state}}\big(\mathcal{E}_{\mathrm{S}}(\bar{\mathbf{X}}) \,\|\, \mathcal{E}_{y_{state}}(\bar{\mathbf{X}})\big), \tag{10}$$

where masked positions in $\bar{\mathbf{X}}$ are replaced with learned [MASK] embeddings.

**Channel importance weights.** To emphasize neurophysiology-relevant regions, we assign each brain state $y_{\mathrm{state}}$ a channel-importance vector $w^{(y_{\mathrm{state}})} \in [0,1]^{C_{\mathrm{pre}}}$. Based on well-established neuroscience priors, frontal, temporal, and central channels are set to 1 for the affect state, while central and parietal channels are set as important for the motor state. For the *other* state and the shared encoder, we assign all channels equal importance because no concrete prior exists. We apply a smooth weighting function:

$$\mathrm{weights}(w) = 0.5 + \sigma\big(T \cdot (w - 0.5)\big), \tag{11}$$

with sharpness parameter $T$=epoch to gradually increase weighting strength.

**Region-aware masked reconstruction loss.** Let $M$ denote masked channel-time positions. The weighted MSE reconstruction loss is

$$\mathcal{L}_{\mathrm{rec}} = \frac{1}{|M|} \sum_{(i,t)\in M} \mathrm{weights}\big(w_i^{(y_{\mathrm{state}})}\big) \cdot \big(X_{i,t} - \hat{X}_{i,t}\big)^2. \tag{12}$$

This prioritizes reconstruction accuracy for brain-state–relevant regions.

**Final pre-training loss.** The overall pre-training objective combines reconstruction and decoupling:

$$\mathcal{L}_{\mathrm{BrainPro}} = \mathcal{L}_{\mathrm{rec}} + \mathcal{L}_{\mathrm{dec}}. \tag{13}$$

## 4 EXPERIMENT

### 4.1 DATASETS AND PRE-PROCESSING

**Pre-training.** We construct a large-scale pre-training corpus by combining diverse EEG datasets spanning affective, motor, and clinical domains. To ensure that the data capture meaningful brain states, we follow a selection protocol similar to LaBraM (Jiang et al., 2024), while excluding certain datasets and incorporating additional motor imagery datasets. In total, the pre-training collection covers approximately **2,400 hours** of EEG recordings from a wide range of paradigms. The datasets are grouped into three categories for process-specific representation learning: (1) *affect*, (2) *motor*, and (3) *others*. A complete list of datasets and their categorization is provided in Appendix G.1.

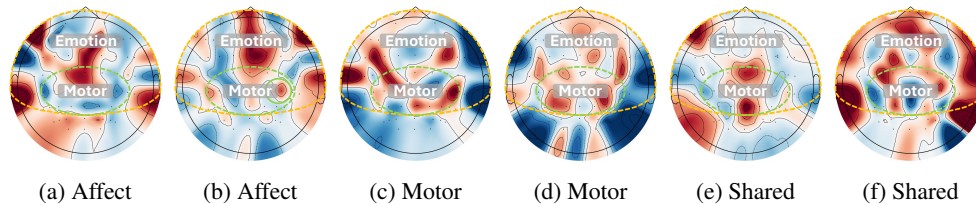

| (a) Affect | (b) Affect | (c) Motor | (d) Motor | (e) Shared | (f) Shared |

Figure 3: Visualization of the learned spatial filters for the Affect, Motor, and Shared encoders after pre-training. Each subplot shows a 2D scalp topography in which warmer colors indicate larger filter weights and cooler colors indicate smaller weights, reflecting the relative contribution of each scalp region to the encoder's representation. Emotion- and motor-related regions are outlined with dashed lines, shown in orange and green respectively.

All EEG recordings are segmented into 10-second clips and uniformly resampled to 200 Hz. Following Jiang et al. (2024), signals are scaled to 0.1 mV, resulting in values between $-1$ and $1$. To improve stability, noisy segments with absolute values greater than 10 are removed, yielding approximately **2,180 hours** of clean data. Further implementation details are provided in Appendix G.2.

**Downstream Evaluation.** To comprehensively evaluate our method, we consider six representative downstream BCI tasks using nine publicly available datasets: emotion recognition, motor imagery, imagined speech, mental disorder diagnosis, mental stress detection, and attention detection. The selected tasks and their corresponding datasets are summarized in Table 1, with additional dataset-specific details in Appendix H.2. For consistency with pre-training, all EEG signals are resampled to 200 Hz. Further preprocessing details are provided in Appendix H.2.

## 4.2 BASELINES AND EVALUATION METRICS

Following prior studies Jiang et al. (2024); Wang et al. (2025), we compare BrainPro against both non-foundation and foundation model baselines. Among non-foundation approaches, we include **EEGNet** (Lawhern et al., 2018) and **Conformer** (Song et al., 2023). For foundation-model baselines, we consider: **BIOT** (Yang et al., 2023), **LaBraM** (Jiang et al., 2024), **EEGPT** (Wang et al., 2024), and **CBraMod** (Wang et al., 2025). Further baseline details about the baseline are in Appendix H.3.

For evaluation metrics, we adopt the ones suitable for binary and multi-class tasks. For binary classification, we report **Balanced Accuracy (ACC-B)**, **AUC-PR**, and **AUROC**. For multi-class classification, following prior studies (Jiang et al., 2024; Wang et al., 2025), we report **Balanced Accuracy (ACC-B)**, **Cohen's Kappa**, and **Weighted F1 (F1-W)**.

## 4.3 IMPLEMENTATION DETAILS

**Pre-training.** Pre-training is conducted end-to-end with a shared encoder and process-specific encoders/decoders. We train for 30 epochs using the AdamW optimizer with a cosine learning rate schedule, warmup, and gradient clipping. The model architecture consists of temporal and spatial encoders, patch makers, and transformer-based encoder-decoder modules. Hyperparameters are chosen to balance efficiency and representational capacity, with the full configuration provided in Appendix G.3 (Table 5).

**Downstream Evaluation.** Each dataset is split into training, validation, and test subsets without overlap. Models are trained on the training set, with validation used for model selection and hyperparameter tuning. For each dataset, we manually activate the shared encoder and the corresponding state-specific encoder(s) based on its labeled brain-state category. Only these activated encoders are fine-tuned. We concatenate the output of these encoders and fed into a MLP classifier (Wang et al., 2025). Final performance is reported on the test set after a single evaluation. To reduce randomness, we fix seeds $\{0, 1, 2, 3, 4\}$ and report the mean and standard deviation across five runs. All experiments are conducted on a cluster with 5 NVIDIA A800 GPUs (80 GB each). Detailed hyperparameter configurations are provided in Appendix H.1. After loading pre-trained weights, we

Table 2: Comparison results of different methods on downstream tasks.

| Methods | FACED (9-Class) | | | SEED-V (5-Class) | | |
|---------|--------|--------|--------|--------|--------|--------|
| | ACC-B | Kappa | F1-W | ACC-B | Kappa | F1-W |
| EEGNet | 0.3692 ± 0.0103 | 0.2880 ± 0.0113 | 0.3693 ± 0.0111 | 0.2408 ± 0.0031 | 0.0536 ± 0.0042 | 0.1908 ± 0.0088 |
| Conformer | 0.4566 ± 0.0108 | 0.3849 ± 0.0121 | 0.4555 ± 0.0098 | 0.3087 ± 0.0095 | 0.1294 ± 0.0134 | 0.2837 ± 0.0229 |
| BIOT | 0.2992 ± 0.0119 | 0.2105 ± 0.0147 | 0.2954 ± 0.0149 | 0.3575 ± 0.0052 | 0.1971 ± 0.0081 | 0.3636 ± 0.0078 |
| EEGPT | 0.2809 ± 0.0116 | 0.1912 ± 0.0129 | 0.2810 ± 0.0120 | 0.2421 ± 0.0048 | 0.0557 ± 0.0077 | 0.2438 ± 0.0055 |
| LaBraM | 0.5224 ± 0.0116 | 0.4610 ± 0.0126 | 0.5259 ± 0.0108 | 0.3986 ± 0.0200 | 0.2491 ± 0.0256 | 0.4040 ± 0.0197 |
| CBraMod | 0.5669 ± 0.0094 | 0.5112 ± 0.0110 | 0.5729 ± 0.0105 | 0.3960 ± 0.0033 | 0.2521 ± 0.0048 | 0.4050 ± 0.0052 |
| BrainPro | **0.5937** ± 0.0087 | **0.5418** ± 0.0092 | **0.6023** ± 0.0061 | **0.4078** ± 0.0075 | **0.2612** ± 0.0089 | **0.4115** ± 0.0071 |

| Methods | BCI-IV-2a (4-Class) | | | SHU (2-Class) | | |
|---------|--------|--------|--------|--------|--------|--------|
| | ACC-B | Kappa | F1-W | ACC-B | AUC-PR | AUROC |
| EEGNet | 0.5521 ± 0.0183 | 0.4028 ± 0.0244 | 0.5346 ± 0.0228 | 0.5664 ± 0.0522 | 0.6609 ± 0.0099 | 0.6791 ± 0.0085 |
| Conformer | 0.4879 ± 0.0183 | 0.3171 ± 0.0244 | 0.4561 ± 0.0206 | 0.6167 ± 0.0172 | 0.6763 ± 0.0054 | 0.6927 ± 0.0072 |
| BIOT | 0.2392 ± 0.0280 | -0.0144 ± 0.0373 | 0.1063 ± 0.0060 | 0.4981 ± 0.0044 | 0.5077 ± 0.0025 | 0.5069 ± 0.0060 |
| EEGPT | 0.3849 ± 0.0226 | 0.1799 ± 0.0301 | 0.3272 ± 0.0336 | 0.5481 ± 0.0139 | 0.5631 ± 0.0165 | 0.5673 ± 0.0197 |
| LaBraM | 0.5255 ± 0.0329 | 0.3674 ± 0.0439 | 0.5095 ± 0.0405 | 0.6175 ± 0.0172 | 0.6821 ± 0.0214 | 0.6720 ± 0.0253 |
| CBraMod | 0.5148 ± 0.0402 | 0.3530 ± 0.0536 | 0.5032 ± 0.0512 | 0.6108 ± 0.0233 | 0.6632 ± 0.0277 | 0.6668 ± 0.0338 |
| BrainPro | **0.5674** ± 0.0148 | **0.4232** ± 0.0198 | **0.5653** ± 0.0169 | **0.6319** ± 0.0107 | **0.7102** ± 0.0076 | **0.7105** ± 0.0076 |

| Methods | Mental Arithmetic (2-Class) | | | Attention (2-Class) | | |
|---------|--------|--------|--------|--------|--------|--------|
| | ACC-B | AUC-PR | AUROC | ACC-B | AUC-PR | AUROC |
| EEGNet | 0.5533 ± 0.0280 | 0.5702 ± 0.0539 | 0.5717 ± 0.0456 | 0.6004 ± 0.0123 | 0.6294 ± 0.0288 | 0.6647 ± 0.0180 |
| Conformer | 0.6867 ± 0.0492 | 0.7691 ± 0.0286 | 0.7039 ± 0.0177 | 0.7198 ± 0.0158 | 0.7819 ± 0.0218 | 0.7992 ± 0.0283 |
| BIOT | 0.5583 ± 0.0306 | 0.5512 ± 0.0516 | 0.5662 ± 0.0250 | 0.6111 ± 0.0411 | 0.7273 ± 0.0132 | 0.7367 ± 0.0162 |
| EEGPT | 0.5650 ± 0.0341 | 0.5860 ± 0.0805 | 0.5937 ± 0.0762 | 0.6674 ± 0.0560 | 0.8015 ± 0.0372 | 0.8103 ± 0.0303 |
| LaBraM | 0.6688 ± 0.0279 | 0.7504 ± 0.0649 | 0.7168 ± 0.0452 | 0.6785 ± 0.0223 | 0.7838 ± 0.0307 | 0.7994 ± 0.0198 |
| CBraMod | 0.7354 ± 0.0410 | 0.8237 ± 0.0225 | 0.7654 ± 0.0203 | 0.6478 ± 0.0258 | 0.7417 ± 0.0175 | 0.7468 ± 0.0198 |
| BrainPro | **0.8083** ± 0.0156 | **0.8980** ± 0.0052 | **0.8512** ± 0.0083 | **0.7222** ± 0.0291 | **0.7975** ± 0.0392 | **0.8064** ± 0.0259 |

Note: **Bold** indicates the best performance. Cyan highlight marks our BrainPro.

re-initialize the temporal position embeddings with Xavier uniform initialization, similar to practices in NLP and vision, where task- or resolution-specific embeddings are re-initialized to improve adaptation. The corresponding ablation is reported in Appendix N.

## 4.4 Interpretation of Learned Spatial Filters

**BrainPro learns meaningful and interpretable shared and state-specialized representations.**
Figure 3 visualizes the spatial filters learned by the Affect, Motor, and Shared encoders after pre-training. Clear and distinct spatial patterns emerge across the three encoders, demonstrating that each learns neurophysiologically meaningful representations. The Affect encoder emphasizes frontal and temporal regions, consistent with affective EEG literature (Alarcão & Fonseca, 2019; Gao et al., 2021), while the Motor encoder highlights central sensorimotor areas (Pfurtscheller et al., 2006) (marked on the figure). In contrast, the Shared encoder produces distributed, non-localized patterns that reflect global EEG structure. Importantly, these explicit and easily visualizable spatial filters provide improved interpretability, allowing direct inspection of what spatial patterns each encoder has learned. More details are in Appendix Q.

## 4.5 Primary Results

**BrainPro improves accuracy and generalization cross diverse BCI tasks.** Table 2 compares methods across six EEG benchmarks. EEGNet and Conformer perform reasonably on simpler datasets but degrade on complex multi-class tasks, particularly in Kappa and F1. BIOT and EEGPT exhibit weaker transferability with unstable results, while LaBraM and CBraMod improve performance by leveraging larger-scale pre-training. BrainPro consistently outperforms all baselines, delivering the highest balanced accuracy and robust gains across metrics. For example, Brain-Pro achieves 0.5937/0.5418/0.6023 (ACC-B/Kappa/F1-W) on FACED and 0.8083/0.8980/0.8512

Table 3: Ablation studies of the main components on downstream tasks.

| Methods | BCI-IV-2A (4-Class) | | | Mental Arithmetic (2-Class) | | |
|---|---|---|---|---|---|---|
| | ACC-B | Kappa | F1-W | ACC-B | AUC-PR | AUROC |
| w/o masking | 0.4314 ± 0.1160 | 0.2419 ± 0.1547 | 0.4084 ± 0.1532 | 0.7483 ± 0.0785 | 0.8930 ± 0.0131 | 0.8556 ± 0.0134 |
| w/o reconstruction | 0.4354 ± 0.0329 | 0.2472 ± 0.0439 | 0.3758 ± 0.0548 | 0.7083 ± 0.0903 | **0.9147** ± 0.0209 | **0.8976** ± 0.0273 |
| w/o decoupling | 0.4870 ± 0.0184 | 0.3160 ± 0.0246 | 0.4725 ± 0.0219 | 0.7317 ± 0.0696 | 0.9102 ± 0.0356 | 0.8860 ± 0.0408 |
| w random retrieval | 0.5125 ± 0.0308 | 0.3500 ± 0.0410 | 0.5052 ± 0.0379 | 0.7733 ± 0.0410 | 0.8879 ± 0.0288 | 0.8606 ± 0.0343 |
| w single encoder | 0.5203 ± 0.0131 | 0.3603 ± 0.0174 | 0.5108 ± 0.0164 | 0.7667 ± 0.0221 | 0.8602 ± 0.0304 | 0.8056 ± 0.0342 |
| w/o pre-training | 0.4479 ± 0.1066 | 0.2639 ± 0.1421 | 0.4224 ± 0.1454 | 0.7117 ± 0.0889 | 0.8926 ± 0.0083 | 0.8647 ± 0.0109 |
| BrainPro | **0.5674** ± 0.0148 | **0.4232** ± 0.0198 | **0.5653** ± 0.0169 | **0.8083** ± 0.0156 | 0.8980 ± 0.0052 | 0.8512 ± 0.0083 |

Note: **Bold** indicates the best performance. Cyan highlight marks our BrainPro.

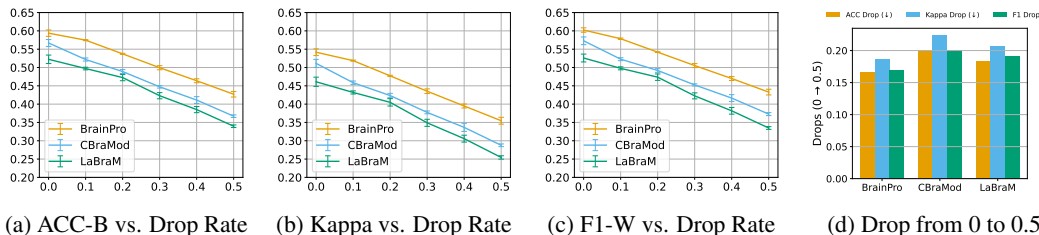

(a) ACC-B vs. Drop Rate  (b) Kappa vs. Drop Rate  (c) F1-W vs. Drop Rate  (d) Drop from 0 to 0.5

Figure 4: Performance of BrainPro, CBraMod, and LaBraM under different levels of channel-drop rate.

(ACC-B/AUC-PR/AUROC) on Mental Arithmetic. More results are in Appendix I. These results confirm that BrainPro not only improves accuracy but also generalizable representations, validating its effectiveness as a large model for EEG decoding.

## 4.6 ABLATION STUDIES

Table 3 summarizes the ablation results on BCI-IV-2A and Mental Arithmetic. (1) **Each pre-training component influences performance:** removing masking, reconstruction, or decoupling lowers accuracy on at least one dataset, showing that these modules contribute measurable performance gains. (2) **Reconstruction supports performance in more complex settings:** while removing reconstruction slightly improves AUC-related metrics on Mental Arithmetic, it noticeably reduces performance on BCI-IV-2A, indicating that reconstruction contributes useful information for tasks with richer spatial-temporal structure. (3) **Retrieval-based spatial learning affects decoding performance:** replacing structured retrieval with random retrieval leads to clear accuracy drops, indicating that the retrieval mechanism plays a functional role in producing effective features. (4) **State decoupling impacts representation quality as reflected in downstream accuracy:** removing the decoupling strategy reduces performance across datasets, suggesting that the decoupling term helps the model obtain more useful features for the evaluated tasks. (5) **Using multiple encoders improves downstream accuracy compared to a single encoder conditioned on state types:** using multiple encoders yields consistently higher performance, indicating that parallel encoders offer additional useful capacity beyond a single encoder with input state-conditioned. (6) **Pre-training contributes substantially to performance:** removing the pre-training stage results in significant accuracy degradation, confirming that the pre-trained initialization plays an important role in downstream results.

## 4.7 CHANNEL-DROP ANALYSIS

**BrainPro exhibits smaller performance degradation under missing-channel conditions.** Figure 4 evaluates decoding performance on FACED under progressively increasing channel-drop rates during inference, where a random subset of channels is removed at each drop level. Detailed results are in Table 12. This provides a targeted assessment of how models behave when electrode information is incomplete or heterogeneous. Across all metrics, ACC-B, Kappa, and F1-W decrease as more channels are removed, and the differences between models become more pronounced at higher

Table 4: Effects of different encoder configurations on downstream performance.

| Method | FACED (9-Class) | | | | | BCI-IV-2A (4-Class) | | | | |
|---|---|---|---|---|---|---|---|---|---|---|
| | ACC-B | Kappa | F1-W | #Params | FLOPs | ACC-B | Kappa | F1-W | #Params | FLOPs |
| S | 0.5635 ± 0.0031 | 0.5088 ± 0.0050 | 0.5710 ± 0.0049 | 14.3M | 782M | 0.5527 ± 0.0218 | 0.4036 ± 0.0290 | 0.5504 ± 0.0236 | 3.87M | 266M |
| S + A | 0.5937 ± 0.0087 | 0.5418 ± 0.0092 | 0.6023 ± 0.0061 | 28.2M | 1.56G | 0.5499 ± 0.0044 | 0.3999 ± 0.0059 | 0.5469 ± 0.0060 | 7.69M | 0.53G |
| S + M | 0.5828 ± 0.0142 | 0.5296 ± 0.0143 | 0.5904 ± 0.0144 | – | – | 0.5674 ± 0.0148 | 0.4232 ± 0.0198 | 0.5653 ± 0.0169 | – | – |
| S + O | 0.5821 ± 0.0144 | 0.5292 ± 0.0150 | 0.5881 ± 0.0133 | – | – | 0.5525 ± 0.0173 | 0.4033 ± 0.0231 | 0.5491 ± 0.0192 | – | – |
| S + A + M | 0.5988 ± 0.0057 | 0.5477 ± 0.0058 | 0.6098 ± 0.0043 | 42.2M | 2.35G | 0.5797 ± 0.0229 | 0.4396 ± 0.0305 | 0.5764 ± 0.0245 | 11.5M | 0.80G |
| S + A + M + O | 0.5824 ± 0.0050 | 0.5272 ± 0.0054 | 0.5841 ± 0.0056 | 56.2M | 3.13G | 0.5660 ± 0.0067 | 0.4213 ± 0.0089 | 0.5631 ± 0.0080 | 15.3M | 1.06G |

Note: "–" indicates the model size and FLOPs are unchanged. Cyan highlight marks configurations used in BrainPro.

drop ratios. BrainPro consistently maintains higher performance than LaBraM and CBraMod across the entire drop spectrum, and its total ACC-B drop at 50% removal is the smallest among all models shown in Figure 4 (d). These results provide empirical evidence that the structured retrieval mechanism contributes to more stable downstream performance when electrodes are missing, supporting the intended behavior of BrainPro under montage variability.

### 4.8 ENCODER-COMBINATION ANALYSIS

**Using multiple state-specific encoders improves downstream performance compared to using the shared encoder alone.** Table 4 reports decoding performance on FACED and BCI-IV-2A under different encoder combinations. Configurations that include a state-specific encoder (S+A or S+M) outperform the shared-only encoder (S) on their corresponding tasks, indicating that the specialized encoders provide additional useful capacity for the evaluated datasets. Using multiple state-specific encoders (S+A+M) further improves performance on both datasets, suggesting that the features learned by different encoders capture complementary aspects of the EEG signals. Adding the "other" encoder (S+A+M+O) does not always yield further gains, showing that *increasing the number of encoders does not guarantee improvement*. Overall, the results demonstrate that parallel encoders contribute measurable benefits over single-encoder configurations. An extended analysis also confirms the performance gain is due to the structure design of BrainPro instead of changes in MLP parameters ( L).

**State-specific encoders learn task-aligned representations.** The relative performance of S+A, S+M, and S+O configurations provides empirical evidence that each state-specific encoder captures information most useful for the tasks associated with its state type. On FACED, which involves affective decoding, S+A outperforms S+M and S+O; similarly, on BCI-IV-2A, which involves motor imagery, S+M outperforms S+A and S+O. These trends indicate that the affect-, motor-, and other-specific encoders each learn state-related representations that align with the characteristics of the downstream tasks. This performance pattern supports the effectiveness of the decoupling strategy in encouraging encoders to specialize according to the corresponding brain-state categories.

## 5 CONCLUSION

In this work, we introduced **BrainPro**, a large EEG model that integrates retrieval-based spatial learning, brain-state decoupling, and region-aware masked reconstruction to capture brain state-aware representations. Extensive experiments across nine public EEG/BCI datasets demonstrate that BrainPro achieves better and consistent performance over a wide range of tasks. Analyses of learned spatial filters reveal clear and meaningful topographies for different encoders, and the channel-drop study shows that these spatial representations remain effective under incomplete electrode configurations. The encoder-combination experiments further confirm that the state-specific encoders capture complementary and task-aligned information, while the model-size analysis shows that the architectural design drives the observed improvements rather than capacity alone. Comprehensive ablations across masking, reconstruction, decoupling, retrieval, and pre-training highlight that each component contributes measurable benefits, and that their integration yields the most effective representations. Together, these results show that combining explicit spatial modeling with brain-state–aware representation learning provides a powerful strategy for building large and versatile EEG models.

## REPRODUCIBILITY STATEMENT

We have taken multiple steps to ensure the reproducibility of our work. The architecture and training procedures of BrainPro are described in Section 3, with additional implementation details and hyperparameters provided in Appendix G and H. All datasets used in our experiments are publicly available, and we describe the preprocessing steps in Appendix H.2. An anonymous link to the source code is provided: https://anonymous.4open.science/r/BrainPro-29A4/.

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

## A    More details about the temporal encoder in BrainPro

Following Jiang et al. (2024), a temporal CNN, $\mathcal{F}_{temp}$, extracts temporal dynamics along the time axis from the input EEG, $X \in \mathbb{R}^{C \times T}$, independently for each channel. Each convolutional block consists of a 1D convolution followed by Group Normalization (GroupNorm), which normalizes activations within groups of channels to stabilize training with small batch sizes, and the Gaussian Error Linear Unit (GELU), a smooth nonlinear activation that weights inputs by their Gaussian cumulative distribution for improved expressivity. It can be formulated as

$$\mathbf{Z}^{(\ell)} = \mathcal{F}_{temp}(\mathbf{Z}^{(\ell-1)}) = \mathrm{GELU}\Big(\mathrm{GroupNorm}\Big(\mathrm{Conv1D}^{(\ell)}(\mathbf{Z}^{(\ell-1)})\Big)\Big), \quad \ell = 1, \dots, L_T. \quad (14)$$

where $\mathbf{Z}^{(0)} = \mathbf{X}$ and each $\mathrm{Conv1D}^{(\ell)}$ applies $K_T$ filters (kernel size $k_\ell$, stride $s_\ell$) along time. Padding is applied to keep the output size the same. The output $\mathbf{H}_{\mathrm{temp}} = \mathbf{Z}^{(L_T)} \in \mathbb{R}^{K_T \times C \times T}$ preserves temporal resolution ($T$) while expanding the per-channel feature depth to $K_T$.

## B    More Details on Tokenization and the Transformer Used in This Paper

Given the spatial representation from the retrieval-based spatial encoder, $\mathbf{H}_{\mathrm{spatial}}$, we divide it into $N_p$ (possibly strided with the step being $s_P$) temporal patches of length $P$:

$$\mathbf{Z}_{\mathrm{patch}} \in \mathbb{R}^{N_p \times K_T \times K_{\mathrm{total}} \times P}, \quad N_p = \left\lfloor \frac{T - P}{s_P} \right\rfloor + 1. \quad (15)$$

Each patch is flattened and projected into a $d$-dimensional token using a learnable embedding matrix $\mathbf{E} \in \mathbb{R}^{d \times (K_T K_{\mathrm{total}} P)}$, followed by adding a positional embedding $\mathbf{p}_i \in \mathbb{R}^d$ onto the flattened and linearly projected tokens:

$$\mathbf{z}_i = \mathbf{E}\,\mathrm{Flatten}(\mathbf{Z}_{\mathrm{patch}}[i]) + \mathbf{p}_i, \quad i = 1, \dots, N_p. \quad (16)$$

This yields a sequence of tokens $\mathbf{Z} = \{\mathbf{z}_i\}_{i=1}^{N_p} \in \mathbb{R}^{N_p \times d}$, which serves as the input to the Transformer layers for modeling temporal contextual information.

To capture long-range spatiotemporal dependencies, each encoder applies $L$ Transformer blocks. Each block follows a pre-norm architecture consisting of multi-head self-attention (MSA) and a position-wise multilayer perceptron (MLP):

$$\mathbf{Z}'^{(\ell)} = \mathbf{Z}^{(\ell)} + \mathrm{MSA}\Big(\mathrm{LN}(\mathbf{Z}^{(\ell)})\Big), \quad (17)$$

$$\mathbf{Z}^{(\ell+1)} = \mathbf{Z}'^{(\ell)} + \mathrm{MLP}\Big(\mathrm{LN}(\mathbf{Z}'^{(\ell)})\Big), \quad \ell = 0, \dots, L-1, \quad (18)$$

where $\mathbf{Z}^{(0)} = \mathbf{Z}$ denotes the input token embeddings and LN is layer normalization.

The MSA module models pairwise interactions among all patches by projecting $\mathbf{Z}^{(\ell)}$ into queries ($Q$), keys ($K$), and values ($V$), and computing

$$\mathrm{Attention}(Q, K, V) = \mathrm{softmax}\left(\frac{QK^\top}{\sqrt{d_k}}\right) V, \quad (19)$$

where $d_k$ is the key dimension. Multiple attention heads operate in parallel and are concatenated to form richer contextual representations.

The MLP module consists of two fully connected layers with hidden dimension $d_{\mathrm{ff}}$ and a nonlinear GELU activation, applied independently to each token. Residual connections around both the MSA and MLP components stabilize training.

After $L$ blocks, we obtain the final token sequence $\mathbf{Z}^{(L)} \in \mathbb{R}^{N_p \times d}$.

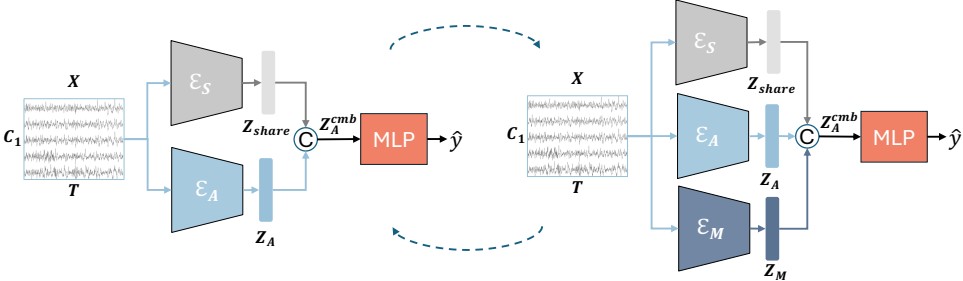

Figure 5: Flexible configuration of BrainPro for downstream tasks. Any subset of encoders can be selected and concatenated for task-specific adaptation. Emotion recognition is used for demonstration purpose.

## C  FLEXIBLE DOWNSTREAM ADAPTATION

For a downstream task, any subset $\mathcal{S}$ of encoders can be activated and concatenated:

$$\mathbf{Z}_\star = \text{Concat}\big(\mathbf{Z}_{\text{shared}}, \{\mathbf{Z}_k\}_{k\in\mathcal{S}}\big), \quad \hat{y} = f_{\text{cls}}(\mathbf{Z}_\star). \tag{20}$$

Figure 5 illustrates flexible encoder selection. The experiments in Appendix C shows the effectiveness of such flexible configuration design. For downstream classification tasks, we construct these tokens into a segment-level vector $\bar{\mathbf{Z}} \in \mathbb{R}^d$ using either mean pooling, moving average (averaging every 5 tokens), or use all of them. A detailed comparison between different token merge mode can be find in Appendix O. We adopt the similar MLP head as Wang et al. (2025) with a hidden factor to adapt the hidden size in MLP.

---

**Algorithm 1** BrainPro Pre-training Pipeline
---

1: **Input:** EEG segment $\mathbf{X} \in \mathbb{R}^{C\times T}$, montage mapping, brain-state label $y_{\text{state}}$
2: **Params:** Shared encoder $\mathcal{E}_{\text{S}}$, brain-state encoders $\{\mathcal{E}_k\}_{k=1}^K$, brain-state decoders $\{\mathcal{D}_k\}_{k=1}^K$, filter banks $\mathbf{W}_{\text{C}}, \mathbf{W}_{\text{R}}$, channel priors $w^{(y_{\text{state}})}$, mask ratio $\rho$

3: **Step 1: Mask input**
4: Randomly mask $\rho$ fraction of patches in $\mathbf{X}$ to obtain $\tilde{\mathbf{X}}$
5: **Step 2: Representation extraction for decoupling**
6: **for** each encoder $\mathcal{E}_j \in \{\mathcal{E}_{\text{S}}, \mathcal{E}_1, \ldots, \mathcal{E}_K\}$ **do**
7:     Temporal encoding: $\mathbf{H}_{\text{temp}} \leftarrow \mathcal{F}_{\text{temp}}(\tilde{\mathbf{X}})$
8:     Spatial retrieval: $\mathbf{H}_{\text{spatial}} \leftarrow \mathcal{F}_{\text{spatial}}(\mathbf{H}_{\text{temp}}, \mathbf{W}_{\text{C}}, \mathbf{W}_{\text{R}})$
9:     Patchify & embed: $\mathbf{Z} \leftarrow \text{PatchifyEmbed}(\mathbf{H}_{\text{spatial}})$
10:     Transformer encoding: $\mathbf{Z}_j \leftarrow \text{Transformer}(\mathbf{Z})$
11:     **if** $j \neq y_{\text{state}}$ and $j \neq$ shared **then**
12:         Detach $\mathbf{Z}_j$ (inactive encoders, used only for decoupling loss)
13:     **end if**
14: **end for**
15: **Step 3: Reconstruction**
16: Concatenate outputs: $\mathbf{Z}_{y_{state}}^{\text{comb}} \leftarrow \text{Concat}(\mathbf{Z}_{y_{\text{state}}}, \mathbf{Z}_{\text{shared}})$
17: Reconstruct masked input: $\hat{\mathbf{X}} \leftarrow \mathcal{D}_{y_{\text{state}}}(\mathbf{Z}_{y_{state}}^{\text{comb}})$
18: **Step 4: Loss computation**
19: Region-aware reconstruction loss: $\mathcal{L}_{\text{rec}} \leftarrow \sum w^{(y_{\text{state}})} \cdot (\mathbf{X} - \hat{\mathbf{X}})^2$
20: Brain-state decoupling loss: $\mathcal{L}_{\text{dec}} \leftarrow$ margin-based cosine loss
21: Total loss: $\mathcal{L}_{\text{BrainPro}} \leftarrow \mathcal{L}_{\text{rec}} + \mathcal{L}_{\text{dec}}$
22: **Step 5: Parameter update**
23: Update $\mathcal{E}_{\text{S}}$ and $\mathcal{E}_{y_{\text{state}}}$ (detach all other encoders)

---

# D  THEORETICAL JUSTIFICATION

In this section, we provide a theoretical analysis showing that (1) retrieval-based spatial filtering enables *explicit* encoding of spatial information across heterogeneous EEG channel montages, whereas (2) self-attention yields *implicit* spatial representations that do not correspond to stable channel identities.

## D.1  PRELIMINARIES

Let an EEG input be represented as $X \in \mathbb{R}^{C \times T}$, $\quad x(t) = X_{:,t}$, where $C$ is the number of channels and $T$ is the number of time samples. Each channel $c$ corresponds to a fixed scalp location $p_c \in \mathbb{R}^3$.

A spatial filter is parameterized by $w = (w_1, \ldots, w_C)^\top \in \mathbb{R}^C$, and produces the spatially filtered signal

$$y(t) = w^\top x(t) = \sum_{c=1}^{C} w_c X_{c,t}.$$

Self-attention operates on channel embeddings $H \in \mathbb{R}^{C \times d}$ via $Q = HW_Q, \quad K = HW_K, \quad V = HW_V$, and produces

$$Z = \text{softmax}\left(\frac{QK^\top}{\sqrt{d_k}}\right) V,$$

where $W_Q, W_K, W_V$ are shared projection matrices.

## D.2  EXPLICIT SPATIAL ENCODING VIA SPATIAL FILTERS

**Proposition 1** (Explicit Spatial Encoding). *If a model contains a learnable parameter $w_c$ that always multiplies the same channel $c$, then the model explicitly encodes spatial information because each parameter corresponds to a fixed scalp location $p_c$.*

*Proof.* The mapping channel $\to$ scalp location is fixed: $c \mapsto p_c$. Since $w_c$ always multiplies $X_{c,t}$, the contribution of scalp location $p_c$ to the output $y(t)$ is governed by a dedicated parameter. Thus spatial information is tied directly to parameters, yielding an explicit spatial representation. $\square$

## D.3  LACK OF EXPLICIT SPATIAL ENCODING IN SELF-ATTENTION

**Proposition 2** (Self-Attention Is Not Spatially Explicit). *Self-attention cannot explicitly encode spatial structure because none of its learnable parameters are tied to channel identity or scalp location. The attention weights $a_{ij}$ depend on input content rather than spatial location.*

*Proof.* Self-attention uses shared parameters $W_Q, W_K, W_V$ for all channels. For channels $i$ and $j$, the interaction weight

$$a_{ij} = \frac{\exp(q_i k_j^\top)}{\sum_{j'=1}^{C} \exp(q_i k_{j'}^\top)}$$

depends on the embeddings $h_i$ and $h_j$, which depend on the input signal and not on fixed locations $p_i$ or $p_j$. Thus $a_{ij}$ does not represent a stable spatial relationship across datasets or subjects. Since no parameter is associated with a fixed channel index, self-attention lacks explicit spatial semantics. $\square$

## D.4  RETRIEVAL-BASED ALIGNMENT RESTORES EXPLICIT SPATIAL STRUCTURE

Suppose dataset $A$ has channels $\{1, \ldots, C_A\}$ with locations $\{p_c^{(A)}\}$, and dataset $B$ has channels $\{1, \ldots, C_B\}$ with locations $\{p_c^{(B)}\}$. A retrieval function

$$\phi : \{1, \ldots, C_B\} \to \{1, \ldots, C_A\}$$

selects, for each channel $c$ in dataset $B$, the most similar channel in dataset $A$:

$$\phi(c) = \arg \min_{j \in \{1, \ldots, C_A\}} d(p_c^{(B)}, p_j^{(A)}),$$

for distance measure $d(\cdot, \cdot)$.

The spatial filter learned on dataset $A$, $w^{(A)} \in \mathbb{R}^{C_A}$, is transferred to dataset $B$ via

$$w_c^{(B)} = w_{\phi(c)}^{(A)}.$$

**Proposition 3** (Retrieval Restores Explicit Spatial Encoding)**.** *If $\phi$ maps channels in dataset $B$ to their spatially closest counterparts in dataset $A$, then the transferred filter $w^{(B)}$ preserves explicit spatial semantics across datasets.*

*Proof.* The retrieval function $\phi$ associates each channel $c$ in dataset $B$ with a channel $\phi(c)$ in dataset $A$ that corresponds to the closest spatial location. Thus $w_c^{(B)}$ inherits the spatial meaning of $w_{\phi(c)}^{(A)}$. By Proposition 1, $w_{\phi(c)}^{(A)}$ explicitly encodes the spatial contribution of a fixed location. Therefore, $w_c^{(B)}$ is tied to a stable spatial meaning, yielding an explicit spatial representation across heterogeneous channel configurations. $\square$

## D.5 CONCLUSION

Spatial filters produce explicit spatial representations because each parameter corresponds to a fixed anatomical location. Self-attention produces only implicit spatial structure because its interaction weights depend on sample-dependent feature similarity rather than channel identity. Retrieval-based alignment allows spatial filters to retain explicit spatial semantics even when aggregating data across heterogeneous EEG montages.

## E RELATED WORK

### E.1 PREVIOUS METHODS FOR EEG DECODING

Traditional EEG decoding relied on handcrafted features and classical classifiers such as common spatial patterns (CSP), linear discriminant analysis (LDA), and support vector machines (SVMs), which required extensive domain expertise and were often task-specific (Lotte et al., 2007). With the rise of deep learning, neural architectures emerged to learn spatiotemporal EEG representations directly from raw signals (Schirrmeister et al., 2017). CNN-based models captured spatial correlations between electrodes (Lawhern et al., 2018; Ding et al., 2023), while recurrent networks such as LSTMs modeled temporal dynamics (Wang et al., 2018; Michielli et al., 2019). Hybrid CNN–LSTM frameworks (Li et al., 2022) and attention-based architectures (Song et al., 2023; Ding et al., 2025) further advanced decoding performance, and graph neural networks were introduced to exploit functional brain connectivity (Song et al., 2020; Zhong et al., 2022; Ding et al., 2024). These approaches achieved strong results in tasks including motor imagery classification, seizure detection, emotion recognition, and sleep staging. However, they typically relied on supervised learning tailored to specific datasets, limiting scalability in the face of EEG's heterogeneous channel configurations, variable sampling rates, and low signal-to-noise ratios.

### E.2 EEG FOUNDATION MODELS

Inspired by the success of large language and vision models, recent studies have proposed *EEG foundation models* (EEG-FMs) pre-trained on large unlabeled corpora with self-supervised objectives (Zhou et al., 2025). These models aim to learn universal neural representations that can be adapted to diverse downstream BCI tasks. Brant (Zhang et al., 2023) and NeuroGPT (Cui et al., 2024), explored masked modeling and contrastive learning strategies to enhance cross-task generalization. BIOT (Yang et al., 2023) introduced a unified encoder that tokenizes heterogeneous biosignals into a sentence-like representation to support cross-dataset pre-training. LaBraM (Jiang et al., 2024) introduced a neural tokenizer and masked EEG modeling trained on 2,500 hours of data,

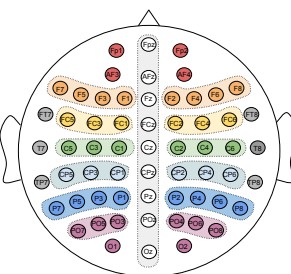

Figure 6: The brain region definition used in spatial learning block.

achieving state-of-the-art results in abnormal detection, event classification, and emotion recognition. EEGPT (Wang et al., 2024) developed a 10-million-parameter pretrained transformer with a dual mask-based self-supervised framework and spatio-temporal representation alignment to improve robustness under low SNR. CBraMod (Wang et al., 2025) employed a criss-cross transformer with asymmetric positional encoding to separately model spatial and temporal dependencies, achieving strong generalizability across diverse BCI datasets.

Despite recent progress, EEG-FMs still face key limitations: they approximate spatial interactions only implicitly or with fixed channels, lack decoupling of diverse brain processes (e.g., motor, emotion), and rely on a single shared encoder that limits flexibility for tasks involving overlapping processes. These constraints reduce adaptability and highlight the need for models that can explicitly capture spatial dependencies, process-specific representations, and mixture-of-experts style flexibility.

## F MORE DETAILS FOR THE BRAIN REGION DEFINITION

EEG captures activity from multiple functional areas of the brain. Treating electrodes as isolated nodes or relying only on global connections overlooks localized cooperative activity. LGGNet (Ding et al., 2024) addresses this by defining local brain regions and modeling both within-area activity (local graphs) and between-area interactions (global graphs), thereby capturing the brain's hierarchical organization. The hemisphere LGG further extends this structure by introducing symmetric subgraphs across hemispheres, motivated by evidence of inter-hemispheric asymmetries in cognitive and emotional processes. This enables more effective modeling of bilateral patterns and asymmetries, which is beneficial for tasks such as emotion recognition and preference prediction. Following LGGNet, BrainPro adopts the hemisphere-based LGG region definition with minor adjustments: Fp1/Fp2 are separated from the AF regions, O1/O2 from the PO regions, and the FT, T, and TP regions are split into distinct areas. The brain regions are shown in Figure 6.

## G MORE DETAILS FOR PRE-TRAINING

### G.1 PRE-TRAINING DATASETS

A detailed description of the pre-training datasets for BrainPro is provided here. Most of these datasets overlap with those used for LaBraM, but we excluded some to ensure that the pre-training data represent meaningful brain states. In addition, we incorporated two extra motor imagery datasets to provide sufficient data for training each brain process encoder. In total, the datasets comprise around **2400 hours**.

- **Emobrain** (Savran et al., 2006): Multimodal emotion dataset with EEG (64 channels, 1024 Hz) and fNIRS from 16 subjects, collected via Biosemi Active 2. Emotions elicited with a subset of IAPS stimuli. (4.94 h)

- **Grasp and Lift EEG Challenge** (Luciw et al., 2014): EEG from 12 subjects (32 channels, 500 Hz) performing grasp-and-lift trials with a BrainAmp EEG amplifier. (11.72 h)

Table 5: Hyperparameters for pre-training.

| Type | Factor | Value |
|------|--------|-------|
| Temporal Encoder | Input channels | {1, 32, 32} |
| | Output channels | {32, 32, 32} |
| | Kernel size | {15, 3, 3} |
| | Stride | {1, 1, 1} |
| | Padding | {7, 1, 1} |
| Spatial Encoder | Channel-wise Filter Number | 32 |
| | Region-wise Filter Number | 32 |
| | Total EEG Channel | 60 |
| | Total Brain Region | 24 |
| Patch Maker | Patch Length | 20 |
| | Patch Stride | 20 |
| | MLP size | 64 |
| Transformer Encoder | Layers | 4 |
| | Hidden size | 32 |
| | MLP size | 64 |
| | Attention heads | 32 |
| Transformer Decoder | Layers | 2 |
| | Hidden size | 32 |
| | MLP size | 64 |
| | Attention heads | 32 |
| Training | Batch size | 160 (32 per GPU) |
| | Peak learning rate | 1e-4 |
| | Minimal learning rate | 1e-5 |
| | Scheduler | Cosine |
| | Optimizer | AdamW |
| | Adam $\beta$ | (0.9, 0.98) |
| | Weight decay | 0.05 |
| | Total epochs | 30 |
| | Warmup epochs | 2 |
| | Gradient clipping | 3 |
| | Mask ratio | 0.5 |

- **EEG Motor Movement/Imagery** (Schalk et al., 2004): Recordings from 109 volunteers (64 channels, 160 Hz) performing baseline (eyes open/closed), movement, and imagery (both fists/feet) tasks using the BCI2000 system. (47.3 h)

- **KU** (Lee et al., 2019): Data from 54 subjects conducting motor imagery tasks. It was recorded with 62 channels, 1000 Hz. ($\sim$ 24.00 h)

- **HGD** (Schirrmeister et al., 2017): Data from over 14 subjects, recorded 128 channels, 500 Hz. ($\sim$7.49 h)

- **Raw EEG Data** (Trujillo, 2020): EEG (64 channels, 256 Hz) collected during information-integration and multidimensional rule-based categorization tasks. (34.35 h)

- **Resting State EEG Data** (Trujillo et al., 2017): EEG from 22 subjects during 8-minute resting (4 minutes eyes closed, 4 minutes eyes open), recorded with 64 channels at 256 Hz using BioSemi active Ag/AgCl electrodes. (3.04 h)

- **SEED Series** (Zheng & Lu, 2015; Zheng et al., 2018; Liu et al., 2022): Emotional EEG datasets including SEED (15 subjects), SEED-IV (15 subjects), SEED-GER (8 subjects), and SEED-FRA (8 subjects). Signals recorded at 1000 Hz from 62 channels with the ESI NeuroScan System while subjects viewed videos. (166.75 h)

- **SPIS Resting State** (Torkamani-Azar et al., 2020): EEG from 10 subjects (64 channels, 2048 Hz) with 2.5-minute eyes-open/closed sessions before a 105-minute sustained attention task. (0.83 h)

- **TUEP** (Veloso et al., 2017): Subset of TUEG with 100 epilepsy and 100 control subjects, verified by a neurologist; EEG recorded with 19–23 channels at 256 Hz. (591.22 h)

- **TUSZ** (Shah et al., 2018): Seizure-annotated EEG corpus (19–23 channels, 256 Hz) including onset, offset, channel, and seizure type information. (1138.53 h)

- **TUSL** (von Weltin et al., 2017): TUEG subset with slowing event annotations (23 channels, 256 Hz), used in seizure detection error analysis. (20.59 h)

- **SHJT EEG Data** (Jiang et al., 2023; 2021; Luo et al., 2022; Li et al., 2021; Tao & Lu, 2020): Data from over 140 subjects, recorded with the ESI NeuroScan System (62 channels, 1000 Hz). (342.23 h)

Table 6: Categories of the pre-training datasets.

| Brain Process Category | Datasets |
|---|---|
| Affect | Emobrain, SEED, SEED-IV, SEED-GER, SEED-FRA, and SHJT EEG Data |
| Motor | Grasp and Lift EEG Challenge, EEG Motor Movement/Imagery, KU, and HGD |
| Others | Raw EEG Data, Resting State EEG Data, SPIS Resting State, TUEP, TUSZ, TUSL, and TUSL |

### G.2 EXPERIMENT SETTINGS

We describe the experimental setting for pre-training as follows. All EEG recordings are segmented into 10-second clips and resampled to 200 Hz. We predefine a 60-channel montage based on the SEED-series datasets (excluding two reference channels) as the reference configuration. Channels from different pre-training datasets are mapped to this predefined montage according to their spatial distance on the scalp. Channels that are absent from the predefined set or cannot be reliably mapped are discarded. To enable process-specific representation learning, we categorize the datasets into three groups, *affect*, *motor* and *other*, for brain-process decoupling training (see Table 6 for details). To make the pre-training stable, we discard the noisy segments that have large absolute values resulting around 785,000 clear samples, around 2,180 hours. These samples from different datasets are shuffled before being fed into the model. To improve training efficiency, samples from the same dataset are grouped within each training batch, ensuring that the spatial configuration requires only one spatial filter retrieval. During training, the gradients of samples belonging to different brain processes are passed to their corresponding encoders and decoders, while the shared encoder processes all samples to capture universal EEG representations. Additional implementation details can be found in our released code (https://anonymous.4open.science/r/BrainPro-29A4/).

### G.3 HYPER-PARAMETERS

The pre-training of BrainPro is in an end-to-end manner, and the hyper-parameter settings are shown in Table 5.

### G.4 PRE-TRAINING RESULTS

The image in Figure 7 shows the training loss curve of BrainPro during pre-training. The curve starts above 1.0 and drops sharply within the first few thousand steps, indicating fast initial convergence. After this rapid decline, the loss continues to decrease, reaching below 0.1 by the end of training. Although some fluctuations are observed, but the overall downward trend remains. We saved checkpoints at epochs 10, 20, and 30, and selected the one from epoch 10 based on validation performance on the FACED dataset. This checkpoint was then used for all remaining downstream datasets.

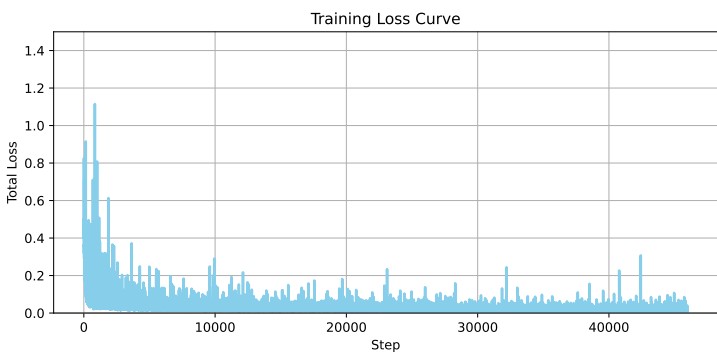

Figure 7: The pre-training loss of BrainPro.

## H    MORE DETAILS FOR DOWNSTREAM TASKS

### H.1    HYPER-PARAMETERS

Table 7 lists the fine-tuned hyper-parameters for each downstream task, while the common training hyper-parameters are shown in Table 8.

Table 7: Tuned hyper-parameters for downstream tasks.

| Dataset | Learning Rate | Hidden Factor | Dropout | Token Selection Mode |
|---|---|---|---|---|
| FACED | 5e-4 | 8 | 0.1 | All |
| SEED-V | 3e-4 | 1 | 0.1 | All |
| SEED-VII | 3e-4 | 8 | 0.1 | All |
| SHU | 1e-4 | 1 | 0.1 | Aggr |
| BCI-IV-2A | 1e-3 | 1 | 0.2 | Aggr |
| Mumtaz2016 | 1e-4 | 2 | 0.1 | Aggr |
| Mental Arithmetic | 1e-4 | 1 | 0.1 | Mean |
| ATTEN | 1e-3 | 4 | 0.1 | Aggr |
| BCIC2020-3 | 5e-4 | 1 | 0.1 | All |

Table 8: Fixed hyper-parameters for fine-tuning.

| Hyper-parameters | Settings |
|---|---|
| Epochs | 50 (Others) / 30 (ATTEN) |
| Batch size | 64 |
| Optimizer | AdamW |
| Adam $\beta$ | (0.9, 0.999) |
| Adam $\epsilon$ | 1e-8 |
| Weight decay | 5e-2 |
| Scheduler | CosineAnnealingLR |
| Cosine cycle epochs | 50 |
| Minimal learning rate | 1e-6 |
| Clipping gradient norm | 1 |
| Label smoothing (multi-class classification) | 0.1 |

### H.2    DOWNSTREAM DATASETS

We evaluate BrainPro on nine downstream datasets covering diverse BCI applications, including emotion recognition, motor imagery, imagined speech, mental disorder diagnosis, stress detection,

and attention decoding. These datasets differ substantially in subjects, channel configurations, sampling rates, trial durations, and class definitions, providing a comprehensive and challenging testbed for evaluating EEG foundation models. All EEG signals are resampled to 200 Hz, segmented into fixed-length windows according to task protocol, and the detailed training, validation, and test split is introduced in this section.

The **FACED** dataset (Chen et al., 2023) is used for the task of emotion recognition. It consists of 32-channel EEG recordings sampled at 250 Hz from 123 participants, each exposed to 28 video clips designed to elicit nine distinct emotional states: amusement, inspiration, joy, tenderness, anger, fear, disgust, sadness, and neutral. Each EEG trial lasts 10 seconds and is resampled to 200 Hz, yielding a total of 10,332 clean EEG segments. Following Wang et al. (2025), subjects 1–80 are used for training, 81–100 for validation, and 101–123 for testing, ensuring subject-independent evaluation.

The **SEED-V** dataset (Liu et al., 2021) is another benchmark for emotion recognition, covering five emotional categories: happy, sad, neutral, disgust, and fear. It contains 62-channel EEG recordings sampled at 1000 Hz from 16 subjects, each completing three sessions with 15 trials per session. The signals are segmented into 1-second windows and resampled to 200 Hz, resulting in 117,744 samples. Each session is evenly divided into training, validation, and testing subsets (5 trials each).

The **SEED-VII** dataset (Jiang et al., 2025) extends the SEED series to seven emotion categories: neutral, sad, fear, disgust, happy, surprise, and anger. It contains 62-channel EEG recordings at 1000 Hz from 20 subjects, each watching 80 film clips designed to elicit emotions in four sessions. Using the same pre-processing steps as SEED-V, EEG signals are segmented into 1-second windows and resampled to 200 Hz, producing 281,679 labeled samples. As there are 20 trials in each session, they are split at the trial level into training (10), validation (5), and test subsets (5).

The **BCI-IV-2A** dataset (Brunner et al., 2024) is a standard benchmark for motor imagery. It consists of EEG from 9 subjects recorded with 22 channels at 250 Hz while performing four classes of motor imagery: left hand, right hand, both feet, and tongue. Each trial lasts 4 seconds, and signals are resampled to 200 Hz. We applied a 0.1-70 Hz band-pass filter. In total, 5,088 trials are obtained. We follow subject-independent splits consistent with Wang et al. (2025), ensuring that train, validation, and test partitions are disjoint at the subject level.

The **SHU-MI** dataset (Ma et al., 2022) supports binary motor imagery classification. EEG was collected from 25 participants imagining either left- or right-hand movements, recorded with 32 channels at 250 Hz. The data are resampled to 200 Hz and segmented into 4-second non-overlapping windows, yielding 11,988 labeled samples. Following Wang et al. (2025), subjects 1–15 are used for training, 16–20 for validation, and 21–25 for testing.

The **BCIC2020-3** dataset (Jeong et al., 2022) focuses on imagined speech recognition. It includes EEG recordings from 64 channels at 256 Hz, where 15 participants imagine speaking one of five words. Each trial lasts 4 seconds and is resampled to 200 Hz, yielding 5,250 labeled samples. We use the official validation set as our test set, and the official training data are further divided into training and validation sets at a ratio of 9:1.

The **Mumtaz2016** dataset (Mumtaz, 2016) is designed for mental disorder diagnosis, specifically distinguishing patients with major depressive disorder (MDD) from healthy controls. EEG signals were recorded from 19 electrodes at 256 Hz, preprocessed with 0.3–75 Hz bandpass and 50 Hz notch filters, and then resampled to 200 Hz. Segments of 4 seconds are extracted, producing 3,525 labeled samples. We follow Wang et al. (2025) for subject-wise splits to ensure reliable evaluation.

The **Mental Arithmetic** dataset (Zyma et al., 2019) supports the task of mental stress detection. EEG was recorded from 36 subjects using 20 electrodes at 500 Hz, under two cognitive states: resting ("no stress") and active mental arithmetic ("stress"). The data are resampled to 200 Hz, band-pass filtered (0.5–45 Hz), and segmented into 4-second windows, producing 1,080 labeled samples. We under-sample the class with more data samples to make the class balanced as Ding et al. (2024). Following Wang et al. (2025), subjects 1–28 are used for training, 29–32 for validation, and 33–36 for testing.

The **ATTEN** dataset is used for mental attention detection. It consists of EEG data from 26 subjects, recorded using 28 channels at 500 Hz during the Discrimination/Selection Response (DSR) task, which assesses cognitive attention. We adopt the pre-processing steps described in (Ding et al., 2025). The signals are resampled to 200 Hz and segmented into 4-second non-overlapping windows,

yielding 4,680 labeled samples. Subjects 1–20 are used as the training set, subjects 21–23 as the validation set, and the last 3 subjects as the test set.

### H.3 BASELINES

**EEGNet** (Lawhern et al., 2018): It is a lightweight convolutional neural network tailored for EEG-based BCI applications. By employing depthwise and separable convolutions, it efficiently extracts discriminative EEG features, enabling effective and computationally efficient EEG decoding.

**Conformer** (Song et al., 2023): It integrates CNNs with Transformers to model the spatio-temporal characteristics of EEG signals. It employs 1-D CNNs to extract local features and leverages a self-attention mechanism to capture global temporal dependencies.

**BIOT** (Yang et al., 2023): It is a transformer-based model developed to address cross-dataset EEG classification challenges under domain shifts. It incorporates a domain-invariant attention mechanism alongside contrastive representation learning, which together improve generalization across diverse recording setups and subject groups.

**LaBraM** (Jiang et al., 2024): It introduces a scalable transformer framework designed to learn universal EEG representations from large-scale brain signal datasets. Through pretraining on a broad range of EEG recordings, the model captures comprehensive temporal and spatial characteristics that can be effectively transferred to downstream BCI tasks. Its architecture integrates efficient self-attention operations with task-specific adapters, supporting flexible fine-tuning for varied applications.

**EEGPT** Wang et al. (2024): It adopts a dual self-supervised learning paradigm that combines masked autoencoding with spatio-temporal representation alignment. By emphasizing high signal-to-noise ratio (SNR) features rather than raw inputs, it improves the quality of learned representations. The model employs a hierarchical design that decouples spatial and temporal processing, thereby enhancing both computational efficiency and adaptability across BCI scenarios.

**CBraMod** Wang et al. (2025): It is a transformer-based EEG foundation model tailored to capture the complex spatial and temporal dependencies in EEG data. It introduces a criss-cross transformer structure with parallel spatial and temporal attention modules, enabling simultaneous yet independent modeling of spatial and temporal dynamics.

### H.4 EVALUATION METRICS

In this section, we describe the evaluation metrics adopted in this work. Following LaBraM (Jiang et al., 2024; Wang et al., 2025), we employ the following measures:

- **Balanced Accuracy**: Accounts for class imbalance by computing the average recall across all classes. It is applied in both binary and multi-class classification settings.

- **AUC-PR**: The area under the precision-recall (PR) curve, used to evaluate performance in binary classification tasks.

- **AUROC**: The area under the receiver operating characteristic (ROC) curve, a standard metric for assessing binary classification performance.

- **Cohen's Kappa**: A statistical measure of inter-rater agreement, commonly applied to imbalanced multi-class classification problems to quantify consistency beyond chance.

- **Weighted F1**: The weighted average of class-wise F1-scores, where each class is weighted by its sample size. This provides a robust assessment of performance in multi-class classification tasks.

## I   MORE EXPERIMENTAL RESULTS

### I.1   SEED-VII

On the SEED-VII dataset (Table 9), we observe lower overall performance compared to FACED or SEED-V, highlighting the challenge of distinguishing seven closely related emotions under high

inter-subject variability. Simpler models such as EEGNet and EEGPT perform poorly, while Conformer and BIOT provide moderate improvements. Foundation models achieve stronger results, with LaBraM performing best overall and CBraMod showing competitive accuracy. BrainPro reaches 0.3315/0.2223/0.3350 (ACC-B/Kappa/F1-W), closely matching CBraMod and LaBraM. Although LaBraM slightly outperforms BrainPro, the margin is small, and BrainPro maintains stable generalization, confirming its effectiveness even in this demanding multi-class setting. This suggests that BrainPro's retrieval-based spatial modeling and state-decoupling contribute to consistent robustness across diverse emotion recognition tasks.

Table 9: Results on the SEED-VII dataset.

| Methods | ACC-B | Kappa | F1-W |
|---|---|---|---|
| EEGNet | 0.1892 ± 0.0056 | 0.0562 ± 0.0071 | 0.1472 ± 0.0123 |
| Conformer | 0.2719 ± 0.0032 | 0.1566 ± 0.0039 | 0.2712 ± 0.0033 |
| BIOT | 0.3005 ± 0.0054 | 0.1859 ± 0.0070 | 0.3055 ± 0.0069 |
| EEGPT | 0.2213 ± 0.0060 | 0.0954 ± 0.0071 | 0.2147 ± 0.0120 |
| LaBraM | **0.3346** ± 0.0122 | **0.2256** ± 0.0146 | **0.3391** ± 0.0121 |
| CBraMod | 0.3318 ± 0.0056 | 0.2236 ± 0.0065 | 0.3384 ± 0.0059 |
| BrainPro | 0.3315 ± 0.0038 | 0.2223 ± 0.0055 | 0.3350 ± 0.0045 |

Note: **Bold** indicates the best performance. Cyan highlight marks our BrainPro.

## I.2 MENTAL DISORDER DIAGNOSIS

On the MDD dataset, all methods achieve relatively high accuracy, suggesting that distinguishing patients from controls is more tractable than fine-grained emotion recognition. EEGNet and Conformer already reach strong performance, while BIOT and LaBraM offer further improvements. BrainPro achieves 0.9161/0.9831/0.9803 (ACC-B/AUC-PR/AUROC), outperforming all baselines across metrics. These results confirm that BrainPro provides robust and reliable representations for clinical EEG applications, where high sensitivity and specificity are crucial.

Table 10: Results on the Major Depressive Disorder dataset.

| Methods | ACC-B | AUC-PR | AUROC |
|---|---|---|---|
| EEGNet | 0.9113 ± 0.0104 | 0.9632 ± 0.0045 | 0.9512 ± 0.0096 |
| Conformer | 0.8422 ± 0.0344 | 0.9442 ± 0.0502 | 0.9613 ± 0.0064 |
| BIOT | 0.8789 ± 0.0190 | 0.9744 ± 0.0083 | 0.9664 ± 0.0136 |
| EEGPT | 0.8475 ± 0.0933 | 0.9695 ± 0.0076 | 0.9669 ± 0.0069 |
| LaBraM | 0.8986 ± 0.0018 | 0.9791 ± 0.0041 | 0.9754 ± 0.0050 |
| CBraMod | 0.8909 ± 0.0037 | 0.9769 ± 0.0047 | 0.9726 ± 0.0062 |
| BrainPro | **0.9161** ± 0.0054 | **0.9831** ± 0.0038 | **0.9803** ± 0.0046 |

Note: **Bold** indicates the best performance. Cyan highlight marks our BrainPro.

## I.3 SPEECH

On the imagined speech dataset, performance is generally modest across methods, reflecting the inherent difficulty of decoding internally generated speech from EEG. Simpler models such as EEGNet show very low balanced accuracy, while Conformer and EEGPT achieve only small gains. Foundation models perform better, with LaBraM and CBraMod leading the results. BrainPro achieves 0.5253/0.4067/0.5244 (ACC-B/Kappa/F1-W), the highest among all methods, indicating that retrieval-based spatial learning and state-decoupling effectively capture subtle neural dynamics underlying imagined speech.

Table 11: Results on the Speech dataset.

| Methods | ACC-B | Kappa | F1-W |
|---|---|---|---|
| EEGNet | 0.2755 ± 0.0183 | 0.0943 ± 0.0228 | 0.2737 ± 0.0180 |
| Conformer | 0.3339 ± 0.0104 | 0.1673 ± 0.0130 | 0.3269 ± 0.0153 |
| BIOT | 0.3016 ± 0.0206 | 0.1270 ± 0.0257 | 0.2999 ± 0.0217 |
| EEGPT | 0.2787 ± 0.0061 | 0.0983 ± 0.0076 | 0.2758 ± 0.0060 |
| LaBraM | 0.4880 ± 0.0161 | 0.3600 ± 0.0201 | 0.4871 ± 0.0165 |
| CBraMod | 0.5061 ± 0.0211 | 0.3827 ± 0.0263 | 0.5050 ± 0.0212 |
| BrainPro | **0.5253** ± 0.0084 | **0.4067** ± 0.0105 | **0.5244** ± 0.0090 |

Note: **Bold** indicates the best performance. Cyan highlight marks our BrainPro.

Table 12: Robustness under random channel dropping on FACED. Values show mean ± std, and $\Delta$ shows performance change relative to 0%.

| Drop | BrainPro | | CBraMod | | LaBraM | |
|---|---|---|---|---|---|---|
| | Value | $\Delta$ | Value | $\Delta$ | Value | $\Delta$ |
| | ACC-B | | | | | |
| 0 | 0.5937 ± 0.0087 | 0 | 0.5669 ± 0.0094 | 0 | 0.5224 ± 0.0116 | 0 |
| 0.1 | 0.5747 ± 0.0014 | -0.0190 | 0.5218 ± 0.0042 | -0.0451 | 0.4972 ± 0.0036 | -0.0252 |
| 0.2 | 0.5373 ± 0.0015 | -0.0564 | 0.4890 ± 0.0054 | -0.0779 | 0.4725 ± 0.0086 | -0.0499 |
| 0.3 | 0.4995 ± 0.0056 | -0.0942 | 0.4473 ± 0.0035 | -0.1196 | 0.4230 ± 0.0085 | -0.0994 |
| 0.4 | 0.4639 ± 0.0054 | -0.1298 | 0.4111 ± 0.0094 | -0.1558 | 0.3851 ± 0.0084 | -0.1373 |
| 0.5 | 0.4270 ± 0.0076 | -0.1667 | 0.3671 ± 0.0036 | -0.1998 | 0.3397 ± 0.0036 | -0.1827 |
| | Kappa | | | | | |
| 0 | 0.5418 ± 0.0092 | 0 | 0.5112 ± 0.0110 | 0 | 0.4610 ± 0.0126 | 0 |
| 0.1 | 0.5187 ± 0.0015 | -0.0231 | 0.4585 ± 0.0045 | -0.0527 | 0.4319 ± 0.0043 | -0.0291 |
| 0.2 | 0.4772 ± 0.0016 | -0.0646 | 0.4234 ± 0.0060 | -0.0878 | 0.4049 ± 0.0101 | -0.0561 |
| 0.3 | 0.4352 ± 0.0063 | -0.1066 | 0.3775 ± 0.0039 | -0.1337 | 0.3490 ± 0.0097 | -0.1120 |
| 0.4 | 0.3949 ± 0.0054 | -0.1469 | 0.3365 ± 0.0111 | -0.1747 | 0.3057 ± 0.0094 | -0.1553 |
| 0.5 | 0.3547 ± 0.0093 | -0.1871 | 0.2876 ± 0.0037 | -0.2236 | 0.2544 ± 0.0046 | -0.2066 |
| | F1-W | | | | | |
| 0 | 0.6023 ± 0.0061 | 0 | 0.5729 ± 0.0105 | 0 | 0.5259 ± 0.0108 | 0 |
| 0.1 | 0.5785 ± 0.0020 | -0.0238 | 0.5228 ± 0.0041 | -0.0501 | 0.4977 ± 0.0041 | -0.0282 |
| 0.2 | 0.5419 ± 0.0011 | -0.0604 | 0.4919 ± 0.0050 | -0.0810 | 0.4736 ± 0.0088 | -0.0523 |
| 0.3 | 0.5053 ± 0.0055 | -0.0970 | 0.4525 ± 0.0029 | -0.1204 | 0.4226 ± 0.0082 | -0.1033 |
| 0.4 | 0.4699 ± 0.0053 | -0.1324 | 0.4165 ± 0.0094 | -0.1564 | 0.3817 ± 0.0091 | -0.1442 |
| 0.5 | 0.4329 ± 0.0078 | -0.1694 | 0.3728 ± 0.0040 | -0.2001 | 0.3349 ± 0.0038 | -0.1910 |

## J  EFFECTS OF LEARNING RATE

Figure 8 shows how different learning rates affect BrainPro on BCI-IV-2a and Mental Arithmetic. On BCI-IV-2a, the highest discriminative performance is achieved at a relatively larger learning rate $(1 \times 10^{-3})$, whereas very small values reduce accuracy and kappa, suggesting that sufficient step size is needed for stable convergence on motor-related tasks. In contrast, Mental Arithmetic benefits from smaller learning rates, with accuracy gradually improving as the rate decreases, while global metrics such as AUC-PR and AUROC remain largely stable. Although these results are based on only two datasets, they highlight that learning rate can influence fine-tuning in a dataset-dependent manner, and that adapting optimization strategies to task characteristics may further enhance Brain-Pro's generalization ability.

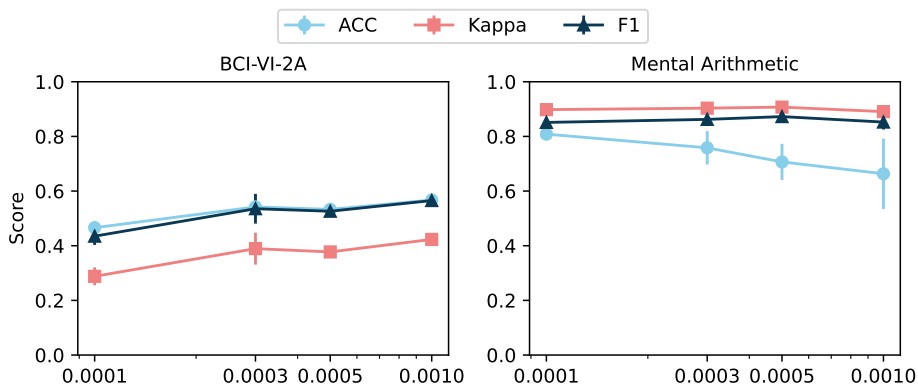

Figure 8: Effects of learning rate on BrainPro pre-training.

## K EFFECTS OF MODEL SIZE

Figure 9 shown the effects on performance with varying model size using FACED dataset. We evaluated models of varying sizes ranging from 10.04M to 33.45M parameters. The results indicate a general trend of performance improvement with increasing model size up to around 28.25M parameters, where the highest scores are observed across ACC-B (0.5980), Kappa (0.5458), and F1-W (0.6034). Beyond this point, at 33.45M, the performance plateaus or slightly declines, suggesting diminishing returns from further scaling. Standard deviations are relatively low for larger models compared to the smallest configuration (10.04M), implying more stable training outcomes as the model grows. Overall, these findings suggest that moderately larger models (about 28M parameters) strike the best balance between performance and stability for the FACED dataset.

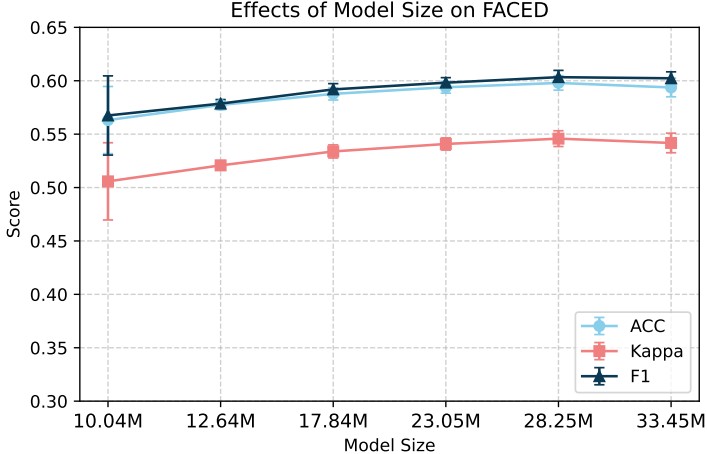

Figure 9: Effect of Model Size on Performance (FACED).

## L EFFECT OF ENCODER COUNT AND MLP SIZE ON PERFORMANCE

**Using multiple encoders improves performance over single-encoder variants, while enlarging the MLP alone does not.** Table 13 shows how downstream performance varies with different encoder counts and MLP sizes. Models that use only a single encoder perform worse on both FACED and BCI-IV-2A, even when their MLP size is increased to match or exceed that of multi-encoder configurations. In contrast, BrainPro, which uses two encoders, consistently outperforms the single-encoder variants despite having a comparable MLP ratio. Although adding a third encoder

Table 13: Effect of encoder count and relative MLP size on performance. MLP Ratio is computed w.r.t. BrainPro.

| Encoder No. | FACED | | | | BCI-IV-2A | | | |
|---|---|---|---|---|---|---|---|---|
| | ACC-B | Kappa | F1-W | MLP Ratio | ACC-B | Kappa | F1-W | MLP Ratio |
| 1 | 0.5635 ± 0.0031 | 0.5088 ± 0.0050 | 0.5710 ± 0.0049 | 0.51 | 0.5527 ± 0.0218 | 0.4036 ± 0.0290 | 0.5504 ± 0.0236 | 0.59 |
| 1 | 0.5808 ± 0.0048 | 0.5255 ± 0.0068 | 0.5840 ± 0.0079 | 1.02 | 0.5403 ± 0.0206 | 0.3871 ± 0.0275 | 0.5368 ± 0.0245 | 1.16 |
| 2 | 0.5937 ± 0.0087 | 0.5418 ± 0.0092 | 0.6023 ± 0.0061 | 1.00 | 0.5674 ± 0.0148 | 0.4232 ± 0.0198 | 0.5653 ± 0.0169 | 1.00 |
| 3 | 0.5938 ± 0.0069 | 0.5427 ± 0.0069 | 0.6039 ± 0.0042 | 0.93 | **0.5889 ± 0.0165** | **0.4519 ± 0.0220** | **0.5868 ± 0.0157** | 1.07 |
| 3 | **0.5988 ± 0.0057** | **0.5477 ± 0.0058** | **0.6098 ± 0.0043** | **1.49** | 0.5797 ± 0.0229 | 0.4396 ± 0.0305 | 0.5764 ± 0.0245 | 1.41 |

Note: **Bold** marks the best performance per dataset block. Cyan highlight marks BrainPro.

can provide additional gains in some settings, the main performance improvement arises from using more than one encoder rather than from simply increasing model capacity. These results indicate that architectural structure plays a larger role in performance than scaling the MLP alone, specifically incorporating multiple complementary encoders.

## M EFFECTS OF CLEAR DATA PRE-TRAINING

Figure 10 illustrates the effect of noisy samples during pre-training. On the BCI-IV-2a dataset, models pre-trained with clear data outperform those trained with noisy data across all classification metrics, with substantial gains in accuracy, Cohen's kappa, and F1-score. On the Mental Arithmetic dataset, the differences are less pronounced, with only moderate improvements in AUC-PR and AUROC, and in some cases noisy pre-training yields comparable results. These findings suggest that while global performance metrics are relatively robust to noise, discriminability is significantly compromised when noisy data are included in pre-training. This emphasizes that high-quality pre-training data is critical for enabling BrainPro to learn robust and generalizable EEG representations.

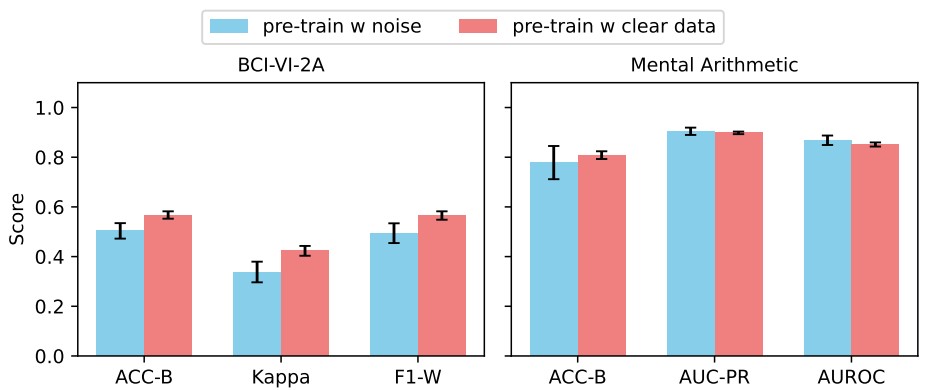

Figure 10: Effects of noisy samples on BrainPro pre-training.

## N EFFECTS OF RESETTING THE TEMPORAL POSITION EMBEDDING

Table 14 reports the results of resetting vs. not resetting temporal position embeddings on both BCI-VI-2A and Mental Arithmetic datasets. The results show that resetting consistently improves performance across all metrics. On BCI-VI-2A, resetting yields notable gains, with ACC-B increasing from 0.5222 to 0.5674 and Kappa improving from 0.3630 to 0.4232, indicating better discriminative ability and robustness. Similarly, for the Mental Arithmetic dataset, resetting provides smaller but consistent improvements across ACC-B, AUC-PR, and AUROC.

The resetting strategy is applied after loading the pre-trained weights by re-initializing the temporal position embeddings with a Xavier uniform distribution. Importantly, after resetting, the embeddings

remain learnable and are optimized during downstream fine-tuning. This ensures that the model retains the ability to encode task-specific temporal dynamics, while avoiding potential misalignment introduced by directly transferring positional priors from pre-training. The motivation behind this strategy lies in the observation that, although pre-training captures general brain state patterns, the temporal ordering of such states may vary substantially across unseen downstream tasks. Resetting the embeddings allows the model to relearn temporal structures tailored to the target dataset, which likely explains the consistent improvements observed

Table 14: Effect of resetting temporal position embedding on BCI-VI-2A and Mental Arithmetic datasets.

| Setting | BCI-VI-2A | | | Mental Arithmetic | | |
|---------|-----------|-------|------|-------------------|--------|-------|
| | ACC-B | Kappa | F1-W | ACC-B | AUC-PR | AUROC |
| w/o reset | 0.5222 ± 0.0406 | 0.3630 ± 0.0541 | 0.5109 ± 0.0490 | 0.8017 ± 0.0122 | 0.8955 ± 0.0113 | 0.8471 ± 0.0184 |
| reset | **0.5674** ± 0.0148 | **0.4232** ± 0.0198 | **0.5653** ± 0.0169 | **0.8083** ± 0.0156 | **0.8980** ± 0.0052 | **0.8512** ± 0.0083 |

**Note:** Bold values indicate superior performance compared to the non-reset baseline.

## O    EFFECTS OF DIFFERENT TOKEN SELECTION MODE

Table 15 compares different token selection strategies for constructing segment-level representations. On BCI-VI-2a, the aggregation-based strategy ("Aggr") achieves the best performance across all three metrics, outperforming both mean pooling and using all tokens, which suggests that aggregation provides a more compact and discriminative summary of motor imagery signals. In contrast, on the Mental Arithmetic dataset, mean pooling yields the best results on all metrics, with substantial gains in AUC-PR and AUROC, highlighting that a simpler averaging strategy is more effective for cognitive-related signals. Using all tokens performs worse on both datasets, likely due to redundancy and noise accumulation. These results demonstrate that the optimal token merging strategy is dataset-dependent, and that BrainPro's flexible design allows for effective adaptation across different EEG domains.

Table 15: Comparison of different token selection strategies on BCI-VI-2A and Mental Arithmetic datasets.

| Token Merge | BCI-VI-2A | | | Mental Arithmetic | | |
|-------------|-----------|-------|------|-------------------|--------|-------|
| | ACC-B | Kappa | F1-W | ACC-B | AUC-PR | AUROC |
| Aggr | **0.56736** ± 0.01482 | **0.42315** ± 0.01976 | **0.56531** ± 0.01691 | 0.73667 ± 0.03151 | 0.83525 ± 0.01750 | 0.78222 ± 0.01410 |
| Mean | 0.40903 ± 0.04703 | 0.21204 ± 0.06272 | 0.36640 ± 0.07028 | **0.80833** ± 0.01559 | **0.89800** ± 0.00520 | **0.85122** ± 0.00827 |
| All | 0.53733 ± 0.03154 | 0.38310 ± 0.04205 | 0.52952 ± 0.03350 | 0.58833 ± 0.03466 | 0.67525 ± 0.01305 | 0.62933 ± 0.01600 |

**Note: Bold** indicates the best performance for each dataset.

Table 16: Comparison of model complexity in terms of parameters and FLOPs.

| Method | Parameters | FLOPs |
|--------|-----------|-------|
| EEGNet | 0.005M | 10.03M |
| Conformer | 0.17M | 50.09M |
| BIOT | 3.28M | 229.35M |
| EEGPT | 51.66M | 9.31G |
| LaBraM | 9.43M | 392.68M |
| CBraMod | 8.45M | 350.79M |
| BrainPro | 7.69M | 531.04M |

## P    PARAMETERS SIZE AND FLOPS

Table 16 compares the parameter counts and FLOPs of different EEG models, using the BCI-IV-2A dataset as the example for evaluation. Lightweight baselines such as EEGNet and Conformer have very few parameters (0.005M and 0.17M, respectively) and correspondingly low FLOPs, making them efficient but limited in representational capacity. In contrast, LaBraM, BIOT, EEGPT,

CBraMod, and our proposed BrainPro are representative large EEG models, with parameter counts ranging from 3M to over 50M and FLOPs spanning hundreds of millions to several billions. Among them, LaBraM is reported with all tokens enabled, since this configuration achieves stronger performance despite higher computational cost. For a fair comparison, the MLP structure and token selection strategy are kept consistent across all foundation models. Notably, BrainPro contains 7.69M parameters and requires 531.04M FLOPs, positioning it as a mid-sized large model that balances complexity and efficiency. Compared to EEGPT (51.66M parameters, 9.31G FLOPs), BrainPro achieves competitive model capacity at a fraction of the computational cost, demonstrating a favorable trade-off between scalability and practicality.

## Q  VISUALIZATION OF LEARNED SPATIAL FILTERS IN PRE-TRAINING

The visualization of the learned spatial filters highlights the self-explainable nature of the retrieval-based spatial learning framework. As shown in Figure 11, the affect encoders predominantly emphasize frontal and temporal regions, which are consistent with neural substrates associated with emotional processing (Alarcão & Fonseca, 2019; Gao et al., 2021), whereas the motor encoders focus on central and parietal regions, aligning with sensorimotor activity (Pfurtscheller et al., 2006). In contrast, the shared encoders exhibit more diverse and distributed spatial patterns, suggesting that they capture cross-domain representations that generalize across both affective and motor states. These results demonstrate that the proposed pre-training strategy learns spatial filters that not only support downstream performance but also provide neuroscientifically meaningful insights into the types of information encoded.

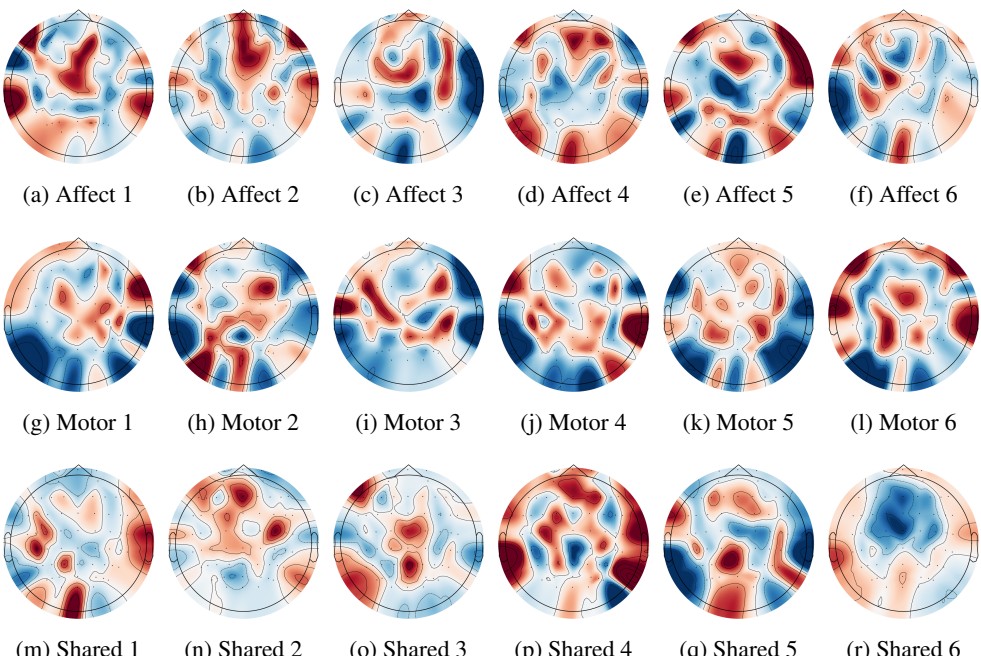

(a) Affect 1    (b) Affect 2    (c) Affect 3    (d) Affect 4    (e) Affect 5    (f) Affect 6

(g) Motor 1    (h) Motor 2    (i) Motor 3    (j) Motor 4    (k) Motor 5    (l) Motor 6

(m) Shared 1   (n) Shared 2   (o) Shared 3   (p) Shared 4   (q) Shared 5   (r) Shared 6

Figure 11: Visualization of learned filters for three groups of encoders after pre-training: Affect (top row), Motor (middle row), and Shared (bottom row).

## R  DISCUSSION

**New insights.**  This work shows that incorporating brain state awareness into large EEG models yields consistent improvements across diverse decoding tasks. The spatial filters learned by the affect, motor, and shared encoders form distinct and interpretable topographies, and the channel-drop analysis confirms that these spatial representations remain effective even when electrodes are missing. The performance of using the shared with different state-specific encoders indicate that the

state-specific encoders capture task-aligned and complementary features, providing measurable benefits beyond the shared encoder. Moreover, for BrainPro, the multi-encoder setting outperform it's single-encoder version even with comparable parameter counts, suggesting that architectural structure, rather than model capacity alone, drives the observed gains. Overall, the results highlight that explicit spatial modeling enabled by retrieval and brain state-aware representation learning jointly contribute to BrainPro's performance advantages across EEG tasks.

**Limitations.** Despite these advances, BrainPro faces several limitations. First, the current brain state taxonomy (affect, motor, others) is coarse-grained and does not capture finer distinctions such as attention, memory, or language processing, potentially restricting specialization. Second, our selective update strategy increases training complexity and may lead to imbalanced optimization when some states are underrepresented in the pre-training corpus. Although the pre-training loss exhibits short-term oscillations, we did not observe any degradation in the quality of the learned representations. BrainPro achieves consistent improvements across diverse downstream tasks, suggesting that these fluctuations do not hinder effective representation learning. Finally, BrainPro uses a 60-channel universal template because it matches the density of most large-scale EEG datasets and ensures all template positions receive sufficient supervision. Although this does not fully utilize every electrode in very high-density (128–256 channel) systems, the retrieval mechanism remains compatible by mapping each electrode to its nearest template anchor. Extending the template to higher resolution is straightforward and will be explored as richer high-density datasets become available.

**Future work.** Several avenues remain open for future exploration. Extending BrainPro to a richer set of brain states, potentially through hierarchical or dynamically discovered state encoders, could improve generalizability and interpretability. Incorporating multimodal neurophysiological signals (e.g., fNIRS, MEG, or eye tracking) may further enrich representation learning and support cross-modal transfer. Improving efficiency, for example via parameter sharing, pruning, or distillation, will be critical for on-device BCI applications. Another promising direction is to integrate interpretable mechanisms (e.g., attention maps aligned with neurophysiological knowledge) to better understand how shared and state-specific representations contribute to decoding. Ultimately, BrainPro opens a path toward scalable and adaptive large EEG models that more faithfully reflect the diversity of human brain states and support practical BCI deployment.

## THE USE OF LARGE LANGUAGE MODELS (LLMS)

In preparing this manuscript, we made use of a large language model (ChatGPT) to assist with grammar checking and language polishing. The model was not used for idea generation, experimental design, data analysis, or interpretation of results; all scientific content and conclusions are solely the work of the authors. The use of ChatGPT was limited to improving readability and clarity of expression.

