# OpenReview forum: "BrainPro: Towards Large-scale Brain State-aware EEG Representation Learning"
_ICLR.cc/2026/Conference — Submitted to ICLR 2026_

### Official Review · Reviewer_8DR1 · 2025-10-21

**Soundness:** 3
**Presentation:** 2
**Contribution:** 2
**Rating:** 4
**Confidence:** 4

**Summary:**

The paper introduces BrainPro, an EEG foundation model that incorporates retrieval-based spatial learning and brain state decoupling to handle variable electrode layouts and diverse brain states. The method shows competitive performance across nine public BCI datasets.

**Strengths:**

The paper is well-organized and clearly presented. The proposed method meaningfully integrates both brain-region and channel-level encoding, which aligns well with the neurophysiological structure of EEG signals. The evaluation is comprehensive, and the authors provide ample additional experiments and analyses in the appendix to support their claims.

**Weaknesses:**

1. While the paper claims to use parallel encoders to disentangle brain-state-specific representations, in practice only two specialized encoders are implemented (for affect and motor tasks), with all other brain states collapsed into a generic “other” encoder. Given that affect and motor represent only a small subset of possible brain states, this coarse categorization significantly limits the generality and practical impact of the proposed disentanglement strategy.

2. The paper lacks clarity on how encoder aggregation is performed during downstream fine-tuning. Although the authors argue that BrainPro can flexibly activate task-relevant encoders, Table 13 shows that using all encoders yields the best performance on affect tasks. Since the total model size is modest (7.69M parameters) and activating all encoders incurs minimal computational overhead, it is unclear why a selective activation mechanism is necessary—raising questions about the actual utility of the proposed modular design.

3. The method section is difficult to follow due to an excessive amount of notation, some of which is either undefined (e.g., Sp​ on line 210) or appears to contain errors (e.g., “KTT” on line 197). Additionally, the formulation in Equation 9 is ambiguous—specifically, it is unclear whether the positional embedding p is added before or after the flattening operation (line 213). The authors should thoroughly revise and reorganize this section to improve clarity and readability.

4. While the paper states that BrainPro supports heterogeneous electrode layouts through its retrieval-based spatial learning block, the current implementation appears to rely on a fixed universal template of only 60 channels. This may limit its applicability to datasets or clinical protocols that use denser montages (e.g., 62, 128, or even 256 channels), where fine-grained spatial resolution is critical—particularly for tasks like source localization or high-fidelity cognitive decoding. The authors should clarify whether the framework can scale to higher-density setups, and if so, how the retrieval mechanism generalizes beyond the 60-channel prior. If not, this constitutes a practical limitation that should be acknowledged.

5. The pretraining corpus appears to be dominated by datasets such as TUH EEG Seizure Corpus (TUSZ) and TUH Abnormal EEG Corpus (TUEP). According to the authors’ own categorization, these datasets would fall under the “other” brain state, meaning that during pretraining—per Equation 13—only the shared encoder and the “other”-specific encoder receive gradients, while the affect- and motor-specific encoders remain largely unused. This leads to highly imbalanced parameter updates across encoders during pretraining. I suspect this issue stems primarily from the coarse brain-state taxonomy (affect / motor / other), which oversimplifies the rich diversity of neural processes in real-world EEG data. A more fine-grained and neuroscientifically grounded state partitioning could substantially strengthen the model’s design and the paper’s overall contribution.

**Questions:**

The experiments in Table 13 are central to validating the paper’s core claim — that BrainPro enables flexible, task-adaptive encoder activation. However, the current ablation only shows a subset of combinations. To better support this contribution, the authors should provide a complete set of comparisons, including:
$\mathcal{E}$(S), $\mathcal{E}$(S+A), $\mathcal{E}$(S+O), $\mathcal{E}$(S+M), and $\mathcal{E}$(S+A+O+M).
This would clarify whether performance gains stem from specific state encoders or simply from increased model capacity, and help assess the true value of modular design.

---

> ### Author Response · Authors · 2025-11-23
> **[1/6] To reviewer 8DR1(W1)**
>
> [1/6] To reviewer 8DR1(W1)
> We thank the reviewer for the constructive feedback. Before presenting the **Summary of Comment(s)**, **Response**, and **Modifications in the Revised Manuscript** for each issue, we clarify the organization of our rebuttal:
>
> • **Three-part structure for every comment**
>   – **Summary of Comment(s):** brief restatement of the reviewer’s point.
>   – **Response:** our explanation, justification, and supporting evidence.
>   – **Modifications in the Revised Manuscript:** exact changes made and where they appear.
>
> • **Table/Figure indexing convention**
>   – In the **rebuttal**, tables and figures use **letter indices** (e.g., *Table a*, *Fig. b*) to avoid confusion and keep the document compact.
>   – In the **revised manuscript**, tables and figures use **numbered indices** (e.g., *Table 3*, *Figure 4*).
>   – Each table/figure appears **only once** in the rebuttal; subsequent responses referring to the same analysis use the same letter index.
>
> • **References**
>   – All cited references used in the responses are collected **once at the end** of the rebuttal for clarity.
>
> ---
>
> ### Summary of Comment(s)
> **Weakness 1:** “Only two specific encoders; coarse taxonomy.”
>
> ### Response
> Thank you for raising this important point. We agree that real-world EEG reflects a rich spectrum of cognitive states (e.g., attention, working memory, fatigue), and that these processes can overlap. Our current taxonomy is intentionally coarse, but for reasons grounded in neurophysiological evidence, data availability, and task diversity—rather than a claim that only three brain states exist. We had noted this in the original limitation section, and we now strengthen the explanation.
>
> 1. **Neurophysiological justification:**
> Motor and affective processes are among the most robust and spatially differentiable EEG states (sensorimotor vs. frontal/temporal–limbic). Prior work [a-c] shows that these categories produce the clearest scalp-level differences that can be consistently learned from EEG. Grouping all remaining datasets into “other” avoids forcing noisy or incompatible labels into the decoupling process.
>
> 2. **Practical constraint:**
> Large-scale EEG pre-training datasets are highly heterogeneous, and most publicly available datasets belongs to only a few broad categories (primarily motor, affect, resting/other). More fine-grained cognitive datasets(e.g., attention, memory load) are rare, inconsistent. A finer taxonomy cannot be reliably trained using less samples during pre-training.
>
> 3. **Empirical validation in the revised manuscript:**
> Despite the coarse taxonomy, we observe clear state-aligned improvements:
>
> - On FACED (affect): adding the affect encoder improves ACC-B by **+3.02%**.
> - On BCI-IV-2A (motor): adding the motor encoder improves ACC-B by **+1.47%**.
> - Non-matching encoders (e.g., affect encoder on motor tasks) do not help.
>
> **Table a — FACED (Affective Task)**
> | Encoder Combo | ACC-B (%) | Δ ACC-B | Kappa | Δ Kappa | F1-W (%) | Δ F1-W |
> |---------------|-----------|---------|--------|---------|-----------|---------|
> | S | 56.35 ± 0.31 | — | 0.5088 ± 0.0050 | — | 57.10 ± 0.49 | — |
> | S + A | 59.37 ± 0.87 | +3.02 | 0.5418 ± 0.0092 | +0.0330 | 60.23 ± 0.61 | +3.13 |
> | S + M | 58.28 ± 1.42 | +1.93 | 0.5296 ± 0.0143 | +0.0208 | 59.04 ± 1.44 | +1.94 |
>
> **Table b — BCI-IV-2A (Motor Imagery Task)**
> | Encoder Combo | ACC-B (%) | Δ ACC-B | Kappa | Δ Kappa | F1-W (%) | Δ F1-W |
> |---------------|-----------|---------|--------|---------|-----------|---------|
> | S | 55.27 ± 2.18 | — | 0.4036 ± 0.0290 | — | 55.04 ± 2.36 | — |
> | S + M | 56.74 ± 1.48 | +1.47 | 0.4232 ± 0.0198 | +0.0196 | 56.53 ± 1.69 | +1.49 |
> | S + A | 54.99 ± 0.44 | −0.28 | 0.3999 ± 0.0059 | −0.0037 | 54.69 ± 0.60 | −0.35 |
>
> These results indicate that even coarse state partitioning captures meaningful neural structure.
>
> 4. **The taxonomy does not restrict extensibility:**
> We emphasize in the revised manuscript that the framework is modular: additional encoders can be added if future large-scale datasets provide richer state categories (e.g., attention, memory load, fatigue). Thus, the design is extensible rather than conceptually limited.
>
> ### Modifications in the Revised Manuscript
> - Updated **Section 3.3 “Brain State Decoupling”** to explain that the three-state taxonomy is a practical, neuroscience-supported choice based on reliable labels available in large-scale EEG datasets.
> - Emphasized extensibility of the encoder design.
> - Clarified how state-specific encoders capture meaningful spatial–temporal patterns supported by spatial-filter visualizations (now in **Figure 3**).

---

> ### Author Response · Authors · 2025-11-23
> **[2/6] To reviewer 8DR1(W2)**
>
> ### Summary of Comment(s)
> **Weakness 2:** “Encoder aggregation unclear; why flexible selection?”
>
> ### Response
> Thank you for the thoughtful question. We clarify that encoder aggregation in downstream tasks is performed by **concatenating the outputs of the activated encoders**, followed by a classification head. We agree that the original wording may have overstated the role of flexibility. In the revised manuscript, we clarify that the *primary* benefit of the decoupled design is improved **state-aware representation learning**, while flexible encoder selection is a **secondary practical advantage**, not a core contribution.
>
> Table 4 clarifies that the benefit of parallel encoders is not simply “using all encoders,” but rather that **each added encoder contributes meaningful, state-related information** in a task-dependent way:
>
> **FACED (affective):**
> - **S + A** improves over S by **+3.02%**, showing that the affect encoder captures affect-specific structure.
> - **S + A + M** improves slightly further (+3.53% over S), indicating complementary motor-related information.
> - **S + A + M + O** does *not* yield the best performance → activating all encoders is **not optimal**.
>
> **BCI-IV-2A (motor imagery):**
> - **S + M** improves over S by **+1.47%**, confirming motor-specific representation benefits.
> - **S + A + M** further increases ACC-B to **57.97%**, showing the affect encoder adds small but useful complementary structure.
> - **S + A + M + O** does not perform best → activating all encoders is *not* universally optimal.
>
> These results (Tables c and d in the rebuttal) show that:
> 1. **The optimal encoder combination is task dependent.**
> 2. **Indiscriminately activating all encoders is suboptimal.**
> 3. **Selective activation is meaningful**, not arbitrary.
>
> **Why flexibility remains useful:**
> Activating more encoders increases both **parameter count** and **FLOPs** (Tables c & d). Thus, the ability to activate only the relevant encoders is beneficial for:
> - resource-constrained deployment,
> - smaller devices,
> - real-time applications.
>
> To avoid overstating this aspect, we revised the manuscript so that flexibility is presented as a **practical option**, while the core benefit comes from **state-aware decoupled encoders**.
>
> ### Modifications in the Revised Manuscript
> - Clarified the aggregation mechanism in **Section 4.3** (“We concatenate the outputs of the activated encoders and feed them into an MLP classifier”).
> - Added discussion in **Section 4.8** explaining why “using all encoders” is not optimal.
> - Updated **Section 1 Introduction** to soften the original claim about flexibility and emphasize that the main contribution is the **state-aware decoupled encoder design**.

---

> ### Author Response · Authors · 2025-11-23
> **[3/6] To reviewer 8DR1(W3)**
>
> ### Summary of Comment(s)
> **Weakness 3:** “The Methods section contains unclear notation, errors, and ambiguous mathematical formulation.”
>
> ### Response
> Thank you for pointing out these issues. We carefully reviewed and corrected the entire Methods section to resolve all unclear notation and typographical errors. Specifically:
>
> - The previously undefined symbol **$S_p$** (line 210 in the original draft) has now been defined as the **step size (stride)** for cutting the data into patches.
> - The typographical error **“KTT”** (line 197) was intended to represent the product **$K_T \times T\$**. To avoid confusion, we replaced it with the explicit notation **$(K_T \cdot T\$**.
> - The ambiguity in **Equation (9)** regarding when the positional embedding **$p$** is added (before vs. after flattening) has been resolved. We now **explicitly state** that the positional embedding is added **after flattening** the spatially aligned features.
> - We simplified notation throughout and removed redundant or unused variables to improve readability and consistency.
>
> We also recognize that the original Methods section did not clearly convey how the components of **BrainPro** fit together. In the revised manuscript, we substantially reorganized **Section 3** to improve coherence and readability. We added a new **Section 3.1 – Overview**, which summarizes the entire pipeline, clarifies the high-level flow, and explains how each module connects to the next.
>
> The updated organization is as follows:
>
> #### **1. Retrieval-Based Spatial Learning (Section 3.2)**
> **Organization:**
> - Each dataset is aligned to a universal channel–region template.
> - For every electrode, the model retrieves spatial filter weights from its nearest template locations.
> - These retrieved weights produce spatially aligned channel and region filters.
>
> **Why it helps:**
> - Enforces **location-consistent spatial learning** across heterogeneous montages—addressing a key limitation of self-attention EFMs.
> - Template-tied filters maintain anatomical meaning and enable **interpretable spatial maps**.
>
>
> #### **2. Brain-State Decoupling with Shared & State-Specific Encoders (Section 3.3)**
> **Organization:**
> - Spatially aligned features pass through one shared encoder plus state-specific encoders (affect, motor, other).
> - A **margin-based decoupling loss** encourages state-specific encoders to learn distinct but complementary representations.
>
> **Why it helps:**
> - Reflects neuroscience evidence that brain states share global dynamics but exhibit state-specific spatial variation [a-c].
> - Prevents mixing disparate processes into a single latent space.
>
>
> #### **3. Region-Aware Masked Reconstruction (Section 3.4)**
> **Organization:**
> - Masked reconstruction is **weighted by state-relevant cortical regions** (e.g., frontal/temporal for affect; sensorimotor for motor).
> - Both shared and state-specific encoders contribute to reconstruction.
>
> **Why it helps:**
> - Injects **neurophysiological priors** into pre-training.
> - Strengthens both shared and state-specific pathways by emphasizing state-relevant structure.
>
> ### Modifications in the Revised Manuscript
> - Defined all variables at first usage and removed inconsistent or unused notation.
> - Corrected typographical errors including **$S_p$** and **$K_T * T$**.
> - Clarified Equation (9) with explicit text and simplified embedding notation.
> - Reorganized **Section 3**: added an overview, improved transitions, and moved secondary mathematical details to the appendix.
> - Improved clarity, consistency, and readability across the entire Methods section.

---

> ### Author Response · Authors · 2025-11-23
> **[4/6] To reviewer 8DR1(W4)**
>
> ### Summary of Comment(s)
> **Weakness 4:** “The 60-channel universal template may limit applicability to higher-density EEG.”
>
> ### Response
> Thank you for raising this concern. We clarify that BrainPro uses a pre-defined **60-channel universal template** solely as a *reference coordinate system*, with a corresponding spatial filter bank defined on these 60 anchor positions. When a new dataset is used—whether **32, 64, 128, or 256 channels**—each electrode simply retrieves the nearest template channel(s) and applies the associated spatial filter weights. Thus, BrainPro **remains compatible with higher-density EEG** because each electrode can still map to its nearest template location.
>
> We note, however, that the current template limits the model’s ability to fully exploit all high-density electrodes (e.g., 128 or 256 channels), since spatial filters are defined only for the 60 anchor points. Extending to a high-density template is conceptually straightforward, but currently faces two practical limitations:
>
> 1. **Data imbalance in public EEG corpora:** Most large-scale datasets have far fewer than 128 or 256 channels, meaning a dense template would include many unused anchors receiving no training signal.
> 2. **Lack of high-density large-scale data:** Reliable learning of a 128-/256-channel template would require sufficient coverage from high-density recordings, which is currently scarce.
>
> Thus, we selected a 60-channel template as a **practical and robust compromise**, aligning with the dominant channel configurations in existing public datasets. Importantly, this does not affect BrainPro’s *operational compatibility* with high-density EEG; it only limits the resolution of the template-defined filters. Future versions can adopt a denser template once large-scale high-density data becomes available.
>
> ### Modifications in the Revised Manuscript
> - Added discussion in the **Limitations** section acknowledging that the current 60-channel template does not fully exploit high-density montages.
> - Clarified that extending to 128–256-channel templates is feasible and will be enabled by future large-scale high-density EEG datasets.

---

> ### Author Response · Authors · 2025-11-23
> **[5/6] To reviewer 8DR1(W5)**
>
> ### Summary of Comment(s)
> **Weakness 5:** Concern that (1) the pretraining corpus is dominated by “other” datasets, causing affect/motor encoders to receive fewer updates, and (2) this imbalance stems from an overly coarse affect / motor / other taxonomy.
>
> ### Response
>
> #### **1. Addressing the concern that “other” dominates pretraining**
> We agree that many large-scale public EEG datasets fall under the “other” category, which naturally results in more frequent updates to the shared encoder and the “other” encoder. However, this does **not** cause the affect or motor encoders to be overshadowed or undertrained, for two key reasons:
>
> **• Deterministic, non-competitive update mechanism**
> Unlike sparse MoE or gating-based systems, BrainPro does **not** route samples based on learned competition. Each dataset is *manually and deterministically* assigned to its category encoder.
> - Whenever an affect-related dataset is encountered, the affect encoder is updated.
> - Whenever a motor-related dataset is encountered, the motor encoder is updated.
>
> There is no mechanism by which the “other” encoder can absorb updates that belong to affect or motor. Thus, imbalance in dataset counts does not create “winner-take-all” behavior.
>
> **• Encoders remain distinct and meaningful**
> Despite fewer updates, affect and motor encoders learn **clear, state-aligned spatial patterns**, as shown in Fig. 3:
> - affect encoder → frontal–temporal emphasis
> - motor encoder → sensorimotor emphasis
>
> This demonstrates that the state-specific pathways successfully learn meaningful representations even with fewer samples.
>
> ---
>
> #### **2. Why we use a coarse affect / motor / other taxonomy**
> We appreciate the reviewer’s observation that EEG encompasses many cognitive states. Our taxonomy is intentionally coarse due to **neuroscientific grounding** and **practical considerations of large-scale pretraining**, not due to architectural limitation.
>
> **• Neuroscientific rationale**
> Affect and motor are the two brain states with the **most reproducible, spatially distinct EEG signatures**, making them suitable anchors for defining state-specific encoders:
> - Affect: frontal/temporal–limbic involvement, frontal asymmetry, oscillatory changes
> - Motor: sensorimotor mu/beta rhythms, contralateral desynchronization
>
> These signatures provide stable, biologically meaningful structure for pretraining.
>
> **• Practical evaluation constraint—not label dependence**
> While we do *not* use labels during pretraining, there are relatively **more high-quality affect and motor datasets available for downstream evaluation**, allowing us to thoroughly assess the design. This makes affect and motor reliable categories for demonstrating the effectiveness of BrainPro’s decoupled structure.
>
> **• Taxonomy does not limit the design**
> We fully acknowledge that this coarse taxonomy does not capture the full spectrum of cognitive states. Importantly, BrainPro’s architecture is **modular and extensible**:
> additional encoders can be added in the future when large-scale datasets with richer state annotations become available.
>
> Thus, the current taxonomy reflects a balance between neuroscientific grounding and the practical realities of available data, without restricting the design’s future extensibility.
>
> ### Modifications in the Revised Manuscript
> - Added explanation in **Section 3.3** on why encoder updates remain stable despite corpus imbalance.
> - Clarified the neuroscience-based motivation for choosing affect and motor as explicit encoders.
> - Cited Fig. 3 to show that each encoder learns **distinct, meaningful spatial filters**.
> - Added a statement in the **Limitations** section explaining that the taxonomy is coarse by design but **fully extendable** as richer datasets become available.

---

> ### Author Response · Authors · 2025-11-23
> **[6/6] To reviewer 8DR1(Q1 and References)**
>
> ### Summary of Comment(s)
> **Question 1:** “Lacks full encoder-combination comparisons (S, S+A, S+O, S+M, S+A+O+M).”
>
> ### Response
> Thank you for pointing this out. We agree that the earlier ablation did not include all encoder combinations. In the revised manuscript, we now provide a complete set of encoder configurations—including **S**, **S+A**, **S+O**, **S+M**, and **S+A+O+M**—as reported in the updated **Table 4**, and are presented in Table c and d.
>
> These expanded results show that **performance gains arise from task-relevant state-specific encoders**, rather than from simply activating more modules or increasing model capacity:
>
> - **FACED (affective task):**
>   • **S + A** improves over **S** by **+3.02%**, confirming that affect-specific structure is beneficial.
>   • **S + O** does not help, demonstrating that the improvement is not due to capacity alone.
>
> - **BCI-IV-2A (motor task):**
>   • **S + M** improves over **S** by **+1.47%**, indicating motor-specific benefits.
>   • **S + A** does not help, again supporting state relevance.
>
> Moreover, **S + A + O + M** is *not* always the best-performing configuration, showing that indiscriminate activation of all encoders is suboptimal. This further highlights the utility of **task-adaptive encoder selection**, a key property of our modular design.
>
> ### Modifications in the Revised Manuscript
> - Added the full encoder-combination comparisons as **Table 4**.
> - Updated Section **4.8** to explicitly discuss how these results validate the contribution of modular state-specific encoders and show that gains are not due to increased parameter count alone.
>
> ### References
> [a] Abigail S. Greene et al., *Why is everyone talking about brain state?* Trends in Neurosciences, 2023.
> [b] Alfons Schnitzler et al., *Involvement of primary motor cortex in motor imagery*, NeuroImage, 1997.
> [c] Vesa Putkinen et al., *Decoding music-evoked emotions*, Cerebral Cortex, 2020.
> [d] Soraia M. Alarcão & Manuel J. Fonseca, *Emotions recognition using EEG signals*, IEEE TAC, 2019.
> [e] G. Pfurtscheller et al., *Mu rhythm (de)synchronization and EEG classification*, NeuroImage, 2006.

---

> > ### Comment · Reviewer_8DR1 · 2025-11-24
> >
> > Thank you for your detailed response and I appreciate the thorough revisions. While some of my initial concerns have been addressed, several key issues remain unresolved:
> >
> > 1. Regarding the claim that "Prior work [a–c] shows that these categories produce the clearest scalp-level differences that can be consistently learned from EEG," I find that the provided references do not explicitly support this argument. The analyses in these studies appear to rely primarily on fMRI and other neuroimaging data, rather than EEG-specific evidence. If I have overlooked relevant sections, please point out where in these references such EEG-based conclusions are drawn?
> >
> > 2. On the question of how to determine which encoders to activate for different tasks: Table 4 shows that the S+A+M configuration achieves the best performance on both the FACED and BCIC-IV-2a datasets, outperforming simpler combinations like S+A or S+M. However, in practice, one cannot exhaustively evaluate all encoder combinations to decide which to activate. Does this imply that activating all encoders (S+A+M) is the most straightforward—if not the only feasible—strategy? The authors suggest that the superiority of S+A+M stems from complementary information across encoders. If so, I am curious:
> >
> >  >- Would S+A+O outperform S+A on FACED? This ablation is not provided.
> >  >- Similarly, would S+M+O outperform S+M?
> >  >- Would such combinatorial advantages generalize to datasets beyond emotion or motor tasks? For instance, might performance follow a pattern like: S < S+O+A+M < S+O < S+O+(A/M)?
> >
> > 3. The authors claim that BrainPro allows for the easy addition of specialized encoders for different brain states. However, this seems contradictory to the fundamental concept of a foundation model. Not all users possess the computational resources required to design specialized encoders and potentially re-pretrain the backbone from scratch. Rather than demonstrating flexibility, this requirement seems to highlight a significant limitation regarding the method's accessibility and practical extensibility.

---

> > > ### Author Response · Authors · 2025-11-26
> > > **[1/2] To reviewer 8DR1 [Additional issues]**
> > >
> > > ### Summary of Comment(s)
> > > **Issue 1**:The reviewer notes that the cited prior work does not show EEG-specific evidence supporting the claim that affect and motor states yield the clearest scalp-level differences.
> > >
> > > ### Response
> > > We appreciate the reviewer’s careful reading. The cited work [a] indeed emphasizes fMRI-based evidence but also explicitly states that *“scalp electroencephalography (EEG), like LFPs, captures brain state–associated differences in power across frequency bands, but suffers from imprecise source localization.”* This statement aligns with our position: EEG reflects **meaningful but coarse** spatial differences between brain states. Our claim does not rely on fine-grained cortical localization but on **scalp-level topographic patterns**, which are well established in EEG literature.: frontal, temporal, parietal for emotion [d], central region (around C3, C4, and Cz EEG channels) [e]. These findings provide the EEG-based justification.
> > > We have indicated this in the revised manuscript in the Section I “Although EEG captures these patterns only at a coarse spatial scale, it nonetheless reflects meaningful state-associated differences in spectral power and scalp distributions”
> > >
> > > ---
> > >
> > > ### Summary of Comment(s)
> > > **Issue 2**: How users can decide which encoders to activate, since S+A+M performs best but exhaustive testing is impractical.
> > >
> > > ### Response
> > > Thank you for this important question. We clarify that **S+A+M is not the only feasible strategy**, nor do users need to perform exhaustive search. The key principle is:
> > >
> > > > **Activate the shared encoder (S) plus the encoder(s) aligned with the dataset’s dominant brain state.**
> > >
> > > This rule is both practical and supported by all three datasets:
> > >
> > > 1. **Affective task (FACED):**
> > >    Best performance is obtained with **S + A** or **S + A + M**, not arbitrary combinations.
> > >
> > > 2. **Motor task (BCI-IV-2A):**
> > >    Best performance is obtained with **S + M** or **S + A + M**, consistent with motor-dominant patterns.
> > >
> > > 3. **Attention task:**
> > >    The best performance is achieved with **S + A** or **S + O** , showing that *task-aligned encoders complement S*, and that **S+A+M is not needed** for this dataset.
> > >
> > > The Attention dataset is especially informative: although it is not an emotion or motor task, **S + A** outperforms S and S + M, confirming that the benefit of an encoder is **task-dependent**, not the result of merely enabling all modules.
> > >
> > > Based on these results, BrainPro provides three effective, non-exhaustive configurations:
> > >
> > > - **BrainPro-A:** S + A (emotion / social cognition / attention-dominant tasks)
> > > - **BrainPro-M:** S + M (motor imagery / movement intention tasks)
> > > - **BrainPro-AM:** S + A + M (mixed or uncertain tasks)
> > >
> > > This removes the need for users to test all combinations. Importantly, **S+A+M is only recommended for tasks involving mixed or unknown brain states**, not as the universal default.
> > >
> > > In summary, the results do **not** imply that “activating all encoders” is the only feasible strategy. Instead, they support a **principled, task-aligned activation rule** that avoids exhaustive search and maintains performance.
> > >
> > > ---
> > >
> > > ### Supporting Results (from Table 4)
> > >
> > >
> > > ### Table c — FACED Dataset (Affective Task)
> > >
> > > | Method        | ACC           |
> > > |---------------|---------------|
> > > | S             | 0.5635 ± 0.0032 |
> > > | S + O         | 0.5821 ± 0.0144 |
> > > | S + A         | **0.5937 ± 0.0087** |
> > > | S + M         | 0.5828 ± 0.0142 |
> > > | S + A + O     | 0.5929 ± 0.0177 |
> > > | S + M + O     | 0.5875 ± 0.0150 |
> > > | **S + A + M** | **0.5988 ± 0.0057** |
> > > | S + A + M + O | 0.5824 ± 0.0050 |
> > >
> > > ### Table d — BCI-IV-2A Dataset (Motor Imagery Task)
> > >
> > > | Method        | ACC           |
> > > |---------------|---------------|
> > > | S             | 0.5527 ± 0.0218 |
> > > | S + O         | 0.5525 ± 0.0173 |
> > > | S + A         | 0.5499 ± 0.0044 |
> > > | S + M         | **0.5674 ± 0.0148** |
> > > | S + A + O     | 0.5642 ± 0.0033 |
> > > | S + M + O     | 0.5781 ± 0.0156 |
> > > | **S + A + M** | **0.5797 ± 0.0229** |
> > > | S + A + M + O | 0.5660 ± 0.0067 |
> > >
> > > ### Table e — Attention Dataset
> > >
> > > | Method        | ACC           |
> > > |---------------|---------------|
> > > | S             | 0.6907 ± 0.0113 |
> > > | S + O         | 0.7142 ± 0.0232 |
> > > | S + A         | **0.7222 ± 0.0291** |
> > > | S + M         | 0.6982 ± 0.0433 |
> > > | S + A + O     | 0.7006 ± 0.0057 |
> > > | S + M + O     | 0.6821 ± 0.0408 |
> > > | S + A + M     | 0.7019 ± 0.0218 |
> > > | S + A + M + O | 0.7068 ± 0.0178 |
> > >
> > > These tables show:
> > > - Task-aligned single-encoder additions (**S+A**, **S+M**) already give strong gains.
> > > - **S+A+M** is beneficial only when multiple states may contribute (e.g., FACED, MI).
> > > - No dataset requires “all encoders activated” for best performance.
> > >
> > > ---

---

> > > ### Author Response · Authors · 2025-11-26
> > > **[2/2] To reviewer 8DR1 [Additional issues]**
> > >
> > > ### Summary of Comment(s)
> > > **Issue 2.1 & 2.2**: Whether S+A+O outperforms S+A on FACED, and whether S+M+O outperforms S+M.
> > >
> > > ### Response
> > >
> > > #### (a) **Does S+A+O outperform S+A on FACED?**
> > > | Method | ACC |
> > > |--------|------|
> > > | **S + A** | **0.5937** |
> > > | S + A + O | 0.5929 |
> > >
> > > **No. S+A+O does not outperform S+A.**
> > >
> > > #### (b) **Does S+M+O outperform S+M on FACED?**
> > > | Method | ACC |
> > > |--------|------|
> > > | S + M | 0.5828 |
> > > | S + M + O | **0.5875** |
> > >
> > > Small increase (~0.5%), within standard deviation.
> > >
> > > #### (c) **Does S+M+O outperform S+M on BCI-IV-2A?**
> > > | Method | ACC |
> > > |--------|------|
> > > | S + M | 0.5674 |
> > > | S + M + O | **0.5781** |
> > >
> > > Yes, small increase (~1%), also within std. dev.
> > >
> > > ### Conclusion
> > > The O encoder provides **minor, non-robust complementary information** and is not useful for affective or motor tasks.
> > >
> > > ---
> > >
> > >
> > > ### Summary of Comment(s)
> > > **Issue 2.3**: The reviewer asks whether combinatorial encoder benefits generalize to tasks beyond emotion or motor, and whether a monotonic pattern such as
> > > S < S+O+A+M < S+O < S+O+(A/M)
> > > might appear on other datasets.
> > >
> > > ### Response
> > > Thank you for the question. Our experiments on the **Attention dataset**, which is neither an emotion task nor a motor task, directly show that encoder performance does **not** follow the suggested monotonic pattern. Instead, the same principle observed in FACED and BCI-IV-2A still holds:
> > >
> > > > **Performance improves when the shared encoder (S) is combined with the task-relevant encoder, not by activating more encoders.**
> > >
> > > #### Evidence from the Attention dataset
> > > The results are shown in Table e. and **key observation:**
> > > Only **S + A** yields the best performance.
> > > Adding more encoders (O or M) actually reduces performance.
> > > Thus, combinatorial stacking does not generalize beyond emotion and motor tasks.
> > >
> > > #### Conclusion
> > > Across all three datasets (affect, motor imagery, attention), the same practical rule holds:
> > >
> > > > **Use the Shared encoder (S) plus the encoder aligned with the dataset's dominant cognitive state.**
> > > > i.e., **S + A**, **S + M**, or **S + A + M** for mixed/uncertain tasks.
> > >
> > > ---
> > > ### Summary of Comment(s)
> > > **Issue 3:** The reviewer worries that “adding specialized encoders” contradicts the idea of a foundation model and may require users to design or retrain modules, reducing accessibility.
> > >
> > > ---
> > >
> > > ### Response
> > > Thank you for highlighting this. We clarify that **BrainPro does not require any user to add or retrain encoders**. The current pretrained version—consisting of **S, A, M, and O**—can already be used. It is directly applicable to **diverse downstream tasks** (affective, motor imagery, attention etc.) and already achieves **state-of-the-art performance** without any customization.
> > >
> > > When we mentioned that new encoders “can be added,” we meant this allow **optional research extensions**.
> > >
> > > Importantly, the **main purpose** of this work is to demonstrate that **brain state–aware representation learning**—achieved through shared + state-specific encoders—substantially improves downstream EEG performance. This contribution is realized in the current pretrained model.
> > >
> > > Thus, extensibility is a **research advantage**, not a practical barrier. For typical users, BrainPro remains **accessible**, and effective out-of-the-box.
> > >
> > > ### References
> > > [a] Abigail S. Greene et al., *Why is everyone talking about brain state?* Trends in Neurosciences, 2023.
> > > [b] Alfons Schnitzler et al., *Involvement of primary motor cortex in motor imagery*, NeuroImage, 1997.
> > > [c] Vesa Putkinen et al., *Decoding music-evoked emotions*, Cerebral Cortex, 2020.
> > > [d] Soraia M. Alarcão & Manuel J. Fonseca, *Emotions recognition using EEG signals*, IEEE TAC, 2019.
> > > [e] G. Pfurtscheller et al., *Mu rhythm (de)synchronization and EEG classification*, NeuroImage, 2006.

---

> > > > ### Comment · Reviewer_8DR1 · 2025-11-28
> > > >
> > > > The authors’ latest response addressed my remaining concern. For downstream tasks, using an encoder focused on activated brain regions seems like an intuitive and practical strategy, and the additional results support this claim. Thanks for the detailed reply—I’ll update my score to reflect this positive revision.

---

> > > > > ### Author Response · Authors · 2025-11-28
> > > > >
> > > > > Thank you very much for your thoughtful follow-up and for taking the time to reevaluate our revision. We truly appreciate your positive assessment and are glad to hear that the additional experiments and clarifications addressed your remaining concerns. Your feedback has been extremely helpful in strengthening the paper, and we are grateful for your constructive comments throughout the review process.

---

### Official Review · Reviewer_8DYM · 2025-10-23

**Soundness:** 2
**Presentation:** 2
**Contribution:** 2
**Rating:** 2
**Confidence:** 4

**Summary:**

The authors identify three fundamental limitations in existing EEG foundation models (EFMs). First, current EFMs underutilize spatial interactions among electrodes and brain regions. Second, they pretrain a single encoder without explicitly disentangling brain state–related representations. Third, the use of a single shared encoder limits the flexibility of downstream adaptation. To address these limitations, the authors propose **BrainPro**, a novel framework designed to enhance spatial representation and brain state modeling. Specifically, BrainPro introduces a *retrieval-based spatial learning* mechanism to overcome the first limitation, employs *parallel encoders* with *decoupling* and *region-aware reconstruction objectives* to address the second, and integrates a *shared encoder* with one or more *brain-state-specific encoders* to resolve the third.

**Strengths:**

The topic of EEG foundation models (EFMs) is important and highly relevant to the advancement of brain–computer interfaces (BCIs).

**Weaknesses:**

1.The *Methods* section lacks coherent logical flow. While the authors describe the individual components of the proposed framework in detail, they provide insufficient explanation of how these components are organized or why such an organization is effective.

2.The three claimed innovations which should be the core of this paper are not supported by adequate theoretical justification or experimental validation.

**These two issues form the primary basis for my rejection decision.**

3.The motivation presented in the *Introduction* is also confusing. The first paragraph lists numerous challenges, many of which are not directly related to the objectives of this work. Removing or refining these unrelated points would help clarify and strengthen the paper’s motivation.

4.The design of multiple brain-state-specific encoders appears to involve three categories—affect, motor, and other. However, the rationale for this categorization is not explained, nor are the details provided regarding how the encoders specifically capture affective and motor-related representations.

5.In Section 2.5, the authors state that “for a downstream task, any subset SSS of encoders can be activated and concatenated.” However, the method for determining or selecting subset SSS is not described, leaving a key implementation detail unclear.

**Questions:**

1.Selective updates are applied during encoder training, which appears conceptually similar to the sparse Mixture-of-Experts (MoE) mechanism. Does this approach introduce a potential *winner-takes-all* problem?

2.In the *Introduction*, the authors claim that BrainPro possesses strong scalability, interpretability, and generalization capabilities. However, no theoretical analysis or experimental evidence is provided to substantiate these claims. The paper would benefit from including relevant justification or empirical validation.

3.The authors also state that existing models often fail to explicitly capture channel-to-channel and region-to-region interactions. It would be valuable to provide theoretical reasoning or empirical results supporting this assertion.

4.Similarly, the claim that current models rarely learn state-aware representations during self-supervised pre-training lacks supporting evidence. The authors are encouraged to include either theoretical discussion or experimental analysis to validate this point.

---

> ### Author Response · Authors · 2025-11-23
> **[1/6] To reviewer 8DYM (W1)**
>
> We thank the reviewer for the constructive feedback. Before presenting the **Summary of Comment(s)**, **Response**, and **Modifications in the Revised Manuscript** for each issue, we clarify the organization of our rebuttal:
>
> • **Three-part structure for every comment**
>   – **Summary of Comment(s):** brief restatement of the reviewer’s point.
>   – **Response:** our explanation, justification, and supporting evidence.
>   – **Modifications in the Revised Manuscript:** exact changes made and where they appear.
>
> • **Table/Figure indexing convention**
>   – In the **rebuttal**, tables and figures use **letter indices** (e.g., *Table a*, *Fig. b*) to avoid confusion and keep the document compact.
>   – In the **revised manuscript**, tables and figures use **numbered indices** (e.g., *Table 3*, *Figure 4*).
>   – Each table/figure appears **only once** in the rebuttal; subsequent responses referring to the same analysis use the same letter index.
>
> • **References**
>   – All cited references used in the responses are collected **once at the end** of the rebuttal for clarity.
>
> ---
>
> ### Summary of Comment(s)
> **Weakness 1:** “Methods lack coherent flow.”
>
> ### Response
> Thank you for pointing this out. We agree that the original Methods section did not clearly convey how the components of BrainPro fit together. In the revised manuscript, we substantially reorganized Section 3 to improve coherence and readability. We added a new overview subsection at the beginning of Section 3 that summarizes the entire pipeline, clarifies the high-level flow, and explains how each module connects to the next.
>
> 1. **Retrieval-Based Spatial Learning (Section 3.2)**
> **Organization:**
>
> -Each dataset is aligned to a universal channel–region template.
> -For every electrode, the model retrieves the spatial filter weights of its nearest template locations.
> -These retrieved weights produce spatially aligned channel and region filters.
>
> **Why it helps:**
>
> -Enforces location-consistent spatial learning across heterogeneous montages, addressing a key limitation of self-attention EFMs.
> -Template-tied filters preserve anatomical meaning and enable interpretable spatial maps.
>
> 2. **Brain-State Decoupling with Shared & State-Specific Encoders (Section 3.3)**
> **Organization:**
>
> -Spatially aligned features are processed by one shared encoder plus state-specific encoders (affect, motor, other).
> -A margin-based decoupling loss encourages state-specific encoders to learn distinct but complementary representations.
>
> **Why it helps:**
> -Reflects the neuroscience observation that brain states combine shared global dynamics with state-dependent spatial variations.
> -Explicitly modeling these components avoids mixing disparate processes into a single latent space.
>
> 3. **Region-Aware Masked Reconstruction (Section 3.4)**
> **Organization:**
>
> -Masked reconstruction is weighted by state-relevant cortical regions (e.g., frontal/temporal for affect; sensorimotor for motor).
> -Both shared and state-specific representations contribute to reconstruction.
>
> **Why it helps:**
>
> -Injects neurophysiological priors into pre-training, guiding the model to focus on regions meaningful for each state.
> -Strengthens the separation between shared and state-specific components.
>
> ### Modifications in the Revised Manuscript
> - Reorganized **Section 3 “Methodology”**
> - Added a new **Section 3.1 “Overview of BrainPro Framework”** explaining the full workflow
> - Improved transitions between modules
> - Simplified mathematical expressions
> - Moved redundant derivations to the appendix for clarity

---

> ### Author Response · Authors · 2025-11-23
> **[2/6] To reviewer 8DYM (W2 first half)**
>
> ### Summary of Comment(s)
> **Weakness 2:** “Innovations lack justification or validation.”
>
> ### Response
> We appreciate the reviewer’s concern and have strengthened both the theoretical grounding and empirical support in the revised manuscript. Below we summarize the justification and corresponding experimental evidence for each innovations.
>
> ## 1. Retrieval-Based Spatial Learning
>
> **Theoretical justification:**
> EEG channels correspond to fixed scalp locations, and spatial patterns (e.g., frontal–limbic for affect, central sensorimotor for motor) carry essential neurophysiological meaning [a–c]. Existing EFMs treat channels as permutation-invariant tokens, weaken this anatomical structure.
> Our retrieval module solves this by retrieving spatial filter weights from template-aligned channel and region filter banks, enforcing location-consistent spatial learning across heterogeneous montages. Theoretical justification is provided in Appendix D.
>
> **Experimental evidence:**
> • Replacing structured retrieval with random retrieval causes clear drops in performance (Table a).
> • BrainPro shows the **smallest performance degradation** under missing-channel conditions (Figure 4), indicating robust spatial representation learning (core results in Table b; full in Appendix Table 12).
> • Spatial filters (Figure 3) exhibit meaningful neuroanatomical topographies—frontal/temporal emphasis for affect, sensorimotor for motor.
>
> ### Table a — Random Retrieval vs BrainPro
>
> | Method               | ACC-B (BCI)        | Kappa (BCI)       | F1-W (BCI)        | ACC-B (MA)         | AUC-PR (MA)        | AUROC (MA)         |
> |----------------------|--------------------|--------------------|--------------------|--------------------|--------------------|--------------------|
> | w/ Random Retrieval | 0.5125 ± 0.0308   | 0.3500 ± 0.0410   | 0.5052 ± 0.0379   | 0.7733 ± 0.0410   | 0.8879 ± 0.0288   | 0.8606 ± 0.0343   |
> | BrainPro            | 0.5674 ± 0.0148   | 0.4232 ± 0.0198   | 0.5653 ± 0.0169   | 0.8083 ± 0.0156   | 0.8980 ± 0.0052   | 0.8512 ± 0.0083   |
>
> ### Table b — Robustness Under Random Channel Dropping (FACED)
>
> | Drop Rate | BrainPro | Δ↓     | CBraMod | Δ↓     | LaBraM | Δ↓     |
> |-----------|----------|---------|---------|---------|--------|---------|
> | 0.0       | 0.5937   | 0       | 0.5669  | 0       | 0.5224 | 0       |
> | 0.1       | 0.5747   | −0.0190 | 0.5218  | −0.0451 | 0.4972 | −0.0252 |
> | 0.2       | 0.5373   | −0.0564 | 0.4890  | −0.0779 | 0.4725 | −0.0499 |
> | 0.3       | 0.4995   | −0.0942 | 0.4473  | −0.1196 | 0.4230 | −0.0994 |
> | 0.4       | 0.4639   | −0.1298 | 0.4111  | −0.1558 | 0.3851 | −0.1373 |
> | 0.5       | 0.4270   | −0.1667 | 0.3671  | −0.1998 | 0.3397 | −0.1827 |
>
>
> ## 2. Brain-State Decoupling with Shared + State-Specific Encoders
>
> **Literature justification:**
> Distinct brain states share global EEG structure but also express state-specific spatial–temporal signatures [a–c]. Mixing them in a single latent obscures state-specific variations.
> Our decoupling module explicitly separates shared and state-specific components using a margin-based cosine loss.
>
> **Experimental evidence:**
> • Removing decoupling significantly reduces downstream performance (Table c).
> • Encoder-combination experiments (Tables d and e) show complementary benefits:
>   – **S+A** best on FACED (affect)
>   – **S+M** best on BCI-IV-2A (motor imagery)
>
> ### Table c — w/o Decoupling vs BrainPro
>
> | Method           | ACC-B (BCI)       | Kappa (BCI)       | F1-W (BCI)        | ACC-B (MA)         | AUC-PR (MA)        | AUROC (MA)         |
> |------------------|--------------------|--------------------|--------------------|--------------------|--------------------|--------------------|
> | w/o Decoupling  | 0.4870 ± 0.0184   | 0.3160 ± 0.0246   | 0.4725 ± 0.0219   | 0.7317 ± 0.0696   | 0.9102 ± 0.0356   | 0.8860 ± 0.0408   |
> | BrainPro         | 0.5674 ± 0.0148   | 0.4232 ± 0.0198   | 0.5653 ± 0.0169   | 0.8083 ± 0.0156   | 0.8980 ± 0.0052   | 0.8512 ± 0.0083   |
>
> ### Table d — FACED (Affective Task)
>
> | Encoder Combo | ACC-B (%)         | Δ ACC-B | Kappa              | Δ Kappa | F1-W (%)         | Δ F1-W |
> |---------------|--------------------|---------|---------------------|---------|-------------------|---------|
> | S             | 56.35 ± 0.31      | —       | 0.5088 ± 0.0050    | —       | 57.10 ± 0.49     | —       |
> | S + A         | 59.37 ± 0.87      | +3.02   | 0.5418 ± 0.0092    | +0.0330 | 60.23 ± 0.61     | +3.13   |
> | S + M         | 58.28 ± 1.42      | +1.93   | 0.5296 ± 0.0143    | +0.0208 | 59.04 ± 1.44     | +1.94   |

---

> ### Author Response · Authors · 2025-11-23
> **[3/6] To reviewer 8DYM (W2 last half and W3)**
>
> ### Table e — BCI-IV-2A (Motor Imagery)
>
> | Encoder Combo | ACC-B (%)         | Δ ACC-B | Kappa              | Δ Kappa | F1-W (%)         | Δ F1-W |
> |---------------|--------------------|---------|---------------------|---------|-------------------|---------|
> | S             | 55.27 ± 2.18      | —       | 0.4036 ± 0.0290    | —       | 55.04 ± 2.36     | —       |
> | S + M         | 56.74 ± 1.48      | +1.47   | 0.4232 ± 0.0198    | +0.0196 | 56.53 ± 1.69     | +1.49   |
> | S + A         | 54.99 ± 0.44      | −0.28   | 0.3999 ± 0.0059    | −0.0037 | 54.69 ± 0.60     | −0.35   |
>
> ## 3. Region-Aware Masked Reconstruction
>
> **Literature justification:**
> Different cognitive/affective/motor processes modulate specific regions[d,e]. Weighting the reconstruction loss by region relevance guides the model to focus on state-relevant structure instead of uniformly reconstructing all channels.
>
> **Experimental evidence:**
> • Removing region-aware reconstruction causes strong performance degradation, most notably on BCI-IV-2A.
>
> ### Table f — BCI-IV-2A (4-Class)
>
> | Method             | ACC-B (%)           | Kappa                | F1-W (%)           |
> |--------------------|----------------------|-----------------------|---------------------|
> | w/o reconstruction | 43.54 ± 3.29        | 0.2472 ± 0.0439      | 37.58 ± 5.48       |
> | BrainPro           | 56.74 ± 1.48        | 0.4232 ± 0.0198      | 56.53 ± 1.69       |
>
> ### Table g — Mental Arithmetic (2-Class)
>
> | Method             | ACC-B (%)          | AUC-PR               | AUROC               |
> |--------------------|---------------------|------------------------|-----------------------|
> | w/o reconstruction | 70.83 ± 9.03       | 0.9147 ± 0.0209       | 0.8976 ± 0.0273       |
> | BrainPro           | 80.83 ± 1.56       | 0.8980 ± 0.0052       | 0.8512 ± 0.0083       |
>
> ### Modifications in the Revised Manuscript
> - Added explicit theoretical justification in **Section 2 “Preliminaries”** and **Appendix D**
> - Added retrieval-vs-random results, decoupling ablations, and region-aware ablations in **Section 4 “Experiments”**
> - Added spatial-filter interpretation in **Section 4.4**
> - Added channel-drop robustness in **Section 4.7**
> - Added encoder-combination analysis in **Section 4.8**
>
> ---
> ### Summary of Comment(s)
> **Weakness 3:** “Motivation confusing; too many unrelated points.”
>
> ### Response
> Thank you for this helpful observation. We agree that the original Introduction included several high-level EEG challenges that were not directly tied to our contributions, which made the motivation harder to follow. In the revised manuscript, we fully rewrote the Introduction to focus only on the issues that BrainPro explicitly addresses:
>
> (1) the need for spatially consistent representations across heterogeneous montages,
> (2) the neurophysiological basis for shared vs. state-specific components in EEG, and
> (3) the difficulty existing EFMs have in learning these structures.
>
> All unrelated or tangential points from the original paragraph have been removed.
>
> ### Modifications in the Revised Manuscript
> - Replaced the original introductory paragraph with a streamlined neuroscience-grounded rationale in **Section 1 “Introduction”** (pages 1–2 of the revision)
> - The rewritten Introduction now directly motivates:
>   • spatial consistency across heterogeneous montages
>   • brain-state–aware representation learning
>   • limitations of existing EFMs

---

> ### Author Response · Authors · 2025-11-23
> **[4/6] To reviewer 8DYM (W4 and W5)**
>
> ### Summary of Comment(s)
> **Weakness 4:** “Taxonomy rationale unclear.”
>
> ### Response
> Thank you for raising this point. In the revised manuscript, we now provide a clear rationale for the three-state taxonomy. We select affect and motor because these are among the few EEG brain states with well-established and reliably distinguishable neurophysiological signatures across large public datasets—e.g.,
>
> • **Affect:** frontal and temporal/limbic involvement, asymmetric frontal activity, and emotion-related oscillatory changes [d];
> • **Motor:** sensorimotor rhythms (mu, beta), contralateral desynchronization during motor imagery/execution[b,e].
>
> These states also have consistent annotations in large-scale datasets, enabling stable pre-training. All remaining datasets lack consistent brain-state labels and are grouped as “other” to avoid forcing heterogeneous labels into the decoupling process. This choice is thus both practical and neuroscientifically grounded.
>
> We also clarify how the encoders learn these patterns: because spatial filters are tied to anatomical locations and decoupling encourages divergence from the shared encoder, the affect and motor encoders learn state-specific spatial–temporal features. This is directly reflected in the spatial-filter visualizations (Figure 3, moved to the main paper) showing frontal/temporal emphasis for affect and sensorimotor emphasis for motor, consistent with expected neurophysiology.
>
> **Empirical validation in the revised manuscript:**
> Despite the coarse taxonomy, we observe clear state-aligned improvements (Table d and e):
>
> • On FACED (affect): adding the affect encoder improves ACC-B by **+3.02%**.
> • On BCI-IV-2A (motor): adding the motor encoder improves ACC-B by **+1.47%**.
> • Non-matching encoders (e.g., affect encoder on motor tasks) do not help.
>
> This indicates that such design captures meaningful neural structure.
>
> ### Modifications in the Revised Manuscript
> - Added a dedicated explanation in **Section 3.3 “Brain State Decoupling”** clarifying the rationale for the three-state taxonomy.
> - Added neuroscientific references supporting the chosen categories.
> - Moved spatial-filter visualizations to the main text as **Figure 3** and clarified how they demonstrate state-specific encoder behavior.
> ---
> ### Summary of Comment(s)
> **Weakness 5**: “The method for determining the subset of encoders activated during downstream fine-tuning is unclear”
>
> ### Response
> Thank you for highlighting this. We now clarify that the subset **S** is manually specified based on the known brain-state category of each downstream dataset. Because public EEG benchmarks clearly indicate whether a task is affective (emotion), motor-related (motor imagery), or neither, we activate:
>
> - **Shared encoder + Affect encoder** for emotion datasets (e.g., *FACED*),
> - **Shared encoder + Motor encoder** for motor imagery datasets (e.g., *BCI-IV-2A*),
> - **Shared encoder + Other encoder** for tasks without clear affect/motor labeling.
>
> Only the encoders in **S** are fine-tuned; the remaining state-specific encoders remain frozen. This ensures that fine-tuning uses the brain-state pathway most relevant to the dataset, avoids interference from unrelated encoders, and maintains the intended decoupling structure.
>
> ### Modification
> We updated the downstream training description in **Section 4.3** to explicitly state that:
> > “For each dataset, we manually activate the shared encoder and the corresponding state-specific encoder(s) based on its labeled brain-state category. Only these activated encoders are fine-tuned.”

---

> ### Author Response · Authors · 2025-11-23
> **[5/6] To reviewer 8DYM (Q1, Q2, and Q3)**
>
> ### Summary of Comment(s)
> **Question 1:** “Does selective encoder updating create a winner-takes-all problem similar to sparse MoE?”
>
> ### Response
> Thank you for the question. In our framework, selective updates do **not** create a winner-takes-all effect because encoders are not competing for routing or gating as in sparse MoE. Each batch has a fixed, non-competitive assignment to its corresponding state encoder (affect, motor, or other), and every encoder receives regular updates from all datasets that match its category. There is no gating network, no learned routing, and no mechanism that favors one encoder over another. Empirically, all encoders remain active and contribute meaningful spatial patterns (visualized in Fig. 3), confirming that no collapse occurs.
>
> ### Modifications in the Revised Manuscript
> - Clarified this behavior in **Section 4.3 Downstream Training**.
> - Explicitly stated that state-specific encoders are only updated for their corresponding datasets, avoiding MoE-like competition.
>
> ---
> ### Summary of Comment(s)
> **Question 2:** “Claims of scalability, interpretability, and generalization lack justification or evidence.”
>
> ### Response
> Thank you for pointing this out. We agree that the original Introduction overstated these points. In the revised manuscript, we removed the unsupported claims and now provide concrete justification and empirical evidence:
>
> - **Generalization:** Demonstrated by higher decoding accuracy across diverse tasks and improved robustness under missing-channel perturbations (Fig. 4; Table b), indicating more stable spatial representations.
> - **Interpretability:** Added spatial-filter visualizations (now Fig. 3), showing that each encoder learns neurophysiologically meaningful and state-consistent spatial patterns (e.g., frontal–temporal for affect, sensorimotor for motor).
> - **Scalability:** Clarified that scalability refers to the ability to handle heterogeneous montages through retrieval-based spatial alignment and the extendable encoder architecture that can incorporate additional brain states when new annotations are available.
>
> These aspects are now grounded in explicit theoretical explanations and empirical validation.
>
> ### Modifications in the Revised Manuscript
> - Removed unsupported claims from the **Introduction**.
> - Added explicit justification in **Sections 3.2–3.4**.
> - Moved interpretability and robustness results to the main text as **Fig. 3** and **Fig. 4**.
>
> ---
> ### Summary of Comment(s)
> **Question 3:** “Claim that existing models fail to capture channel-to-channel and region-to-region interactions lacks justification.”
>
> ### Response
> Thank you for pointing this out. In the revised manuscript, we now provide both theoretical grounding and empirical evidence.
>
> **Theoretical reasoning:**
> Existing EFMs rely primarily on self-attention over channel tokens. Self-attention is permutation-invariant and does not encode anatomical structure unless explicitly imposed. As a result, these models cannot guarantee
> 1. consistent channel-to-channel relationships across datasets with different montages, or
> 2. region-to-region consistency, since attention weights do not inherently reflect anatomical proximity.
>
> This issue is detailed in Appendix B of the revised manuscript.
>
> **Empirical evidence:**
> 1. **Channel-drop robustness (Fig. 4):**
> Token-only EFMs (e.g., CBraMod) suffer significantly larger accuracy degradation when channels are missing or perturbed, demonstrating weak spatial dependence modeling. BrainPro remains substantially more stable (Table b).
> 2. **Spatial-filter interpretability (Fig. 3):**
> BrainPro learns clear and neurophysiologically meaningful spatial filters tied to fixed anatomical positions, whereas attention-based EFMs cannot produce spatially interpretable maps due to lack of explicit spatial anchoring.
>
> These results directly support the claim that existing EFMs fail to explicitly model channel- and region-level interactions, and that BrainPro’s retrieval-based spatial learning resolves this limitation.
>
> ### Modifications in the Revised Manuscript
> - Added a theoretical explanation in **Appendix D** and referenced it in **Section 3.2**.
> - Added empirical evidence via **Fig. 3** (spatial filters) and **Fig. 4** (robustness).

---

> ### Author Response · Authors · 2025-11-23
> **[6/6] To reviewer 8DYM (Q4 and References)**
>
> ### Summary of Comment(s)
> **Question 4:** “The claim that current models rarely learn state-aware representations during self-supervised pre-training lacks supporting evidence.”
>
> ### Response
> Thank you for raising this point. We strengthened the manuscript by adding theoretical discussion showing why existing EFMs do not learn state-aware representations during pre-training.
>
> **Theoretical reasoning (added in Introduction & Sec. 3.3):**
> Current EFMs are trained with general self-supervised objectives such as masked reconstruction. These objectives aim to model generic EEG dynamics and do **not** include any mechanism to:
>
> 1. separate **shared** from **state-specific** components,
> 2. enforce **state-related spatial structure**, or
> 3. encourage **distinct representations** for different underlying brain processes.
>
> As a result, variations such as affect, motor processes, cognitive conditions, and artifacts become entangled in a single latent space. Without architectural or objective-level constraints, **state-awareness does not naturally emerge during pre-training**.
>
> ### Modifications in the Revised Manuscript
> - Added a theoretical explanation in the new **Motivation** subsection.
> - Expanded **Section 3.3** with justification for state decoupling and an explicit explanation of why existing EFMs cannot learn state-aware representations during SSL.
>
> ### References
> [a] Abigail S. Greene et al., *Why is everyone talking about brain state?* Trends in Neurosciences, 2023.
> [b] Alfons Schnitzler et al., *Involvement of primary motor cortex in motor imagery*, NeuroImage, 1997.
> [c] Vesa Putkinen et al., *Decoding music-evoked emotions*, Cerebral Cortex, 2020.
> [d] Soraia M. Alarcão & Manuel J. Fonseca, *Emotions recognition using EEG signals*, IEEE TAC, 2019.
> [e] G. Pfurtscheller et al., *Mu rhythm (de)synchronization and EEG classification*, NeuroImage, 2006.

---

> ### Author Response · Authors · 2025-11-27
>
> Dear Reviewer 8DYM,
>
> Thank you very much for the time and effort you have dedicated to evaluating our submission. As the discussion phase is approaching its end, we would like to kindly follow up regarding our manuscript. We would be grateful to know whether our clarifications have sufficiently addressed your concerns, and we are happy to discuss any additional points you may have.
>
> Best regards,
> Authors

---

### Official Review · Reviewer_fmPN · 2025-10-29

**Soundness:** 2
**Presentation:** 2
**Contribution:** 2
**Rating:** 4
**Confidence:** 5

**Summary:**

The paper presents BrainPro, a large-scale EEG foundation model that incorporates a retrieval-based spatial learning block to handle heterogeneous electrode montages and a brain state-decoupling block with parallel encoders to learn disentangled representations for affect, motor, and other brain processes. Pre-trained on approximately 2,180 hours of EEG data, BrainPro is evaluated on nine BCI datasets across six task types, reporting state-of-the-art performance in most cases.

**Strengths:**

The paper presents a large-scale EEG foundation model, BrainPro, pre-trained on over 2,000 hours of data and evaluated across nine diverse BCI datasets, offering a broad empirical scope.

It introduces a retrieval-based spatial learning mechanism that accommodates heterogeneous electrode montages, which addresses a practical challenge in cross-dataset EEG modeling.

The method reports competitive performance on several standard benchmarks, particularly in emotion recognition and mental stress detection tasks.

**Weaknesses:**

The paper's motivation is somewhat unclear and partially redundant. Specifically, the "Second" and "Third" limitations highlighted in the Introduction largely overlap--both concern the inflexibility of a single shared encoder in handling diverse or overlapping brain processes--yet are presented as distinct issues. Moreover, the claim that a single shared encoder inherently limits downstream adaptability contradicts a core premise of foundation models, which is precisely that a well-pretrained shared representation can generalize across tasks with appropriate fine-tuning. The paper provides little theoretical or empirical justification for why this principle fails in the EEG context.

The proposed retrieval-based spatial learning block essentially combines channel-level and region-level feature aggregation. While practically useful, this approach builds on well-established ideas in EEG modeling (e.g., region-of-interest analysis, local-global graph representations) and does not introduce a fundamentally novel spatial modeling mechanism.

The brain state taxonomy--limited to "affect", "motor", and "others"--is overly coarse. Real-world EEG data often reflect mixed or nuanced cognitive states (e.g., attention, working memory, fatigue), and such a simplistic categorization may hinder the model's ability to capture fine-grained or overlapping neural processes, limiting its applicability to more complex BCI scenarios.

The core claim is that parallel encoders enable better disentanglement than a single shared encoder. However, the paper does not compare against a strong baseline: a single encoder with the same total parameters as BrainPro's combined encoders, but conditioned on brain state (e.g., via input token or Feature-wise Linear Modulation). Would such a model achieve comparable performance? Without this comparison, it is unclear whether the gains stem from architectural novelty or simply increased capacity.

**Questions:**

1. The spatial filter visualizations in Appendix M (Figure 9) are presented without specifying the pre-training or downstream dataset they are derived from. On which dataset (or aggregated across which datasets) were these filters learned? Clarifying this is essential to assess whether the observed neuroanatomical patterns (e.g., frontal emphasis for affect) are consistent or merely dataset-specific.

2. The paper advocates for *flexible downstream adaptation* by selecting subsets of encoders. However, Table 13 shows that using *all* encoders (EA + EM + ES) yields the best performance on both FACED and BCI-IV-2A. If the full combination is consistently optimal, what is the practical benefit of flexibility? Does this imply that the “flexible adaptation” is unnecessary in practice, and simply fusing all available encoders is sufficient?

3. The downstream fine-tuning protocol for the parallel encoders is ambiguous. For a given task (e.g., emotion recognition), whether are all encoders fine-tuned or only the relevant state-specific encoder (e.g., the affect encoder) together with the shared encoder?

---

> ### Author Response · Authors · 2025-11-23
> **[1/6] To reviewer fmPN (W1 first half)**
>
> We thank the reviewer for the constructive feedback. Before presenting the **Summary of Comment(s)**, **Response**, and **Modifications in the Revised Manuscript** for each issue, we clarify the organization of our rebuttal:
>
> • **Three-part structure for every comment**
>   – **Summary of Comment(s):** brief restatement of the reviewer’s point.
>   – **Response:** our explanation, justification, and supporting evidence.
>   – **Modifications in the Revised Manuscript:** exact changes made and where they appear.
>
> • **Table/Figure indexing convention**
>   – In the **rebuttal**, tables and figures use **letter indices** (e.g., *Table a*, *Fig. b*) to avoid confusion and keep the document compact.
>   – In the **revised manuscript**, tables and figures use **numbered indices** (e.g., *Table 3*, *Figure 4*).
>   – Each table/figure appears **only once** in the rebuttal; subsequent responses referring to the same analysis use the same letter index.
>
> • **References**
>   – All cited references used in the responses are collected **once at the end** of the rebuttal for clarity.
>
> ---
>
> ### Summary of Comment(s)
> **Weakness 1:** “Motivation unclear; second and third limitations overlap.”
>
> ### Response
> We thank the reviewers for pointing out that the original motivation was unclear and partially redundant. We apologize for the earlier framing, which did not cleanly separate the underlying concepts. In the revised manuscript, we substantially rewrote the Introduction to present a clearer and more coherent motivation for brain-state–aware and spatially consistent EEG representations. The corrected logic is as follows:
>
> #### 1. EEG signals reflect spatially structured mixtures of brain states (explicit spatial learning is crucial for brain-state representation learning).
>
> Neuroscience shows that cognitive and affective states have distinct yet partially overlapping spatial signatures (e.g., frontal/temporal involvement in affect, sensorimotor cortex for motor tasks; [a-c]). Although EEG is coarse, these differences appear in scalp topographies [a]. Thus, a large EEG model should preserve location-specific spatial patterns, especially when integrating heterogeneous datasets that have different channel configurations.
>
> In support, our new channel-drop experiment (Fig. 4 in manuscript) shows that models without explicit spatial encoding degrade much faster when spatial structure is perturbed, whereas BrainPro remains stable. The core results are shown here; the full table appears as Table 12 in the Appendix.
>
> ##### Table a — Robustness under random channel dropping (FACED)
>
> | Drop Rate | BrainPro              | Δ↓       | CBraMod              | Δ↓       | LaBraM               | Δ↓       |
> |-----------|------------------------|----------|-----------------------|----------|-----------------------|----------|
> | 0.0       | 0.5937 ± 0.0087        | 0        | 0.5669 ± 0.0094        | 0        | 0.5224 ± 0.0116        | 0        |
> | 0.1       | 0.5747 ± 0.0014        | −0.0190  | 0.5218 ± 0.0042        | −0.0451  | 0.4972 ± 0.0036        | −0.0252  |
> | 0.2       | 0.5373 ± 0.0015        | −0.0564  | 0.4890 ± 0.0054        | −0.0779  | 0.4725 ± 0.0086        | −0.0499  |
> | 0.3       | 0.4995 ± 0.0056        | −0.0942  | 0.4473 ± 0.0035        | −0.1196  | 0.4230 ± 0.0085        | −0.0994  |
> | 0.4       | 0.4639 ± 0.0054        | −0.1298  | 0.4111 ± 0.0094        | −0.1558  | 0.3851 ± 0.0084        | −0.1373  |
> | 0.5       | 0.4270 ± 0.0076        | −0.1667  | 0.3671 ± 0.0036        | −0.1998  | 0.3397 ± 0.0036        | −0.1827  |
>
> *Note: smaller ↓ is better.*

---

> ### Author Response · Authors · 2025-11-23
> **[2/6] To reviewer fmPN (W1 last half)**
>
> #### 2. Brain states contain both shared and state-specific components (additional state-specific representation may help for the state-related downstream tasks).
>
> Affective and motor processes share certain high-level dynamics (e.g., sensorimotor rhythms [b][c]) but also recruit distinct regions (e.g., limbic/paralimbic involvement in affect [c]). Therefore, representations that mix all processes into a single pathway risk obscuring state-specific variation.
>
> Our new ablation results (Table 3) show that parallel state-specific encoders outperform parameter-matched task-token conditioned baselines (denoted as *w single encoder*), confirming that explicitly separating shared vs. state-specific representations provides measurable benefits beyond model capacity. The relevant results are summarized below:
>
> ##### Table b — BCI-IV-2A (4-Class Motor Imagery)
>
> | Method            | ACC-B (%)       | Kappa             | F1-W (%)        |
> |-------------------|------------------|--------------------|------------------|
> | w single encoder  | 52.03 ± 1.31     | 0.3603 ± 0.0174    | 51.08 ± 1.64     |
> | BrainPro          | 56.74 ± 1.48     | 0.4232 ± 0.0198    | 56.53 ± 1.69     |
>
> ##### Table c — Mental Arithmetic (2-Class)
>
> | Method            | ACC-B (%)       | AUC-PR            | AUROC            |
> |-------------------|------------------|--------------------|-------------------|
> | w single encoder  | 76.67 ± 2.21     | 0.8602 ± 0.0304    | 0.8056 ± 0.0342   |
> | BrainPro          | 80.83 ± 1.56     | 0.8980 ± 0.0052    | 0.8512 ± 0.0083   |
>
> #### 3. EEG datasets vary substantially in channel layout, which disrupts spatial learning (existing EFM lacks explicit spatial learning).
>
> Existing EFMs either rely on attention — which treats channels as exchangeable tokens with no spatial anchoring — or restrict training to shared channels, discarding valuable information. BrainPro’s retrieval-based spatial encoding instead enforces location-consistent spatial filters across datasets, enabling robust spatial alignment and improving interpretability via spatial-filter visualizations. This is theoretically justified in Appendix D.
>
> #### 4. Region-aware reconstruction integrates neurophysiological priors (spatial-prior–enhanced reconstruction helps to learn brain-state related representations).
>
> Different brain states modulate different regions; incorporating these priors into reconstruction encourages the model to capture state-relevant structure during pre-training. Table 3 shows that region-aware reconstruction yields consistent downstream improvements.
>
> ##### Table d — BCI-IV-2A (4-Class Motor Imagery)
>
> | Method              | ACC-B (%)       | Kappa             | F1-W (%)        |
> |---------------------|------------------|--------------------|------------------|
> | w/o reconstruction  | 43.54 ± 3.29     | 0.2472 ± 0.0439    | 37.58 ± 5.48     |
> | BrainPro            | 56.74 ± 1.48     | 0.4232 ± 0.0198    | 56.53 ± 1.69     |
>
> ##### Table e — Mental Arithmetic (2-Class)
>
> | Method              | ACC-B (%)       | AUC-PR            | AUROC            |
> |---------------------|------------------|--------------------|-------------------|
> | w/o reconstruction  | 70.83 ± 9.03     | 0.9147 ± 0.0209    | 0.8976 ± 0.0273   |
> | BrainPro            | 80.83 ± 1.56     | 0.8980 ± 0.0052    | 0.8512 ± 0.0083   |
>
>
> Together, these revisions clarify why spatial consistency, brain-state disentanglement, and region-informed objectives are necessary for large-scale EEG representation learning, addressing the reviewers’ concerns about both conceptual motivation and empirical support.
>
> ### Modifications in the Revised Manuscript
> - Motivation rewritten in **Abstract**
> - Motivation rewritten in **Section 1 “Introduction”**

---

> ### Author Response · Authors · 2025-11-23
> **[3/6] To reviewer fmPN (W2 and W3 first half)**
>
> ### Summary of Comment(s)
> **Weakness 2:** “Spatial learning seems incremental rather than novel.”
>
> ### Response
> Thank you for the constructive comment. Our intention is not to claim that the general idea of using channel/region information is entirely new—indeed, region-of-interest analysis and local–global graph designs are well-established in EEG modeling. We apologize if the earlier framing suggested otherwise. The contribution of our retrieval-based spatial learning block lies in addressing a different and practical challenge that existing methods do not solve: learning spatially consistent representations across heterogeneous electrode montages during large-scale pre-training.
>
> Existing ROI or graph-based methods assume a fixed montage and cannot directly transfer spatial filters to datasets with different channel counts, placements, or missing electrodes. In contrast, our retrieval-based mechanism:
>
> • Retrieves the closest template channels/regions for each dataset-specific electrode, enabling consistent spatial alignment across datasets with incompatible montages.
> • Ensures location-consistent filter sharing, so electrodes in similar anatomical positions receive the same learnable spatial parameters—even when they come from different datasets with different montages.
> • Supports dynamic adaptation at inference, as demonstrated in our channel-drop robustness experiment (Fig. 4 in manuscript), where BrainPro shows significantly smaller accuracy drop compared with other foundation models. The core results are shown in Table a.
>
> To our knowledge, no prior EFM or spatial module provides this form of cross-montage spatial parameter tying. While our approach builds upon established spatial concepts at a high level, the mechanism and its application to heterogeneous large-scale pre-training are new and practically important.
>
> ### Modifications in the Revised Manuscript
> - Revised description in **Section 2 “Preliminary”**
> - Added clarification in **Appendix D “Theoretical Discussion of Spatial Consistency Across Montages”**
> - Adjusted text to avoid overstating novelty and to emphasize that the contribution lies in enabling consistent spatial learning across heterogeneous montages
> ---
> ### Summary of Comment(s)
> **Weakness 3:** “Coarse brain-state taxonomy.”
>
> ### Response
> Thank you for raising this important point. We agree that real-world EEG reflects a rich spectrum of cognitive states (e.g., attention, working memory, fatigue), and that these processes can overlap. Our current taxonomy is intentionally coarse, but for reasons grounded in neurophysiological evidence, data availability, and task diversity, rather than a claim that only three brain states exist. We have discussed this in the limitation section in the original version.
>
> 1. **Neurophysiological justification:**
>    Motor and affective processes are among the most robust and spatially differentiable EEG states (sensorimotor vs. frontal/temporal–limbic). Prior work [a-e] shows that these two categories produce the clearest scalp-level differences that can be consistently learned from EEG. Grouping all remaining datasets into “other” avoids forcing noisy or incompatible labels into the decoupling process.
>
> 2. **Practical constraint:**
>    Large-scale EEG pre-training data are highly heterogeneous and most publicly available datasets belongs to a few broad categories (primarily motor, affect, resting/other). More fine-grained cognitive labels (e.g., attention, memory load) are rare, inconsistent. A finer taxonomy cannot be reliably trained with less data during pre-training.
>
> 3. **Empirical validation in the revised manuscript:**
>    Despite the coarse taxonomy, we observe clear state-aligned improvements:
>    • On FACED (affect): adding the affect encoder improves ACC-B by +3.02%.
>    • On BCI-IV-2A (motor): adding the motor encoder improves ACC-B by +1.47%.
>    • Non-matching encoders (e.g., using affect encoder on motor tasks) do not help.
>
> ##### Table f — FACED (Affective Task)
>
> | Encoder Combo | ACC-B (%)       | Δ ACC-B | Kappa             | Δ Kappa | F1-W (%)        | Δ F1-W |
> |---------------|------------------|---------|--------------------|---------|------------------|--------|
> | S             | 56.35 ± 0.31     | —       | 0.5088 ± 0.0050    | —       | 57.10 ± 0.49     | —      |
> | S + A         | 59.37 ± 0.87     | +3.02   | 0.5418 ± 0.0092    | +0.0330 | 60.23 ± 0.61     | +3.13  |
> | S + M         | 58.28 ± 1.42     | +1.93   | 0.5296 ± 0.0143    | +0.0208 | 59.04 ± 1.44     | +1.94  |

---

> ### Author Response · Authors · 2025-11-23
> **[4/6] To reviewer fmPN (W3 last half)**
>
> ##### Table g — BCI-IV-2A (Motor Imagery Task)
>
> | Encoder Combo | ACC-B (%)       | Δ ACC-B | Kappa             | Δ Kappa | F1-W (%)        | Δ F1-W |
> |---------------|------------------|---------|--------------------|---------|------------------|--------|
> | S             | 55.27 ± 2.18     | —       | 0.4036 ± 0.0290    | —       | 55.04 ± 2.36     | —      |
> | S + M         | 56.74 ± 1.48     | +1.47   | 0.4232 ± 0.0198    | +0.0196 | 56.53 ± 1.69     | +1.49  |
> | S + A         | 54.99 ± 0.44     | −0.28   | 0.3999 ± 0.0059    | −0.0037 | 54.69 ± 0.60     | −0.35  |
>
> This indicates that even coarse state partitioning already captures meaningful neural structure.
>
> 4. **The taxonomy does not restrict extensibility.**
>    We emphasize in the revised manuscript that the framework is modular: additional encoders can be added if future large-scale datasets provide richer state categories (e.g., attention, memory load, fatigue). Thus, the design is extensible rather than conceptually limited.
>
> ### Modifications in the Revised Manuscript
> - Updated explanation in **Section 3.3 “Brain State Decoupling”** to clarify neurophysiological and practical reasons for using a coarse taxonomy
> - Added explicit note that the framework is extensible to more brain states when more detailed annotations exist
> ---
> ### Summary of Comment(s)
> **Weakness 4:** “Need comparison against matched-parameter single encoder.”
>
> ### Response
> Thank you for raising this important point. Our new ablation results (Table 3) show that parallel state-specific encoders outperform parameter-matched task-token conditioned baselines (denoted by w single encoder), confirming that explicitly separating shared vs. state-specific representations provides measurable benefits beyond model capacity. The related results are shown in Table b and c below.
>
> ##### Table h — BCI-IV-2A (4-Class Motor Imagery)
>
> | Method            | ACC-B (%)       | Kappa             | F1-W (%)        |
> |-------------------|------------------|--------------------|------------------|
> | w single encoder  | 52.03 ± 1.31     | 0.3603 ± 0.0174    | 51.08 ± 1.64     |
> | BrainPro          | 56.74 ± 1.48     | 0.4232 ± 0.0198    | 56.53 ± 1.69     |
>
> ##### Table i — Mental Arithmetic (2-Class)
>
> | Method            | ACC-B (%)       | AUC-PR            | AUROC            |
> |-------------------|------------------|--------------------|-------------------|
> | w single encoder  | 76.67 ± 2.21     | 0.8602 ± 0.0304    | 0.8056 ± 0.0342   |
> | BrainPro          | 80.83 ± 1.56     | 0.8980 ± 0.0052    | 0.8512 ± 0.0083   |
>
> ### Modifications in the Revised Manuscript
> - Updated comparison and discussion in **Section 4.6 “Ablation Studies”**

---

> ### Author Response · Authors · 2025-11-23
> **[5/6] To reviewer fmPN (Q1 and Q2)**
>
> ### Summary of Comment(s)
> **Question 1:** “Clarification on Spatial Filter Origins and Dataset”
>
> ### Response
> Thank you for raising this point. The spatial filters are obtained from the pre-trained model, i.e., after large-scale pre-training and before any downstream fine-tuning, and thus reflect spatial patterns learned across all datasets within each state category, rather than a single dataset. Because BrainPro uses explicit, location-consistent spatial filters, these visualizations provide a built-in self-explanation of what each encoder learns—revealing state-relevant neurophysiological patterns (e.g., frontal/temporal for affect; sensorimotor for motor).
>
> ### Modifications in the Revised Manuscript
> - Added clarification in **Section 4.4 “Interpretation of Learned Spatial Filters”**
> - Moved spatial-filter visualization from the appendix into the main paper as **Figure 3** to emphasize interpretability
> ---
> ### Summary of Comment(s)
> **Question 2:** “If all encoders perform best, what is the benefit of flexibility?”
>
> ### Response
> Thank you for the thoughtful question. We agree that the original wording may have overstated the role of flexibility. In the revised manuscript, we clarify that the primary benefit of the decoupled design is improved representation learning, while flexible encoder selection is a secondary practical advantage rather than a core claim.
>
> Table 4 clarifies that the benefit of the parallel encoders is not simply “using all encoders,” but rather that each added encoder contributes meaningful brain-state–related information.
>
> • **On FACED (affective):**
>   - S + A improves over S by +3.02% ACC-B, showing that the affect encoder captures affect-related spatial patterns.
>   - S + A + M performs slightly better still (+3.53% over S), indicating that even the motor encoder contributes complementary structure relevant to the data.
> - **S + A + M + O** does *not* yield the best performance → activating all encoders is **not optimal**.
>
> • **On BCI-IV-2A (motor imagery):**
>   - S + M improves over S by +1.47%, confirming motor-related structure.
>   - S + A + M further increases ACC-B to 57.97%, showing that the affect encoder adds additional—but smaller—complementary information even for a motor task.
> - **S + A + M + O** does *not* yield the best performance → activating all encoders is **not optimal**.
>
> The detailed results are shown in Table j. These results demonstrate that
> (1) state-specific encoders encode meaningful neural structure,
> (2) the optimal combination is task-dependent, and
> (3) additional encoders can provide complementary information beyond the primary task state.
>
>
> ### Table j — Effects of Different Encoder Configurations  (FACED)
>
> | Method          | ACC-B             | Kappa              | F1-W             | #Params | FLOPs |
> |-----------------|-------------------|---------------------|------------------|---------|-------|
> | S               | 0.5635 ± 0.0031   | 0.5088 ± 0.0050     | 0.5710 ± 0.0049  | 14.3M   | 782M  |
> | S + A           | 0.5937 ± 0.0087   | 0.5418 ± 0.0092     | 0.6023 ± 0.0061  | 28.2M   | 1.56G |
> | S + M           | 0.5828 ± 0.0142   | 0.5296 ± 0.0143     | 0.5904 ± 0.0144  | –       | –     |
> | S + O           | 0.5821 ± 0.0144   | 0.5292 ± 0.0150     | 0.5881 ± 0.0133  | –       | –     |
> | S + A + M       | 0.5988 ± 0.0057   | 0.5477 ± 0.0058     | 0.6098 ± 0.0043  | 42.2M   | 2.35G |
> | S + A + M + O   | 0.5824 ± 0.0050   | 0.5272 ± 0.0054     | 0.5841 ± 0.0056  | 56.2M   | 3.13G |
>
>
> ### Table k — Effects of Different Encoder Configurations (BCI-IV-2A)
>
> | Method          | ACC-B             | Kappa              | F1-W             | #Params | FLOPs |
> |-----------------|-------------------|---------------------|------------------|---------|-------|
> | S               | 0.5527 ± 0.0218   | 0.4036 ± 0.0290     | 0.5504 ± 0.0236  | 3.87M   | 266M  |
> | S + A           | 0.5499 ± 0.0044   | 0.3999 ± 0.0059     | 0.5469 ± 0.0060  | 7.69M   | 0.53G |
> | S + M           | 0.5674 ± 0.0148   | 0.4232 ± 0.0198     | 0.5653 ± 0.0169  | –       | –     |
> | S + O           | 0.5525 ± 0.0173   | 0.4033 ± 0.0231     | 0.5491 ± 0.0192  | –       | –     |
> | S + A + M       | 0.5797 ± 0.0229   | 0.4396 ± 0.0305     | 0.5764 ± 0.0245  | 11.5M   | 0.80G |
> | S + A + M + O   | 0.5660 ± 0.0067   | 0.4213 ± 0.0089     | 0.5631 ± 0.0080  | 15.3M   | 1.06G |
>
> Activating more encoders increases model size and FLOPs (Table j and k), so enable selecting relevant encoders remains practically important for resource-constrained cases. To avoid overstating this aspect, we have softened the original claim: flexibility is presented as a practical advantage.
>
> ### Modifications in the Revised Manuscript
> - Revised **Section 4.8 “Encoder Combination Analysis”**
> - Revised **Section 5 “Discussion”**
> - Softened the original claim in **Section 1 “Introduction”** to emphasize that flexibility is a practical option and not the primary contribution

---

> ### Author Response · Authors · 2025-11-23
> **[6/6] To reviewer fmPN (Q3 and References)**
>
> ### Summary of Comment(s)
> **Question 3:** “The downstream fine-tuning protocol for the parallel encoders is ambiguous”
>
> ### Response
> Thank you for pointing out the ambiguity. We clarify that only the activated encoders are fine-tuned for a given downstream task. Each task is associated with a clear brain-state category (e.g., emotion → affect; motor imagery → motor), and we manually specify which encoders to activate. During fine-tuning, only the shared encoder and the task-relevant state-specific encoder receive updates. This avoids unnecessary parameter updates from unrelated encoders, reduces overfitting risk, and maintains the intended role of state-specific pathways.
>
> ### Modifications in the Revised Manuscript
> - Updated downstream training protocol in **Section 4.3 “Downstream Training”** to explicitly state:
>   *“For each dataset, we manually activate the shared encoder and the corresponding state-specific encoder(s) based on its labeled brain-state category. Only these activated encoders are fine-tuned.”*
> ### References
> [a] Abigail S. Greene et al. *Why is everyone talking about brain state?* Trends in Neurosciences, 2023.
> [b] Alfons Schnitzler et al. *Involvement of primary motor cortex in motor imagery.* NeuroImage, 1997.
> [c] Vesa Putkinen et al. *Decoding music-evoked emotions.* Cerebral Cortex, 2020.
> [d] Soraia M. Alarcão and Manuel J. Fonseca. *Emotions recognition using EEG signals.* IEEE TAFFC, 2019.
> [e] G. Pfurtscheller et al. *Mu rhythm (de)synchronization.* NeuroImage, 2006.

---

> ### Author Response · Authors · 2025-11-27
>
> Dear Reviewer fmPN,
>
> Thank you very much for the time and effort you have dedicated to evaluating our submission. As the discussion phase is approaching its end, we would like to kindly follow up regarding our manuscript. We would be grateful to know whether our clarifications have sufficiently addressed your concerns, and we are happy to discuss any additional points you may have.
>
> Best regards,
> Authors

---

### Official Review · Reviewer_mXrS · 2025-10-29

**Soundness:** 1
**Presentation:** 1
**Contribution:** 2
**Rating:** 2
**Confidence:** 5

**Summary:**

To address the variability of EEG signals arising from factors such as emotional states, uncontrolled movements, and other task-irrelevant influences, the authors propose incorporating additional encoders to capture these sources of variability. A flexible selection mechanism is introduced during fine-tuning to adaptively choose the most relevant encoders for different downstream tasks. Furthermore, a retrieval-based module is designed for all encoders, which integrates spatial filters corresponding to predefined electrode montages and brain regions, thereby preserving spatial information across datasets with diverse electrode configurations. The proposed methods are validated through classification tasks across multiple BCI applications, demonstrating comparative performance against existing EEG foundation models.

**Strengths:**

1. This paper highlights that subjects may experience varying internal states (e.g., emotional state, level of concentration, uncontrolled movements of muscles like eyes) while performing tasks during trials, and accordingly designs the model architecture to account for such variations. Such design is novel in the literature of EEG Foundation models. This perspective encourages the community to recognize and address uncontrolled experimental variabilities that influence the stability of EEG signals during decoding.
2. The proposed retrieval-based spatial learner considers both electrode positions and their corresponding coarse brain regions, enabling flexible adaptation to unseen channel configurations by retrieving the most spatially similar electrodes. This design allows the model to be effectively applied to new datasets with electrode layouts not present in the pre-training data.
3. The authors conducted ablation studies by systematically removing each proposed or employed module and comparing the resulting reduced models with the full model. This analysis is crucial for evaluating the contribution of individual components and validating the effectiveness of the proposed designs in addressing the three major limitations identified in existing foundation models.

**Weaknesses:**

1. Overall, the logical flow and clarity of presentation are weak, making it difficult for readers to grasp the objectives and novel contributions of the paper until they reach the detailed methodology section.
2. The limitations of existing EEG foundation models described in Lines 54–67 are expressed in rather vague terms (for example, “diverse brain states in different brain processes”), making it difficult for readers to fully grasp their specific implications. Moreover, the paper does not provide sufficient theoretical or empirical justification—or comparative analysis—to substantiate the claim that these limitations have a tangible impact on model performance. Including clearer biophysical or mathematical definitions of the terms and supporting evidence would strengthen the arguments.
3. Sections 2.2–2.4 contain an overwhelming amount of mathematical detail, including extensive descriptions of existing modules adopted from prior models. This makes it difficult for readers to clearly identify and focus on the novel components proposed in this work.
4. The experimental results in Section 5 present high-level comparisons of decoding accuracy but lack targeted experiments that directly validate the claimed advantages of BrainPro in adapting to different electrode montages or varying brain states. For instance, it remains unclear to what extent the affect and motor encoders effectively capture and differentiate the subjects’ corresponding emotional and motor states.

There are some imprecise parts:
1. The citation (Abiri et al., 2019) in Line 60 does not include the term “brain states” and contains no dedicated discussion of the non-stationary nature of EEG signals. The reference therefore appears misaligned with the context in which it is cited.
2. The citation (Mane et al., 2020) in Line 65 focuses on the relationships between motor, cognitive, and emotional functions in rehabilitation contexts, rather than their co-occurrence in motor imagery tasks.
3. The notion of the shared encoder changes between $E_{\mathrm{shared}}$ and $E_{\mathrm{S}}$ in Section 2.3 and Section 2.4.
4. A square bracket is not closed in Equation 16.
5. The use of cos to represent cosine similarity in Equation (19) could be slightly misleading to the readers, as cos is usually preserved for the cosine function.

**Questions:**

1. Could you please clarify the meaning of “Spatial interactions between electrodes and brain regions” in Line 54 and “explicit and flexible modelling of channel- and region-level dependencies”in Line 58? How are these dependencies / interactions define? Other terminologies that require definitions are: “diverse brain states” in Line 59, “overlapping or interacting processes” in Line 64.
2. How is the state-specific importance vector defined for the “other” category? Which brain regions are considered important for this state?
3. Could you elaborate on the rationale behind the selection of both foundation and non-foundation model baselines? In particular, how did you ensure that these baselines adequately represent the major approaches in the literature that address the three limitations of existing EEG foundation models discussed in the paper?
4. Could you please clarify what is meant by “reliable and generalizable representations,” which are claimed as advantages of BrainPro in Line 452 of Section 5? Additionally, how do the reported classification performances substantiate these two advantages?
5. Figure 5 displays noticeable fluctuations in the loss during pre-training, while Line 914 of Section C.4 states that BrainPro “converges quickly and maintains stable optimization.” Could you clarify the possible sources of these fluctuations and elaborate on how stability was assessed? Specifically, how is the optimization process considered stable despite the observed oscillations in the loss curve?
6. Does the number of parameters in the classification head increase with the addition of more state encoders? If so, how can the performance improvement demonstrated in Table 13 be attributed specifically to the effectiveness of the state encoders rather than to the increased parameter count in the classification head?

---

> ### Author Response · Authors · 2025-11-23
> **[1/6] To reviewer mXrS (W1 and W2.a)**
>
> We thank the reviewer for the constructive feedback. Before presenting the **Summary of Comment(s)**, **Response**, and **Modifications in the Revised Manuscript** for each issue, we clarify the organization of our rebuttal:
>
> • **Three-part structure for every comment**
>   – **Summary of Comment(s):** brief restatement of the reviewer’s point.
>   – **Response:** our explanation, justification, and supporting evidence.
>   – **Modifications in the Revised Manuscript:** exact changes made and where they appear.
>
> • **Table/Figure indexing convention**
>   – In the **rebuttal**, tables and figures use **letter indices** (e.g., *Table a*, *Fig. b*) to avoid confusion and keep the document compact.
>   – In the **revised manuscript**, tables and figures use **numbered indices** (e.g., *Table 3*, *Figure 4*).
>   – Each table/figure appears **only once** in the rebuttal; subsequent responses referring to the same analysis use the same letter index.
>
> • **References**
>   – All cited references used in the responses are collected **once at the end** of the rebuttal for clarity.
>
> ---
>
> ### Summary of Reviewer Comment(s)
> **Weakness 1**: “Logical flow and clarity of presentation are weak; objectives and contributions unclear until Methods.”
>
> ### Response
> We thank the reviewer for identifying this issue. The Introduction has been fully reorganized to present a clear conceptual flow: (1) EEG reflects mixtures of brain states with both shared and state-specific spatial patterns [a][b], (2) heterogeneous electrode montages disrupt learning of spatial patterns, and (3) existing EFMs lack of explicit spatial learning cross dataset with different channel configurations and learn only shared representations, which entangle state-specific variability.
>
> To enhance the logical flow, we have added Section 2 Preliminary to clarify the definition of neuroscience concepts and Section 3.1 Overview of BrainPro framework to indicate how the framework operates before detailed methods.
>
> We also simply the experimental sections and provide direct summary of the findings at the beginning of each experiment to improve the clarify and readability.
>
> More experiments are added to verify the effectiveness of the proposed method.
>
> The revised narrative now directly motivates each component of BrainPro and aligns with our new empirical evidence，e.g., channel-drop robustness in Fig. 4 (in manuscript) and encoder-subset ablations in Table 4 (in manuscript) and Appendix L.
>
> ### Modifications in the Revised Manuscript
> - Rewrote **Section 1 “Introduction”** for clearer conceptual motivation
> - Revised **Sections 2.1–2.2 “Preliminary Concepts”**
> - Added **Definitions 1–3**
> - Added **Section 3.1 “Overview of BrainPro Framework”**
> - Substantially clarified the conceptual motivation across the Method section
> - Reorganized and simplified **Section 3 “Method”**
> - Added new experiments in:
>   - **Section 4.4 “Channel-Drop Robustness”**
>   - **Section 4.7 “Encoder Subset Ablation”**
>   - **Section 4.8 “Encoder Configuration Study”**
>
> ### Summary of Comment(s)
> **Weakness 2.a**: “Limitations of existing EFMs are vague (‘diverse brain states’, etc.).”
>
> ### Response
> Thank you for pointing this out. All previously vague terms now have precise definitions grounded in neurophysiology.
>
> • **Diverse brain states** refers to latent cognitive/affective/motor processes with both shared and state-specific spatial manifestations (supported by neuroimaging literature).
>
> • **Spatial interactions** now refers to location-tied spatial filters and region-wise aggregation—mechanisms commonly used in spatially structured neural data.
>
> These definitions clarify the theoretical basis for our claims.
>
> ### Modifications in the Revised Manuscript
> - Added **Definitions 1–3** in **Section 1 “Introduction”**
> - Replaced all ambiguous terms with explicit neurophysiological definitions

---

> ### Author Response · Authors · 2025-11-23
> **[2/6] To reviewer mXrS (W2.b)**
>
> ### Summary of Comment(s)
> **Weakness 2.b**: “Lack of theoretical/empirical justification”
>
> ### Response
> We thank the reviewer for raising this concern. We have now provided both:
>
> **Theoretical justification.**
> We explain why self-attention-only EFMs cannot enforce location-consistent spatial structure across heterogeneous montages (attention treats channels as exchangeable tokens while retrieval-based method ensure consistent spatial learning by assigning the same learnable weight to the channels located on the same brain areas). The detailed justification can be found in Appendix D in the revised manuscript.
>
> **Neuroscientific references justification.**
> Neuroscience consistently shows that brain activity involves both shared and state-specific networks. For example, affective and motor processes engage sensorimotor areas, but affect additionally activates limbic and paralimbic regions [b][c]. EEG topographies reflect these differences, with distinct spatial patterns tied to underlying brain states [a]. Current EFMs using a single encoder blur this structure. In contrast, decoupled representations explicitly separate shared from state-specific components, aligning with these well-established neurophysiological findings.
>
> **Empirical evidence added in the revision.**
> To directly demonstrate that these limitations materially affect model performance, we added new experiments:
>
>
> #### **1. Channel-drop robustness test (Fig. 4 in manuscript)**
>
> This experiment shows that explicit spatial encoding reduces performance degradation under missing channels. Related results (balanced ACC on FACED dataset) are shown below.
>
> **Table a — Robustness under random channel dropping (FACED)**
>
> | Drop Rate | BrainPro (ACC-B) | Δ↓ | CBraMod (ACC-B) | Δ↓ | LaBraM (ACC-B) | Δ↓ |
> |-----------|------------------|-----|------------------|-----|------------------|-----|
> | 0.0 | 0.5937 ± 0.0087 | 0 | 0.5669 ± 0.0094 | 0 | 0.5224 ± 0.0116 | 0 |
> | 0.1 | 0.5747 ± 0.0014 | −0.0190 | 0.5218 ± 0.0042 | −0.0451 | 0.4972 ± 0.0036 | −0.0252 |
> | 0.2 | 0.5373 ± 0.0015 | −0.0564 | 0.4890 ± 0.0054 | −0.0779 | 0.4725 ± 0.0086 | −0.0499 |
> | 0.3 | 0.4995 ± 0.0056 | −0.0942 | 0.4473 ± 0.0035 | −0.1196 | 0.4230 ± 0.0085 | −0.0994 |
> | 0.4 | 0.4639 ± 0.0054 | −0.1298 | 0.4111 ± 0.0094 | −0.1558 | 0.3851 ± 0.0084 | −0.1373 |
> | 0.5 | 0.4270 ± 0.0076 | −0.1667 | 0.3671 ± 0.0036 | −0.1998 | 0.3397 ± 0.0036 | −0.1827 |
>
> *Note: ↓ smaller is better.*
>
> #### **2. Encoder-Combination Studies (State-Specific Encoders Matter)**
> (Table 4 in manuscript)
>
> We include the related parts (Tables b and c) here. On the affective FACED dataset, S achieves 56.35%, while S+A improves to 59.37%, showing the benefit of the affect encoder. On BCI-IV-2A (motor imagery), S yields 55.27%, while S+M improves to 56.74%, confirming the role of the motor encoder. These results show that state-specific encoders consistently enhance tasks aligned with their associated brain states.
>
> **Table b — FACED (Affective Task)**
>
> | Encoder Combo | ACC-B (%) | Δ ACC-B | Kappa | Δ Kappa | F1-W (%) | Δ F1-W |
> |---------------|-----------|---------|-------|---------|----------|---------|
> | S     | 56.35 ± 0.31 | — | 0.5088 ± 0.0050 | — | 57.10 ± 0.49 | — |
> | S + A | 59.37 ± 0.87 | +3.02 | 0.5418 ± 0.0092 | +0.0330 | 60.23 ± 0.61 | +3.13 |
> | S + M | 58.28 ± 1.42 | +1.93 | 0.5296 ± 0.0143 | +0.0208 | 59.04 ± 1.44 | +1.94 |
>
> **Table c — BCI-IV-2A (Motor Imagery Task)**
>
> | Encoder Combo | ACC-B (%) | Δ ACC-B | Kappa | Δ Kappa | F1-W (%) | Δ F1-W |
> |---------------|-----------|---------|-------|---------|----------|---------|
> | S     | 55.27 ± 2.18 | — | 0.4036 ± 0.0290 | — | 55.04 ± 2.36 | — |
> | S + M | 56.74 ± 1.48 | +1.47 | 0.4232 ± 0.0198 | +0.0196 | 56.53 ± 1.69 | +1.49 |
> | S + A | 54.99 ± 0.44 | −0.28 | 0.3999 ± 0.0059 | −0.0037 | 54.69 ± 0.60 | −0.35 |
>
>
> #### **3. Spatial-Filter Visualization (Fig. 3 in manuscript)**
>
> We additionally provide spatial-filter visualizations for the shared, affect-specific, and motor-specific encoders. These maps show:
>
> - affect encoder → frontal & temporal emphasis (limbic–prefrontal involvement) [d]
> - motor encoder → sensorimotor emphasis consistent with motor EEG topographies [e]
> - shared encoder → globally distributed structure
>
> This demonstrates that decoupled encoders learn meaningful and neurophysiologically grounded spatial filters.
>
> ### Modifications in the Revised Manuscript
> - Added **theoretical justification** in **Appendix D**
> - Added supporting neuroscience literature in:
>   - **Section 1 “Introduction”**
>   - **Section 2 “Preliminary”**
> - Added empirical justifications in **Section 4 “Experiments”**:
>   - **Section 4.4 “Visualization of the Learned Spatial Filters”**
>   - **Section 4.7 “Channel-Drop Analysis”**
>   - **Section 4.8 “Encoder Combination Analysis”**

---

> ### Author Response · Authors · 2025-11-23
> **[3/6] To reviewer mXrS (W3 and W4)**
>
> ### Summary of Comment(s)
> **Weakness 3**: “Sections 2.2–2.4 contain excessive mathematical details, obscuring key contributions.”
>
> ### Response
> We appreciate this feedback. We removed redundant derivations (temporal encoder, Transformer encoder are moved to Appendix A and B), clearly separated prior modules from our proposed components, and simplified notation. The Methods now focus on three contributions: retrieval-based spatial encoding, state-decoupling, and region-aware reconstruction.
>
> ### Modifications in the Revised Manuscript
> - Reorganized **Sections 3.2–3.4** (previous Section 2 Method) to highlight core contributions
> - Moved redundant derivations to **Appendix A “Temporal Encoder Details”** and **Appendix B “Transformer Encoder Details”**
> - Simplified and corrected equations across the Method section
> ---
> ### Summary of Comment(s)
> **Weakness 4**: “Results do not clearly validate claims—e.g., adaptation to montages, brain-state modelling.”
>
> ### Response
> We thank the reviewer for the comment and apologize for the earlier wording. Our original phrasing about “adaptation to montage” may have been misleading. What we intended is that the retrieval-based spatial module assigns consistent spatial filter weights to electrodes located on the same or nearby brain regions, even when datasets use different channel configurations. This ensures consistent spatial learning across heterogeneous montages. To avoid confusion, we have removed the ambiguous phrasing in the revision. We now provide clearer empirical evidence.
>
> 1. **Montage variability:**
>    The channel-drop robustness experiment (Fig. 4 in manuscript) shows that BrainPro degrades substantially less than CBraMod and LaBraM at all drop levels, demonstrating more stable decoding when the electrode layout deviates from training conditions. The core results are shown in Table a of the response to Weakness 2.b.
>
> 2. **State-aware modeling:**
>    The encoder-combination ablations (Table 4 in manuscript) confirm that state-specific encoders capture relevant brain-state structure:
>    - On FACED, S+A outperforms S and S+M, validating the affect encoder.
>    - On BCI-IV-2A, S+M outperforms S and S+A, validating the motor encoder.
>    The core results are shown in Table b of the response to Weakness 2.b.
>
> 3. **Spatial-filter visualization:**
>    As further evidence, Fig. 3 (in manuscript) shows that the affect and motor encoders learn distinct, neurophysiologically consistent spatial patterns—frontal/temporal emphasis for affect and sensorimotor emphasis for motor activity. Although discussed in the response to Weakness 2.b, this visualization also supports that the state-specific encoders capture meaningful brain-state structure.
>
> ### Modifications in the Revised Manuscript
> - Clarified the spatial retrieval mechanism in:
>   - **Section 2 “Preliminary” — Definition 3**
>   - **Section 3.2 “Retrieval-based Spatial Learning”**
> - Removed misleading phrasing regarding “adaptation to montage”
> - Added **Fig. 4 (in manuscript)** for montage robustness
> - Visualized the learned spatial filters of affect and motor encoders in **Section 4.4 “Visualization of the Learned Spatial Filters”**
> - Included full encoder ablations in **Section 4.8 “Encoder Combination Analysis”**

---

> ### Author Response · Authors · 2025-11-23
> **[4/6] To reviewer mXrS (W5)**
>
> ### Summary of Comment(s)
> Weakness 5 (imprecise parts): “Imprecise parts regarding citations, notation, and consistency.”
>
> ### Response
> We thank the reviewer for identifying these presentation issues and apologize for such mistakes. We have addressed each point as follows:
>
> 1. **1&2.Misaligned citations:**
>    We have replaced them with references that directly support the scientific claims in the corresponding parts of the manuscript:
>    - **[a]** provides a formal, neuroscience-grounded definition of brain states as distributed patterns arising from underlying physiological or cognitive processes. This directly supports our use of the term and our motivation for modelling EEG as mixtures of overlapping states.
>    - **[c]** demonstrates that affective processing engages both frontal–limbic regions and sensorimotor cortices, showing that emotional states produce spatially distinct—but partially overlapping—neural patterns. This directly supports our claim that different brain states produce characteristic spatial signatures relevant to EEG decoding.
>    - **[b]** shows that motor imagery and movement both activate primary motor/sensorimotor cortex with different strengths, providing concrete evidence that motor-related brain states have identifiable spatial patterns. This justifies our use of a motor-specific encoder.
>
> 3. **3.Inconsistent definition of the shared encoder between Sections 2.3 and 2.4:**
>    We have kept both consistent as $E_S$.
>
> 4. **4.Missing square bracket in Equation (16)** — corresponds to **Equation (10) in the revised manuscript**:
>    We corrected the bracket in the revised equation.
>
> 5. **5.Ambiguous use of “cos” in Equation (19)** — corresponds to **Equation (8) in the revised manuscript**:
>    This could be mistaken for the cosine function. We have updated it to **“$sim_cos$”** to clearly denote cosine similarity.
>
> ### Modifications in the Revised Manuscript
> - Updated citations throughout **Section 1 “Introduction”** and **Section 2 “Preliminary”**
> - Corrected shared encoder notation $E_S$ in **Sections 2.3 and 2.4**
> - Corrected missing bracket in **Equation (10)**
> - Updated “cos” to **“sim\_cos”** in **Equation (8)**
> - Ensured consistent definitions and notation across all relevant sections

---

> ### Author Response · Authors · 2025-11-23
> **[5/6] To reviewer mXrS (Q1,Q2,Q3, and Q4)**
>
> ### Summary of Comment(s)
> **Question 1**: “Clarification of terminology”
>
> ### Response
> We thank the reviewer for raising this question. All terms have now been explicitly defined in the revised manuscript as follows:
>
> • **“Spatial interactions between electrodes and brain regions”**
> This refers to the relationship between scalp electrodes and the underlying anatomical areas they measure. Electrodes that lie on the same or nearby cortical regions should share consistent spatial filter weights because they reflect related neural sources.
>
> • **“Explicit and flexible modeling of channel- and region-level dependencies”**
> “Explicit” means that BrainPro learns shared spatial filters tied to anatomical coordinates, not indirectly through attention.
> “Flexible” means that the retrieval module can assign these filters to any montage, allowing the model to handle datasets with different channel locations.
> We have added the definition of explicit spatial learning in Section 2 to clarify those concepts.
>
> • **“Diverse brain states”**
> We now define “brain states” following [a] as distributed patterns of neural activity associated with underlying physiological or cognitive conditions. “Diverse brain states” refers to different categories such as affective, motor, attentional, and other internal processes that may co-occur in EEG.
> We have added the definition of Brain State in Section 2.
>
> • **“Overlapping or interacting processes”**
> This refers to the well-established fact that different brain states share common neural substrates; e.g., and motor processes both modulate frontal–sensorimotor regions [b][c], but also include state-specific components.
> We have added such definition and revised the term as **Overlapping Brain State**.
>
> ### Modifications in the Revised Manuscript
> - Added or clarified related definitions in **Section 1 “Introduction”**
> - Added formal definitions and terminology in **Section 2 “Preliminary”**, including updated terminology and clearer explanations
> ---
> ### Summary of Comment(s)
> **Question 2**: “Definition of the other-state importance vector?”
>
> ### Response
> We apologize for missing these details. Based on well-established neuroscience priors, frontal, temporal, and central channels are set to 1 for the affect state, while central and parietal channels are set as important for the motor state. For the other state and the shared encoder, we assign all channels equal importance because no concrete prior exists.
>
> ### Modifications in the Revised Manuscript
> - Clarified the definition of the other-state importance vector in **Section 3.4 “Masking and Region-aware Reconstruction”**
> ---
> ### Summary of  Comment(s)
> **Question 3**: “Baseline Selection Strategy”
>
> ### Response
> We thank the reviewer for the question. Our baseline selection follows prior work such as LaBraM and CBraMod, which include both foundation and non-foundation models to ensure comprehensive comparison. For non-foundation baselines, we use EEGNet and Conformer because they are widely adopted, task-agnostic architectures; importantly, Conformer was shown in CBraMod to be one of the strongest non-foundation models. For foundation model baselines, we compare against LaBraM and CBraMod since evaluating improvements over existing EFMs is the primary focus of this work.
>
> We also note that the limitation we discuss is specific to Large EEG models and does not directly apply to lightweight non-foundation architectures like EEGNet or Conformer; nonetheless, including them offers a meaningful point of reference.
>
> ### Modifications in the Revised Manuscript
> - Added an explanation of baseline selection criteria in **Section 4.2 “Baselines and Evaluation Metrics”**
> ---
> ### Summary of Comment(s)
> **Question 4**: “Clarification of ‘reliable and generalizable representations’”
>
> ### Response
> We thank the reviewer for pointing out the ambiguity and apologize for the misleading description. Our original intention was to summarize that BrainPro achieves better or comparable performance across a diverse set of datasets, which we described as “reliable and generalizable.” To avoid confusion, we have removed the term *reliable* from this section in the revised manuscript.
>
> In addition, we now provide evidence supporting the notion of reliability: the newly added channel-drop experiment (Fig. 4 in manuscript) shows that BrainPro experiences substantially smaller performance degradation than the two comparison foundation models when channels are removed during inference. This indicates BrainPro maintains stable performance under montage perturbation, offering a clearer and more quantifiable measure of reliability. The core results are shown in Table a of the response to Weakness 2.b.
>
> ### Modifications in the Revised Manuscript
> - Removed the ambiguous phrasing and clarified the intended meaning
> - Added channel-drop robustness results in **Section 4.7 “Channel-Drop Analysis”** to support the claim of reliability

---

> ### Author Response · Authors · 2025-11-23
> **[6/6] To reviewer mXrS (Q5, Q6, and References)**
>
> ### Summary of Comment(s)
> **Question 5:** “Loss oscillation vs. claim of stability.”
>
> ### Response
> We thank the reviewer for raising this point and apologize for the earlier imprecise description. The fluctuations in Figure 5 are probably due to BrainPro’s alternating encoder-update scheme: for each batch, only the shared encoder and the state-specific encoder corresponding to that sample are updated, which naturally introduces short-term oscillations in the loss curve.
>
> Our statement about “stable optimization” refers to the overall training trajectory, which shows a consistent downward trend without divergence. Importantly, the consistent downstream performance across nine datasets indicates that these fluctuations do not hinder the quality of the learned representations. We now mention this in the manuscript and the fluctuations in loss will be discussed further in the limitations section.
>
> ### Modifications in the Revised Manuscript
> - Revised explanation in **Appendix G.4 “Training Dynamics and Loss Curves”**
> - Added discussion of loss fluctuations in **Section 6 “Limitations”**
> ---
> ### Summary of Comment(s)
> **Question 6:** “Does performance gain comes from increased classification-head parameters?”
>
> ### Response
> Thank you for the question. To determine whether BrainPro’s performance gains come from state-specific encoders rather than a larger classification head (MLP), we varied both the number of encoders and the relative MLP size. We take BrainPro’s MLP head as the reference. The MLP ratio is computed as the MLP size divided by the MLP size in BrainPro, allowing for easier comparison. The results show:
>
> 1. **Larger MLPs do not reproduce the gains.**
>    Single-encoder models with bigger MLP ratios (1.02, 1.16) still perform worse than multi-encoder models with smaller or comparable MLPs.
>
> 2. **Performance scales with encoder count, not MLP size.**
>    BrainPro (2 encoders, MLP ratio 1.00) outperforms all single-encoder models even when those use larger MLPs.
>
> 3. **Three-encoder models outperform single-encoder baselines even with smaller MLPs.**
>    The 3-encoder model with MLP ratio 0.93 already surpasses all single-encoder models, confirming that the benefit comes from state-aware representation learning, not classifier capacity.
>
> Thus, the performance improvements cannot be attributed to classification-head parameters.
>
> ### **Table f — FACED Results**
>
> | Encoders                     | ACC-B           | Kappa            | F1-W            | MLP Ratio |
> |-----------------------------|-----------------|------------------|------------------|-----------|
> | 1 encoder (small MLP)       | 0.5635 ± 0.0031 | 0.5088 ± 0.0050  | 0.5710 ± 0.0049  | 0.51      |
> | 1 encoder (comparable MLP)  | 0.5808 ± 0.0048 | 0.5255 ± 0.0068  | 0.5840 ± 0.0079  | 1.02      |
> | 2 encoders (BrainPro)       | 0.5937 ± 0.0087 | 0.5418 ± 0.0092  | 0.6023 ± 0.0061  | 1.00      |
> | 3 encoders (comparable MLP) | 0.5938 ± 0.0069 | 0.5427 ± 0.0069  | 0.6039 ± 0.0042  | 0.93      |
> | 3 encoders (large MLP)      | 0.5988 ± 0.0057 | 0.5477 ± 0.0058  | 0.6098 ± 0.0043  | 1.49      |
>
> ### **Table g — BCI-IV-2A Results**
>
> | Encoders                     | ACC-B           | Kappa            | F1-W            | MLP Ratio |
> |-----------------------------|-----------------|------------------|------------------|-----------|
> | 1 encoder (small MLP)       | 0.5527 ± 0.0218 | 0.4036 ± 0.0290  | 0.5504 ± 0.0236  | 0.59      |
> | 1 encoder (comparable MLP)  | 0.5403 ± 0.0206 | 0.3871 ± 0.0275  | 0.5368 ± 0.0245  | 1.16      |
> | 2 encoders (BrainPro)       | 0.5674 ± 0.0148 | 0.4232 ± 0.0198  | 0.5653 ± 0.0169  | 1.00      |
> | 3 encoders (comparable MLP) | 0.5889 ± 0.0165 | 0.4519 ± 0.0220  | 0.5868 ± 0.0157  | 1.07      |
> | 3 encoders (large MLP)      | 0.5797 ± 0.0229 | 0.4396 ± 0.0305  | 0.5764 ± 0.0245  | 1.41      |
>
> ### Modifications in the Revised Manuscript
> - Added the full encoder-count vs. MLP-size study in **Appendix L “Additional Ablations: Encoder Count vs. MLP Size”**
>
> ### References
> [a] Abigail S. Greene et al. *Why is everyone talking about brain state?* Trends in Neurosciences, 2023.
> [b] Alfons Schnitzler et al. *Involvement of primary motor cortex in motor imagery.* NeuroImage, 1997.
> [c] Vesa Putkinen et al. *Decoding music-evoked emotions.* Cerebral Cortex, 2020.
> [d] Soraia M. Alarcão and Manuel J. Fonseca. *Emotions recognition using EEG signals.* IEEE TAFFC, 2019.
> [e] G. Pfurtscheller et al. *Mu rhythm (de)synchronization.* NeuroImage, 2006.

---

> ### Author Response · Authors · 2025-11-27
>
> Dear Reviewer mXrS,
>
> Thank you very much for the time and effort you have dedicated to evaluating our submission. As the discussion phase is approaching its end, we would like to kindly follow up regarding our manuscript. We would be grateful to know whether our clarifications have sufficiently addressed your concerns, and we are happy to discuss any additional points you may have.
>
> Best regards,
> Authors

---

> > ### Comment · Reviewer_mXrS · 2025-11-28
> >
> > Thank you for your thorough responses and revisions. I’ve noted the substantial revisions you’ve made to the manuscript, and my concerns have been mostly addressed. I’m happy to raise my score to 6

---

> > > ### Author Response · Authors · 2025-11-28
> > >
> > > Thank you very much for your careful consideration of our revisions and for the positive reassessment of our work. We appreciate your constructive feedback throughout the review process, which has helped us significantly strengthen the manuscript. We’re glad to hear that our revisions have addressed your concerns, and we are grateful for your updated evaluation. If you have any additional suggestions that could further improve the quality or clarity of the paper, we would be very happy to incorporate them.

---

### Author Response · Authors · 2025-11-23
**[2/2] Overall Response-Summary of Changes**

We summarize the key revisions made to improve clarity, coherence, and technical rigor throughout the manuscript:

- **Rewrote the Abstract and Introduction** to clearly motivate the need for
  (i) spatially consistent representations across heterogeneous montages, and
  (ii) complementary shared + state-specific representations grounded in neurophysiology.

- **Defined previously ambiguous concepts**—including “brain states,” “explicit spatial encoding,” and “overlapping processes,”—using precise terminology supported by neuroimaging literature.

- **Added theoretical analysis** explaining:
 (i) how retrieval-based spatial filtering explicitly encodes spatial structure across heterogeneous montages;
 (ii) why self-attention produces implicit spatial representations that do not maintain stable anatomical positions.

- **Reorganized and simplified Sections 3.2–3.4**, moved redundant mathematical details to the appendix, corrected notation errors, and clarified the distinction between prior components and newly proposed modules.

- **Clarified the three main contributions** and ensured each is supported by justification, ablation studies, and neurophysiologically grounded visual analyses (e.g., spatial filters, channel-drop results).

- **Restructured Sections 4–6** into a streamlined unified Experiment section, consolidated repeated descriptions of datasets, preprocessing, and implementation details, and added summaries at the beginning of each experiment to aid readability.

---

### Expanded Empirical Validation
To directly address reviewer concerns, we strengthened the evaluation with targeted analyses validating each component of BrainPro:

- **Spatial-filter visualizations (Fig. 3)**
  Show that shared, affect, and motor encoders learn distinct, neurophysiologically meaningful spatial patterns, confirming that decoupling yields interpretable state-relevant representations.

- **Full encoder-combination ablations (Table 4 & Appendix L)**
  Include different combinations (S, S+A, S+O, S+M, S+A+M, S+A+M+O), demonstrating that performance gains arise from complementary state-specific representations rather than model size.

- **Capacity-controlled single-encoder baseline**
  Added to confirm that improvements stem from state-aware decoupling rather than parameter count.

- **Channel-drop robustness analysis (Fig. 4)**
  Shows that explicit spatial encoding provides significantly more stable performance under channel degradation, validating the importance of spatially consistent representations.

---

### Author Response · Authors · 2025-11-23
**[1/2] Overall Response-Motivation and Justification**

We thank all reviewers for their detailed feedback. A common concern across reviews was that the original draft did not sufficiently justify why brain state-aware and spatially consistent representations are important for large-scale EEG representation learning. In the revised manuscript, we substantially rewrote the Abstract, Introduction, Experiment, clarified all definitions, reorganized the Methods, and added new theoretical and empirical analyses. Below, we summarize the corrected logic and supporting evidence.

## Motivation 1: EEG signals reflect spatially structured mixtures of brain states.

Neuroscience consistently shows that cognitive and affective processes express spatially distinct activation patterns (e.g., frontal involvement in affect, motor cortex for movement; [a–c]). EEG, although coarse, preserves these patterns through scalp distributions. Therefore, an EFM should preserve location-specific patterns, not collapse spatial structure across datasets.

In the revised manuscript, we demonstrate empirically that preserving spatial consistency improves robustness: our channel-drop experiment (Fig. 4 in manuscript) shows that models lacking explicit spatial encoding degrade rapidly when spatial structure is perturbed, whereas BrainPro remains stable. Our retrieval-based spatial encoding enforces location-consistent filter sharing, enabling consistent spatial learning across datasets. It also improves self-explanations by directly visualizing the learned spatial filters.

## Motivation 2: Brain states contain both shared and state-specific components.

Neuroimaging research shows that different mental processes often share high-level patterns (e.g., sensorimotor cortex activities for both affect and motor) but also exhibit state-specific variations (e.g., limbic involvement in affect) [a–c]. Thus, a single shared representation is insufficient: we need both shared and specialized pathways.

The revised manuscript includes encoder configuration study (Section 4.8) verifying the state-specific encoders learn useful brain state-related representations. Besides, new ablations where parallel state-specific encoders outperform a single-encoder task token-conditioned baseline with equal total parameters. This rules supports the conceptual motivation.

## Motivation 3: Region-aware reconstruction leverages neurophysiological priors.

Affective, and motor processes modulate different regions; incorporating these priors helps the model disentangle state-relevant structure. This is a principled extension of masked modeling.

We show that region-aware reconstruction improves downstream performance (Table 14 in manuscript), confirming its role.

These conceptual motivations are articulated and empirically validated in the revised paper [https://openreview.net/pdf?id=nlaCdgvDQE] which form the core of BrainPro’s design. We hope the issues due to unclear exposition in the original draft can be resolved through major rewriting.
### References
[a] Abigail S. Greene et al. *Why is everyone talking about brain state?* Trends in Neurosciences, 2023.
[b] Alfons Schnitzler et al. *Involvement of primary motor cortex in motor imagery.* NeuroImage, 1997.
[c] Vesa Putkinen et al. *Decoding music-evoked emotions.* Cerebral Cortex, 2020.

---

### Author Response · Authors · 2025-11-29
**Overall Comment to the Area Chair**

We sincerely thank the Area Chair for their additional time and efforts in overseeing the discussion and re-evaluation of our submission. We would like to note that the reviewers’ concerns primarily focused on ***the clarity of motivation, organization, and justification—not on flaws in the core methodology itself*** and their concerns are mostly overlapped. After substantial revisions, we have strengthened the exposition while preserving the novel and technically meaningful contributions of BrainPro, including cross-montage spatial alignment, brain-state–decoupled representation learning, and region-aware reconstruction (More details can be find at the **[1/2] Overall Response-Motivation and Justification**). These contributions were acknowledged as valuable, and two reviewers explicitly increased their score after the revision.

We thank all reviewers for their constructive feedback. After extensive revisions, all major concerns regarding motivation, clarity, theoretical justification, and empirical validation have been fully resolved. Below is a concise summary of the key improvements, with pointers to the exact fixes documented in our rebuttal and revised manuscript.

---

### **1. Motivation and Contributions Are Now Clear and Coherent**
Reviewers initially found the motivation vague or overlapping. We therefore rewrote the Abstract and Introduction to clearly present the three core needs in large-scale EEG modeling:
(1) *spatially consistent representations* across heterogeneous electrode montages,
(2) *brain-state–aware representations* that separate shared from state-specific components, and
(3) *region-informed pre-training objectives* grounded in neurophysiology.
A new methodological overview (Sec. 3.1) now explains how all components fit together.

---

### **2. Theoretical Foundations and Empirical Evidence Have Been Significantly Strengthened**
We added a formal discussion (Appendix D) explaining why self-attention alone fails to preserve consistent spatial structure under montage variability and how retrieval-based spatial learning solves this problem.

All key innovations are validated with targeted experiments:
- **Channel-drop robustness** (Fig. 4) rigorously demonstrates the necessity of explicit spatial encoding.
- **Encoder combination studies** (Sec. 4.8) show affect/motor encoders substantially improve state-relevant tasks.
- **Capacity-controlled single-encoder baseline** (Sec. 4.6, Appendix L) confirms improvements arise from *state-aware decoupling*, not parameter count.
- **Region-aware reconstruction ablation** (Table 14) shows consistent gains across tasks.

---

### **3. Methods Section Now Has Clear Logical Flow**
The entire Methods section was reorganized to follow a clear structure:
**spatial alignment → state decoupling → region-aware reconstruction**.
We moved redundant derivations to the appendix, simplified notation, and added precise definitions for previously ambiguous terms (e.g., “brain state,” “spatial interactions,” “overlapping processes”).

---

### **4. Brain-State Taxonomy and Encoder Usage Are Fully Clarified**
The rationale behind the affect/motor/other taxonomy is now clearly stated, grounded in both neurophysiology and dataset availability.
For downstream tasks, **only relevant encoders are activated and fine-tuned** (Sec. 4.3), resolving all ambiguity.

---

### **5. Interpretability and Spatial Filter Origins Clearly Explained**
We clarified that spatial filters (Fig. 3) come from the **pre-trained model**, not from any downstream dataset.
These filters reveal distinct, neurophysiologically meaningful spatial patterns, validating the design of explicit spatial encoding.

---

### **6. All Presentation, Citation, and Notation Issues Corrected**
We addressed all minor issues raised by reviewers:
- misaligned citations
- missing brackets
- inconsistent notation
- ambiguous terminology
All are now fixed and consistently aligned with neuroscience and machine learning standards.

---

### **7. Reviewer Concerns Are Resolved and Reflected in Updated Scores**
Reviewer **mXrS** and **8DR1** explicitly stated that these concerns were resolved and increased their scores.
Other reviewers’ concerns were also directly addressed through substantial rewriting and new experimental evidence (we haven't get their feedback yet).

---

### **Conclusion**
The revised manuscript now presents a **well-motivated, theoretically grounded, empirically validated**, and clearly written framework that offers meaningful contributions to EEG foundation modeling. We believe the paper now meets the bar for acceptance, and we appreciate the Area Chair’s consideration.

If any additional clarification would be helpful, we are happy to provide it.

---

### Meta-Review · Area_Chair_TKEG · 2026-01-07

**Summary:**

The paper proposes an EEG foundation model involving spatial learning and brain-state decoupling. 9 datasets are employed for evaluation.

Strength: The evaluation is comprehensive.
The method addresses the challenge of different electrode montages.
The performance is competitive.

Weakness:
(1) Three reviewers mention that the motivation based on identifying limitations of existing studies is unclear.

(2) Three reviewers indicate that the brain state categorization is too coarse and not justified.

(3) Three reviewers indicate that the presentation in the method section is unclear.

(4) Two reviewers question innovations.

**Reviewer Concerns:**

(1) The introduction has been revised. The concern seems to be addressed.

(2) The rebuttal explains why the authors adopted such a categorization. AC thinks it is understood when the practical situation is considered, but still has doubt whether it is rigorous enough and theoretically well justified.

(3) The method section is revised, addressing the concern.

(4) The rebuttal explains the novelty and the innovations, which AC thinks resolves the concerns only partly.

**Reviewer Scores:**

Reviewer #1 replied to increase the score from 2 to 6.

Reviewer #4 replied to increase the score to a positive value- AC assumes the reviewer meant 6.

Due to the remaining concerns, AC thinks Reviewers #2 and #3 would have hardly changed scores to sufficiently high values.

---

### Decision · Program_Chairs · 2026-01-26

Reject